# Reproducing and Dissecting Denoising Language Models for Speech Recognition

## Abstract

Denoising language models (DLMs) have been proposed as a powerful alternative to traditional language models (LMs) for automatic speech recognition (ASR), motivated by their ability to use bidirectional context and adapt to a specific ASR model's error patterns. However, the complexity of the DLM training pipeline has hindered wider investigation. This paper presents the *first independent, large-scale empirical study* of DLMs. We build and release a *complete, reproducible pipeline* to systematically investigate the impact of key design choices. We evaluate dozens of configurations across multiple axes, including various data augmentation techniques (e.g., SpecAugment, dropout, mixup), different text-to-speech systems, and multiple decoding strategies. Our comparative analysis in a common subword vocabulary setting demonstrates that *DLMs outperform traditional LMs*, but only after a distinct compute tipping point. While LMs are more efficient at lower budgets, DLMs scale better with longer training, mirroring behaviors observed in diffusion language models. However, we observe smaller improvements than those reported in prior character-based work, which indicates that the DLM's performance is conditional on factors such as the vocabulary. Our analysis reveals that a key factor for improving performance is to condition the DLM on *richer information from the ASR's hypothesis space*, rather than just a single best guess. To this end, we introduce *DLM-sum, a novel method for decoding from multiple ASR hypotheses*, which consistently outperforms the previously proposed DSR decoding method. We believe our findings and public pipeline provide a crucial foundation for the community to better understand, improve, and build upon this promising class of models. The code is publicly available at https://anonymous.4open.science/r/2025-dlm/.

## 1 Introduction

Automatic speech recognition (ASR) systems often rely on external language models (LMs) to refine initial hypotheses by leveraging vast amounts of text-only data. Traditionally, these LMs are autoregressive, processing text from left to right, which limits their ability to use the full context of a sentence when correcting an error.

An alternative is the denoising language model (DLM) (Gu et al., 2024), an encoder-decoder architecture designed to perform error correction directly in the text domain. It operates by taking a complete hypothesis from an ASR model, which may contain recognition errors, as input and generating a fully corrected version as its output. DLMs are motivated by two key theoretical advantages: they can leverage the full bidirectional context of a noisy ASR hypothesis to make more informed corrections, and they can be directly trained on the specific error patterns of an upstream ASR model. Prior work demonstrated state-of-the-art results using this method with a character-level vocabulary.

To be effective, a DLM must be trained on a massive dataset of (noisy hypothesis, correct text) pairs. While such pairs can be generated from standard transcribed audio corpora, their scale is often limited. To overcome this, the approach pioneered in prior work is to leverage large text-only corpora. This is achieved by first synthesizing audio from the correct text using text-to-speech (TTS); an ASR model then transcribes this synthetic audio to generate the corresponding noisy hypothesis. However, this full pipeline – combining TTS synthesis, ASR inference, and extensive data augmentation – is highly complex. This complexity, combined with the lack of a public implementation, has cre-

ated a significant barrier to entry, hindering wider adoption, independent verification, and a deeper understanding of the model's sensitivities. We make the following contributions:

- **A reproducible open-source pipeline:** We build and release the first complete pipeline for DLM training and inference, providing a robust and reproducible baseline to accelerate future research.

- **A systematic empirical study:** We conduct a large-scale study of the DLM design space, evaluating dozens of configurations across multiple axes including data augmentation techniques, TTS systems, and decoding strategies.

- **A novel decoding method:** We introduce **DLM-sum**, a *novel decoding method* that conditions the DLM on multiple ASR hypotheses in a single pass, improving robustness and consistently outperforming the DSR decoding method from prior work.

- **State-of-the-art on LibriSpeech and Loquacious:** We achieve state-of-the-art (SOTA) results on the LibriSpeech benchmark under the strict condition of using only the official LibriSpeech training data for all components (ASR, DLM, and TTS). Similarly, we demonstrate the effectiveness of DLMs on the Loquacious dataset.

- **Comparative analysis with language models:** We provide a direct, comparative analysis against a traditional LM and find that, in a common subword vocabulary setting, *our best DLM outperforms our best traditional LM*. However, we observe *smaller improvements* than those reported in prior character-based work (Gu et al., 2024), which indicates that the DLM's performance is *conditional on factors such as the vocabulary*.

- **Scaling Laws and Diffusion Parallels:** We analyze the compute-performance trade-off, identifying a tipping point where DLMs surpass autoregressive LMs. We discuss similarities to diffusion models.

- **Novel analytical insights:** Our in-depth analysis uncovers several non-obvious model behaviors. We identify that the DLM's performance consistently improves when it is conditioned on *richer information from the ASR's hypothesis space* than a single best guess. We demonstrate this through the success of our *DLM-sum* decoding method and promising results from our exploratory *dense k-probability input to the DLM*. This finding also provides a compelling hypothesis for the performance gap to prior character-based work, as character sequences may inherently offer a more fine-grained representation of errors than a single, incorrect subword token.

## 2 DENOISING LANGUAGE MODEL

Let $a_1^S$ represent a correct token sequence and let $\tilde{a}_1^{\tilde{S}}$ be a corresponding recognized (potentially noisy) hypothesis from an ASR model. The objective of a denoising language model (DLM) is to learn the conditional probability distribution $p_{\text{DLM}}(a_1^S \mid \tilde{a}_1^{\tilde{S}})$ of the correct sequence $a_1^S$ given the noisy hypothesis $\tilde{a}_1^{\tilde{S}}$. The DLM is implemented as an attention-based encoder-decoder (AED) model. The encoder reads the *entire* input hypothesis $\tilde{a}_1^{\tilde{S}}$, and the decoder then autoregressively generates the corrected sequence $a_1^S$. The specific architectural details of our implementation are described in Section 3.2.

### 2.1 DECODING STRATEGIES

To generate a final corrected hypothesis, the DLM's output distribution, $p_{\text{DLM}}(a_1^S \mid \tilde{a}_1^{\tilde{S}})$, is integrated into a search algorithm. The objective is to find an optimal output sequence $\overset{*}{a}_1^{\tilde{S}}$ according to a scoring function, which typically combines the DLM's score with the original ASR model's score. We investigate several such decoding strategies.

**Greedy Decoding.** Following Gu et al. (2024), our greedy decoding decision rule is:

$$\overset{*}{a}_1^{\tilde{S}} = \arg\max_{a_1^S} p_{\text{DLM}}(a_1^S \mid \hat{a}_1^{\hat{S}}), \qquad (1) \qquad \hat{a}_1^{\hat{S}} = \arg\max_{\tilde{a}_1^{\tilde{S}}} p_{\text{ASR}}(\tilde{a}_1^{\tilde{S}} \mid x_1^T). \qquad (2)$$

Equation (1) is approximated with label-synchronous greedy search. Equation (2) is approximated with framewise argmax followed by removing repeated labels and blanks in case of CTC (3.2).

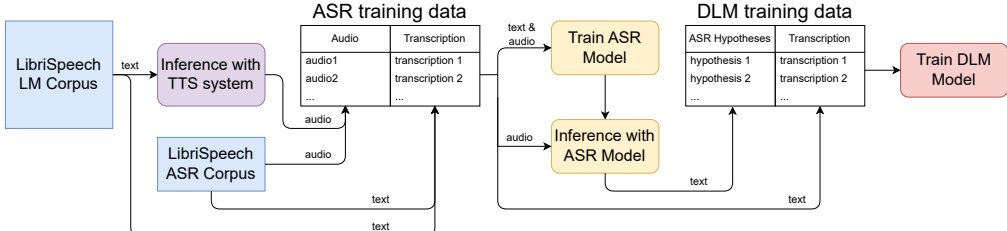

Figure 1: Pipeline for training data generation. We introduce additional data augmentation in the "Inference with ASR model" and "Train DLM model" steps.

**DSR Decoding.** The denoising speech recognition (DSR) decoding method uses

$$\hat{\overset{*}{a}}_1^{\overset{*}{S}} = \underset{a_1^S \in \text{hyps}}{\arg\max}\Big(\log p_{\text{ASR}}(a_1^S \mid x_1^T) + \lambda_{\text{DLM}} \cdot \log p_{\text{DLM}}(a_1^S \mid \hat{a}_1^{\hat{S}}) - \lambda_{\text{prior}} \cdot \log p_{\text{prior}}(a_1^S)\Big) \quad (3)$$

with $\hat{a}_1^{\hat{S}}$ as before (Equation (2)) and rescores on

$$\text{hyps} := \text{n-best}[p_{\text{DLM}}(\cdot \mid \hat{a}_1^{\hat{S}})] \cup \text{m-best}[p_{\text{ASR}}(\cdot \mid x_1^T)]. \quad (4)$$

The DSR decoding as introduced in Gu et al. (2024) was slightly simpler: only using the DLM n-best list for rescoring and not including the prior probability term. The prior probability is the same as we use in LM rescoring (see Section 3.2 and Appendix D.1.3). The scales $\lambda_{\text{DLM}}, \lambda_{\text{prior}} \in \mathbb{R}^+$ are tuned on LibriSpeech dev-other. In our case, we use time-synchronous beam search for the ASR model and label-synchronous beam search for the DLM (in Equation (4); see Appendix C).

**DLM-Sum Decoding.** Previous work (Gu et al., 2024) has only investigated to feed the single best ASR hypothesis into the DLM for decoding (as in greedy decoding and DSR decoding). Instead of just using the single best ASR hypothesis $\hat{a}_1^{\hat{S}}$ in Equation (2), in *DLM-sum decoding*, we want to take multiple ASR hypotheses into account to approximate

$$p_{\text{sum}}(a_1^S \mid x_1^T) = \sum_{\tilde{S}, \tilde{a}_1^{\tilde{S}}} p_{\text{DLM}}(a_1^S \mid \tilde{a}_1^{\tilde{S}}) \cdot p_{\text{ASR}}(\tilde{a}_1^{\tilde{S}} \mid x_1^T). \quad (5)$$

We approximate this sum as

$$p_{\text{sum}'}(a_1^S \mid x_1^T) = \sum_{\tilde{a}_1^{\tilde{S}} \in \text{n-best}[p_{\text{ASR}}(\cdot \mid x_1^T)]} p_{\text{DLM}}(a_1^S \mid \tilde{a}_1^{\tilde{S}}) \cdot \Big(p_{\text{ASR}}(\tilde{a}_1^{\tilde{S}} \mid x_1^T)/Z\Big) \quad (6)$$

with $Z = \sum_{\tilde{a}_1^{\tilde{S}} \in \text{n-best}[p_{\text{ASR}}(\cdot \mid x_1^T)]} p_{\text{ASR}}(\tilde{a}_1^{\tilde{S}} \mid x_1^T)$. We use time synchronous beam search and deduplication to generate the n-best list of ASR hypotheses. The final decision rule becomes

$$\hat{\overset{*}{a}}_1^{\overset{*}{S}} = \underset{a_1^S}{\arg\max}\Big(\log p_{\text{ASR}}(a_1^S \mid x_1^T) + \lambda_{\text{DLM}} \log p_{\text{sum}'}(a_1^S \mid x_1^S) - \lambda_{\text{prior}} \cdot \log p_{\text{prior}}(a_1^S)\Big), \quad (7)$$

which is approximated by label-synchronous beam search on the joint scores, unlike DSR decoding which uses rescoring. See Appendices C and C.9 for more details on the search procedure.

Note that we recover a one-pass version of the DSR decoding decision rule if we only use a single ASR hypothesis in the sum, and the greedy decision rule if we further let $\lambda_{\text{prior}} = 0$, $\lambda_{\text{DLM}} \to \infty$ and beam size 1. So we consider this method a generalization of the previous two. Typically we use a beam size of 12 for the one-pass search, and an ASR n-best list size of 20.

## 3 EXPERIMENTAL DESIGN

### 3.1 DLM TRAINING DATA SOURCES AND GENERATION PIPELINE

Our primary goal is to generate (noisy hypothesis, correct text) pairs to train the DLM. We use the text-only LibriSpeech LM corpus as the source for the correct text transcriptions. Figure 1 illustrates

our two-stage process for generating the corresponding noisy hypotheses. First, we synthesize audio from the source text using a TTS system. Second, this synthetic audio is transcribed by a pre-trained ASR model to produce the noisy hypothesis. We perform TTS and ASR inference in a single forward pass, which avoids the need to store large intermediate audio files on disk. In addition to this synthetic data, we generate hypotheses from the real audio of the LibriSpeech ASR training corpus.

## 3.2 MODELS

**ASR Models.** We use a non-autoregressive Conformer-based (Gulati et al., 2020) connectionist temporal classification (CTC) model (Graves et al., 2006) for ASR. We use three different vocabularies: Character (char)[1], SentencePiece[2] with 128 subwords (spm128) and SentencePiece with 10240 subwords (spm10k). We train all ASR models on LibriSpeech ASR 960h training data and optionally also with TTS audios generated from the LibriSpeech LM text data in a 1:1 ratio. All Conformer models have 16 layers, model dimension 1024, feedforward dimension 4096 and 8 attention heads. See Appendix D.1.1 for model and training details. The ASR baseline performances and model sizes are shown in Table D.1.

**Language Models.** A standard language model can be combined with the ASR model in a similar way to DSR decoding (Section 2.1):

$$\hat{a}_1^{*\tilde{S}} = \arg\max_{a_1^S} \Big( \log p_{\text{ASR}}(a_1^S \mid x_1^T) + \lambda_{\text{LM}} \cdot \log p_{\text{LM}}(a_1^S) - \lambda_{\text{prior}} \cdot \log p_{\text{prior}}(a_1^S) \Big) \tag{8}$$

The prior probability $p_{\text{prior}}(a_1^S)$ is estimated from the ASR model (Appendix D.1.3). We use either two-pass rescoring or one-pass search (Appendix C).

We use Transformer-based (Vaswani et al., 2017) Llama-based (Touvron et al., 2023) attention-based decoder-only language models, trained on the LibriSpeech LM dataset. We use the same vocabulary as we do in our ASR models (mostly spm10k). We train models with 8 and 32 layers respectively, model dimension 1024, feedforward dimension 4096, and 8 attention heads. More details are given in Appendix D.1.2. Our results for both two and single pass rescoring are shown in Table D.3.

**TTS Models.** We train a Glow-TTS model (Kim et al., 2020) on the train-clean-460h subset of the Librispeech ASR training data. To allow for direct comparison with Gu et al. (2024), we also use the same pre-trained YourTTS (Casanova et al., 2022) model. The YourTTS model has been trained on the LibriTTS (Zen et al., 2019) and CML-TTS (Oliveira et al., 2023) datasets. Both TTS models are stochastic and have length scale and temperature $\tau$ (or noise scale) which can be adjusted during inference to control the speed and diversity of the generated audio. See Appendix D.1.4 for more details. ASR WER results for hypotheses generated with Glow-TTS and YourTTS are shown in Table D.8.

**Denoising Language Models.** Our DLMs (Section 2) share the Llama-based (Touvron et al., 2023) architecture of our LMs (Section 3.2) but utilize an encoder-decoder structure with cross-attention. Our DLMs have 24 encoder layers, 8 decoder layers, model dimension 1024, feedforward dimension 4096 and 8 attention heads. We use the same vocabulary as we do in our ASR models (mostly spm10k). We compare greedy decoding, DSR decoding and DLM-sum decoding. See Appendix D.1.5 for further details.

## 3.3 DATA AUGMENTATION STRATEGIES

During the combined TTS + ASR inference (Section 3.1), we can apply various data augmentation techniques to expose the DLM to a wider range of errors. This is especially important as the ASR models have overfitted significantly to the TTS audios, resulting in very low WERs of about 2%, while errors on real audio in the validation and test sets is higher, with up to 4.8% WER (cf. Table D.1). See Appendix D.2 for more details.

---

[1]Char. vocab.: A case-insensitive alphabet, space, apostrophe, and a special end-of-sentence token.

[2]SPM vocab.: A data-driven subword vocabulary generated using the SentencePiece algorithm (Kudo & Richardson, 2018).

**Early ASR checkpoints.** We use intermediate checkpoints from the ASR training process in which the ASR system has not yet converged, and use those hypotheses as training data for the DLM. Results for different checkpoints are shown in Table D.9. Even for epoch 10 out of 100, the WER is reasonably low with 12.4% on dev-other.

**SpecAugment.** We use SpecAugment (Park et al., 2019) during ASR inference as shown by Gu et al. (2024). See Appendix D.2.2 for details and Table D.10 for hypotheses WERs.

**Dropout.** We use dropout (Srivastava et al., 2014) at inference time to generate diverse hypotheses, following prior work (Hrinchuk et al., 2020). However, instead of keeping a fixed dropout rate, we randomly sample a dropout rate for each input sequence from a uniform distribution $\mathcal{U}(p_{min}, p_{max})$. Results for $p_{min} = 0.0$ and different $p_{max}$ are shown in Figure D.3. Both ASR models respond similarly to increasing dropout, with WER increasing slowly at first, and then faster beyond 50%.

**Token Substitution.** We sample the random substitution rate for each sentence $p \sim \mathcal{U}(p_{min}, p_{max})$, and then each token in that sentence is substituted with a random token from the vocabulary with probability $p$. Training data WER for different Token substitution rates is shown in Table D.12. We use rates that increase the hypotheses WER up to 55%.

**Mixup.** We linearly interpolate the spectrogram of the current audio sequence with the spectrograms of $n$ other randomly chosen sequences from a buffer (see Appendix D.2.5 for details).

**Sampling from ASR Model.** Instead of taking the greedy decoding output of the ASR model, we sample from the ASR model using a variant of top-k sampling (details in Appendix D.2.6). For $k = 32$, the hypotheses WER increases to 10% on dev-other.

**Combining Multiple Data Augmentation Techniques.** We combine multiple data augmentation techniques to further increase the diversity of the generated hypotheses. We construct multiple combination variants, ordered in increasing amounts of data augmentation applied, where `baseline` uses no data augmentation at all, `low` uses a small amount, `medium` uses a moderate amount, and the `high` configuration uses a higher amount, using the best parameter for every data augmentation method as determined by individual DLM training experiments, reaching 60% WER on hypotheses on dev-other. More variants, the exact configurations and their respective hypotheses WERs are specified in Appendix D.2.8.

### 3.4 Alternative DLM Inputs: Dense k-Probability

So far we have restricted ourselves to selecting a single label sequence as the ASR hypothesis (Equation (2)). The ASR model, however, produces a substantially richer output in the form of a probability distribution over all possible labels for each audio frame. Storing the full probability distribution on disk is infeasible so we restrict ourselves to the top $k$ most probable labels for each label position. Then we use a weighted embedding of these $k$ labels as the input to the DLM encoder. See Appendix D.3 for details.

## 4 Results and Analysis

### 4.1 DLM vs. Standard Language Model

We use the 32 layer standard LM and compare it to our DLM with 24 encoder and 8 decoder layers[3]. LMs and DLM performance is compared in Table 1 (see Appendix E.1 for further comparisons). When training for 10 epochs, the *DLM using DLM-sum decoding outperforms the standard LM* in both one-pass and rescoring modes[4]. These results represent the *state-of-the-art for the LibriSpeech benchmark* under a strict data constraint[5].

---

[3]Both models have a model dimension of 1024. The standard LM has 422M parameters, while the DLM has 466M parameters. Note that a larger standard LM with 1280 model dimension and 663M parameters was slightly worse in WERs, see Table E.1.

[4]We trained both with 5 and 10 epochs, and we chose the best result for each model for this comparison. When only trained for 5 epochs, there is no performance difference between DLM and LM.

[5]A slightly better performance can be achieved with model dimension 1280, that result is shown in Table 2.

Table 1: Standard LM vs DLM performance comparison. The ASR model was trained with or without TTS data, spm10k vocab. The DLM uses data augmentation configuration `low`.

| ASR trained with TTS | LM | Decoding | WER [%] | | | |
|---|---|---|---|---|---|---|
| | | | dev-clean | dev-other | test-clean | test-other |
| No | None | greedy | 2.29 | 5.02 | 2.42 | 5.33 |
| | Standard | rescoring | 1.93 | 4.18 | 2.09 | 4.50 |
| | | one-pass | 1.85 | 3.93 | 2.00 | 4.27 |
| | DLM | greedy | 2.49 | 4.63 | 2.41 | 5.19 |
| | | DSR | 1.76 | 3.95 | 1.91 | 4.37 |
| | | DLM-sum | 1.68 | 3.70 | 1.83 | 4.15 |
| Yes | None | greedy | 1.75 | 4.13 | 2.03 | 4.44 |
| | Standard | rescoring | 1.59 | 3.57 | 1.80 | 3.84 |
| | | one-pass | 1.56 | 3.41 | 1.73 | 3.70 |
| | DLM | greedy | 2.31 | 4.06 | 2.32 | 4.57 |
| | | DSR | 1.49 | 3.43 | 1.79 | 3.70 |
| | | DLM-sum | 1.49 | 3.29 | 1.72 | 3.53 |

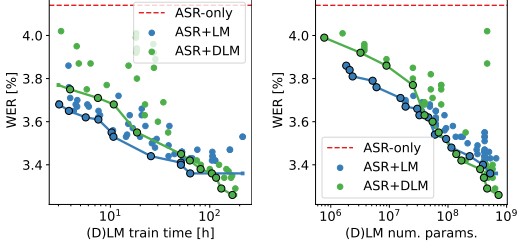

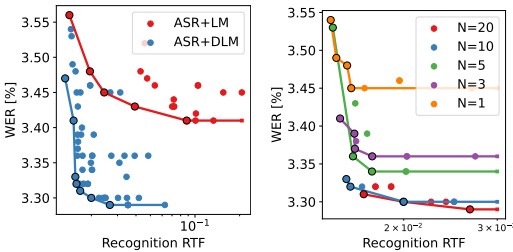

Figure 2: Various DLM and standard LM sizes and training epochs, resulting in different training compute budgets. Results on LibriSpeech dev-other, using DLM-sum and one-pass standard LM decoding.

(a) Comparing standard LM and DLM recognition speed in terms of real-time factor (RTF).

(b) DLM-sum recognition speed w.r.t. number of ASR hypotheses used.

Figure 3: Recognition speed comparisons.

We note that Gu et al. (2024) reports stronger absolute numbers, however, their system utilizes a TTS model trained on external data. Other factors likely also contribute to the performance gap, such as the different vocabulary (we use spm10k, they use char; see Appendix B for a detailed comparison and discussion of differences).

To analyze the scaling behavior of DLMs in comparison to standard LMs, we run various experiments with different model sizes and training epochs, resulting in different training compute budgets. Results by training time and number of parameters are shown in Figure 2. There is a tipping point, both in training time and model size, after which DLMs start to outperform standard LMs, while for smaller compute budgets standard LMs are better. This is similar to recent observations for diffusion language models under a fixed data-constrained setting (Prabhudesai et al., 2025; Ni et al., 2025) (Appendix G.1). Overfitting occurs for standard LMs after a certain point, consistent to previous findings (Ni et al., 2025).

Despite the pure DLM training time, the DLM requires more total preprocessing time and its pipeline is more complex compared to standard LM training, specifically the TTS model and DLM training data generation. Although using TTS data for ASR training is beneficial in general (Table 1).

Figure 3a compares the recognition speed of standard LMs and DLMs. We see that DLMs are generally faster than standard LMs in one-pass decoding, as they perform well already with smaller beam sizes (E.3.1) and the encoder can be run in parallel for all ASR hypotheses over all frames.

We further test our DLM approach on the **Loquacious dataset** (Parcollet et al., 2025). Results are shown in Table 3. We see that the DLM improves over the ASR baseline and standard LM. See Appendix E.6 for details.

Table 2: Comparison to state-of-the-art results and other related work on LibriSpeech test sets. Error correction-based approaches are in the middle section. For details on best model, see Appendix E.5.

| System | External Data | test-clean [%] | test-other [%] |
|---|---|---|---|
| Conformer (Gulati et al., 2020) | - | 1.9 | 3.9 |
| E-Branchformer + ILME (Kim et al., 2023) | - | 1.81 | 3.65 |
| LAS + SC + LM (Guo et al., 2019) | - | 4.28 | - |
| Hrinchuk et al. (2020) | BERT | 3.5 | 9.27 |
| N-best T5 (Ma et al., 2023a) | T5 | 2.53 | 6.27 |
| LLM-based correction (Pu et al., 2023) | ChatGPT, more datasets | 1.3 | 3.4 |
| Denoising LM (Gu et al., 2024) | YourTTS | 1.5 | 3.3 |
| Conformer + larger DLM (Ours) | - | 1.70 | 3.44 |

Table 3: Trained ASR model, (D)LM on Loquacious, WERs [%] on Loquacious evaluation sets.

| Method | Loquacious | | LibriSpeech | | CommonVoice | | VoxPopuli | | Yodas | |
|---|---|---|---|---|---|---|---|---|---|---|
| | dev | test | dev | test | dev | test | dev | test | dev | test |
| ASR only | 6.45 | 7.16 | 4.08 | 4.26 | 9.20 | 11.17 | 6.61 | 6.46 | 11.98 | 12.24 |
| ASR + LM | 5.63 | 6.44 | 3.35 | 3.56 | 7.20 | 9.02 | 6.30 | 6.30 | 12.40 | 12.89 |
| ASR + DLM | 5.52 | 6.26 | 3.40 | 3.58 | 7.24 | 9.07 | 6.02 | 5.99 | 11.43 | 11.78 |

Table 4: Training data generation strategy comparison, using the best setting for each individual augmentation, and some combined augmentation schemes, to train the DLM.

| Method | Setting | WER [%] | | | | Details Table |
|---|---|---|---|---|---|---|
| | | dev-clean | dev-other | test-clean | test-other | |
| Baseline | `baseline` | 1.55 | 3.67 | 1.84 | 4.00 | E.3 |
| Inc. TTS noise | $(0.3, 1.5)$ | 1.51 | 3.61 | 1.76 | 3.83 | E.7 |
| Combined TTS | Glow-TTS + YourTTS | 1.57 | 3.64 | 1.82 | 3.91 | E.9 |
| Early ASR Chkpt. | Epoch 40 | 1.52 | 3.57 | 1.76 | 3.88 | E.10 |
| SpecAugment | Time + freq. | 1.49 | 3.47 | 1.76 | 3.76 | E.11 |
| Dropout | $(0.0, 0.5)$ | 1.49 | 3.52 | 1.81 | 3.85 | E.12 |
| Token Sub. | 20% | 1.50 | 3.47 | 1.80 | 3.79 | E.13 |
| Mixup | $\lambda_{max} = 0.4$ | 1.51 | 3.58 | 1.77 | 3.87 | E.14 |
| ASR sampling | $k = 16$ | 1.55 | 3.67 | 1.83 | 3.95 | E.15 |
| Resplit Subw. | - | 1.63 | 3.81 | 1.95 | 4.13 | E.16 |
| Combined aug. | `low` | 1.51 | 3.40 | 1.74 | 3.66 | E.3 |
| | `medium` | 1.56 | 3.42 | 1.75 | 3.72 | |
| | `high` | 1.67 | 3.56 | 1.81 | 3.89 | |

## 4.2 Impact of Training Data Generation Strategies

Table 4 summarizes the impact of data augmentation techniques on DLM performance. Most configurations are quite similar in performance. The combination of multiple augmentation methods gives the best performance. Choosing the best configuration from every individual ablation (`medium` and `high` combined configurations) does not lead to optimal DLM performance. Rather our `low` configuration gives the best results. See Appendix E.2.1 for more details and further results.

**Augmentation Methods.** When *increasing the TTS noise* (E.2.3) beyond usual values to achieve a stronger variation, we see a slight increase in performance until the maximum scale of 1.5. Any *variation in the length scale* had no effects. *Combining both TTS systems* (E.2.4) gave only smaller improvements, which is surprising as YourTTS is trained on additional data beyond LibriSpeech. This is inconsistent with results from Gu et al. (2024), where the combination gives consistent improvement.

*SpecAugment* (E.2.6), *dropout* (E.2.7), *token substitution* (E.2.8) and *mixup* (E.2.9) each display some improvements over the baseline. Also, using an *earlier ASR checkpoint* (E.2.5), in this case from epoch 40 instead of 100, yields slightly better DLM training.

*Sampling ASR outputs* (E.2.10) does not bring improvements over the baseline. Furthermore, *re-splitting subwords* (E.2.11) even causes minor degradations.

**Training Data Conditions.** There is a $7 - 23\%$ relative improvement across all decoding conditions when using an ASR system trained with TTS data during the recognition process. In contrast, for the generation of DLM training data it does not matter if we use the baseline ASR system or an ASR system trained with additional TTS data. See Appendix E.2.13 for details.

We do not see substantial improvements when generating different DLM training data for each training epoch by using new seeds for the augmentation methods. We assume that the initial training data amount already has a sufficient size and variety. See Appendix E.2.14 for details.

We observe minimal degradation when removing the original LibriSpeech ASR training data from the DLM training. Still, over-sampling the LibriSpeech data up to a point that one third of the data is the original ASR data does not change the performance. See Appendix E.2.15 for details. In comparison, when only using the original LibriSpeech ASR training data and no TTS data, we see a substantial degradation, even when generating large amount of data using data augmentation methods. See Appendix E.2.16 for details.

**Relevance of TTS-ASR Data.** Previous work used heuristics such as masking or random substitutions for pretraining of error correction models (Hrinchuk et al., 2020; Dutta et al., 2022; Ma et al., 2023a). We compare the DLM performance when trained on data generated via TTS-ASR versus heuristic error generation (E.2.17) in Table E.23: Training on TTS-ASR data is clearly superior, consistent to the findings of Gu et al. (2024).

### 4.3 Analysis of Inference and Model Behavior

**Decoding Methods.** Table 1 compares our different decoding methods (DLM greedy, DSR decoding, DLM-sum). We note that the DLM greedy WER can be even worse than the ASR baseline (without LM). This is different to Gu et al. (2024), where the DLM greedy decoding clearly outperforms the ASR baseline. We assume that the different vocabulary (subwords vs. characters) is an important contributing factor to this difference (Appendix G.2). The DLM with DSR decoding is already slightly outperforming a standard LM. The DLM-sum decoding consistently outperforms DSR decoding. The prior contributes only minimally to DSR and DLM-sum performance. Beam size of 8 and 32 ASR hypotheses (in the sum in Equation (6)) seem to be sufficient for onepass DLM-sum decoding[6]. Figure 3b shows the DLM-sum recognition speed with respect to the number of ASR hypotheses used. We see that using more ASR hypotheses is generally faster and better. See Appendix E.3.1 for further comparisons.

**Search and Model Errors.** Counted search errors are $\leq 1\%$ across the board, while model error rates are significantly higher. Including the DLM beam (DSR) in rescoring nearly halves the Oracle WER versus using the ASR beam alone (DLM rescore). See Appendix E.3.2 for details.

**Training convergence behaviour.** Convergence of DSR and DLM-sum decoding methods during training is quite stable, already surpassing the ASR baseline after the first epoch of training. Greedy decoding is an unreliable indicator for training convergence. See Appendix E.3.3 for details.

**WER Distribution.** We group individual sentences into bins based on their WER, and compare the distribution of sentence WERs across different training data configurations. Notably, configurations with similar overall WER can exhibit distinct sentence-level distributions. See Appendix E.3.4 for details.

**Correlations.** We investigate correlations between training data metrics and DLM performance. For this, we collect the results of all ablation experiments, compute metrics on the their training data and final models, and create scatter plots (Figures E.20 to E.23). Greedy and DSR performance are weakly correlated, if at all. DSR and DLM-sum performance are strongly correlated. Our best DLMs have training data with a WER between 10% and 20%, but this is not a strong predictor of DLM performance. The correlation to DLM performance is a bit stronger with the measured LM perplexity of the training data. We calculate the expected calibration error (ECE) (Lee & Chang, 2021), and find that our DLMs are quite well calibrated, with ECE values below 0.1. There does not seem to be a strong correlation between ECE and DLM performance however. The mean entropy

---

[6]When scales are tuned properly.

of the DLM output distribution has an almost linear relationship with ECE, and thus is not strongly correlated with DLM performance either. See Appendix E.3.5 for details.

**Error Analysis by Categorization.** We analyze substitution errors, which make up the majority of mistakes, by categorizing them, and comparing ASR and DLM performance across these categories. DLMs struggle more with rare words compared to common or medium-frequency words, but generally correct errors across all word frequencies about equally well. When categorized by part of speech (e.g., proper nouns, verbs, etc.), apart from statistically insignificant outliers, the DLM improves errors across all categories in a fairly uniform manner. See Appendix E.3.7 for details.

**Correction vs. Degradation Analysis.** While advanced decoding methods like DSR and DLM-sum show clear improvements over greedy search, our analysis reveals a surprising reason for their effectiveness. The primary benefit of DSR over a greedy search is a *drastic reduction in miscorrections* (newly introduced errors by the DLM over the ASR output) while maintaining a similar number of correct fixes. DLM-sum then further improves performance by also significantly increasing the number of correct fixes, with only a slight increase in miscorrections. See Appendix E.3.10 for details.

**Error Examples.** Manual inspection of the recognition output reveals that, under greedy decoding, hallucinations impact our DLMs quite significantly. In one instance, just four sentences alone account for an increase of 0.35% in absolute WER on the test-other set. We also note that DLMs show a tendency to correct more errors in longer sentences, and refrain from making corrections in shorter sentences. See Appendix E.3.11 for details.

**Softmax Temperature.** Through our correlation study (Appendix E.3.5), we hypothesize that a higher entropy of the DLM output distribution leads to better performance in model combination decoding methods like DLM-sum. Artificially increasing the entropy of a baseline model to match that of our best model through softmax temperature does not improve performance though (Appendix E.3.6).

**Out-of-Domain Generalization.** DLMs generalize worse on out-of-domain (OOD) evaluation sets compared to standard LMs. See Appendix E.4.6, Table E.44.

**Generalization to Other ASR Models.** We further test the generalization of the DLM to ASR outputs from another ASR model, different from the one used for DLM training data generation. Results are shown in Table E.45. We see that the DLM improves over the ASR baseline and standard LM even for the other ASR model. See Appendix E.4.7 for details.

### 4.4 Ablations on Model and Training Variations

**Randomness.** We train three different DLMs with unique random seeds for weight initialization to assess the statistical significance of our results. Performance on DSR and DLM-sum decoding stays within $\pm 0.06\%$ absolute WER around the mean. Greedy decoding results are less reliable. See Appendix E.4.1 for details.

**Different Vocabularies.** We compare DLMs trained on char, spm128 and spm10k vocabularies. The spm10k vocabulary performs best, followed by spm128 and then char. This likely stems from the lower baseline performance of spm128 and char ASR models. See Appendix E.4.3 for details. We further discuss character vs. subword vocabularies in Appendix G.2.

**Joint AED and CTC Model.** We design an auxiliary CTC loss in the DLM encoder for error correction, which is able to make (limited) insertion, substitution and deletion corrections. On the trained DLM, we run evaluations with joint AED and CTC decoding. Overall, we see no improvement over AED-only DLMs. See Appendix E.4.4 for details.

**Dense k-Probability Input to DLM.** Following Section 3.4, we save the top $k = 5$ labels and probabilities from the ASR output. Results are shown in Table 5. DLM-sum is not applicable here, because the dense k-probability model does not condition on ASR hypotheses, but on the ASR output probabilities. We see much better greedy performance, and DSR roughly matches that of the baseline DLM-sum performance. This shows that the dark knowledge (Hinton et al., 2014) in the dense ASR output distribution provides additional useful information for the DLM. This confirms our hypothesis that the DLM can benefit from a richer input representation. See Appendix E.4.5 for further details.

Table 5: Dense $k$-probability input to DLM. Baseline model uses standard labelwise argmax from ASR model, while $k = 5$ experiment sees top 5 labels from label-synchronous search of ASR model. DLM-sum decoding is not applicable for dense-input models.

| DLM | $k$ | Decoding | WER [%] | | | |
|---|---|---|---|---|---|---|
| | | | dev-clean | dev-other | test-clean | test-other |
| None | - | greedy | 1.75 | 4.13 | 2.03 | 4.44 |
| Baseline | - | greedy | 1.95 | 4.05 | 2.25 | 4.60 |
| | | DSR | 1.54 | 3.54 | 1.77 | 3.84 |
| | | DLM-sum | 1.45 | 3.45 | 1.76 | 3.69 |
| Dense Input | 5 | greedy | 1.49 | 3.66 | 1.75 | 3.91 |
| | | DSR | 1.48 | 3.48 | 1.72 | 3.71 |
| | 10 | greedy | 1.47 | 3.65 | 1.73 | 3.85 |
| | | DSR | 1.45 | 3.54 | 1.72 | 3.73 |

## 5 RELATED WORK

The core question is how the large amount of text-only data can be leveraged to improve performance of speech recognition. A popular approach is to train a separate LM on the text-only data, and combine it with the ASR model during inference time through shallow fusion, deep fusion or cold fusion (Toshniwal et al., 2018). A LM can also be used during training of the ASR model with minimum WER training (Prabhavalkar et al., 2018; Peyser et al., 2020; Meng et al., 2021b).

A straightforward approach to use text-only data is to generate synthetic audio using TTS systems, and then train the ASR model on this additional data, similar to backtranslation in neural machine translation (Hayashi et al., 2018; Rossenbach et al., 2020).

Another approach is to use error correction models (Tanaka et al., 2018; Hrinchuk et al., 2020; Peyghan et al., 2025). To train such a model, one needs pairs of (noisy hypothesis, correct) text. Audio is typically fed through an ASR model which then generates the noisy hypotheses. Text-to-speech (TTS) can additionally be used to increase the amount of audio data available for error correction model training (Guo et al., 2019; Gu et al., 2024). A comparison of our results to prior work is shown in Table 2.

Advances in large language models (LLMs) have also inspired research into using them for error correction (Ma et al., 2023b; 2025; Tur et al., 2024). LLMs can be used to pick the best hypothesis from an n-best list, or to directly generate corrected text from a single hypothesis.

## 6 CONCLUSION

Our comprehensive analysis leads to several findings:

- We investigated DLMs and show that they can outperform traditional LMs under data-constrained settings, given enough compute budget. The scaling behavior is similar to diffusion LMs.
- Our novel DLM-sum decoding method consistently outperforms greedy and DSR decoding.
- ASR + DLM decoding is faster than ASR + LM decoding.
- We achieve state-of-the-art results on LibriSpeech and Loquacious under a data-constrained setting.
- Providing the DLM with richer information about the ASR hypothesis space is beneficial, as shown by two findings: The improvements of dense $k$-probability input to the DLM in DSR decoding and the consistent improvements of DLM-sum decoding. We assume that using character-based hypotheses has a similar effect.
- We provide a fully open-source, reproducible pipeline to reproduce all the numbers reported in this work, i.e. for training the ASR models, TTS models, LMs and DLMs, data generation with TTS, all the used decoding strategies, and all the evaluations.

See Appendix G for further discussions.

# 7 REPRODUCIBILITY STATEMENT

We spend great effort to make our work reproducible. We release all code used for training and evaluating our models, as well as all code for the complete pipeline, as well as the best model checkpoints and generated data. The released code can generate every single number reported in this paper, including all of the analysis. The code is publicly available at https://anonymous.4open.science/r/2025-dlm/.

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

## CONTENTS

Table B.1: Overview of main differences and similarities between our DLM implementation and that of Gu et al. (2024).

| Aspect | Our work | Gu et al. (2024) |
|---|---|---|
| Vocabulary | spm10k, spm128, char | char |
| DLM Architecture | Transformer enc-dec | Transformer enc-dec |
| Encoder layers | 24 | 16 |
| Decoder layers | 8 | 4 |
| Model dim | 1024 | 1280 |
| Size | 466M | 484M |
| Positional Emb | Rotary | Sinus |
| Feedforward | SwiGLU | ReLU |
| Batch Size | 20k tokens | 160k tokens (distributed) |
| GPUs | $1 \times$ H100 96GB | $8 \times$ A100 80GB |
| LR Schedule | One Cycle | Step decay |
| ASR architecture | Conformer-CTC | Transformer CTC |
| Other | - | Dropout and Layer dropout 10% |

## A ABBREVIATIONS AND NOTATION

**ASR**      Automatic Speech Recognition
**LM**      Language Model
**DLM**      Denoising Language Model
**TTS**      Text-To-Speech
**WER**      Word Error Rate
**DSR**      Denoising Speech Recognition
**POS**      Part Of Speech
**CTC**      Connectionist Temporal Classification (Graves et al., 2006)
**Conformer**      Convolutional Transformer (Gulati et al., 2020)

$x_1^T = x_1 \dots x_T$      input feature sequence of length $T$ (audio features)
$a_1^S = a_1 \dots a_S$      label sequence of length $T$
$\tilde{a}_1^{\tilde{S}} = \tilde{a}_1 \dots \tilde{a}_{\tilde{S}}$      noisy label sequence of length $\tilde{S}$ from the ASR model
n-best$[p_{\text{ASR}}(\cdot|x_1^T)]$      $n$ most probable ASR hypotheses
$\mathcal{U}(\text{min}, \text{max})$      uniform random distribution between min and max

## B DIFFERENCES TO PREVIOUS WORK

An overview of the main differences and similarities from our implementation to Gu et al. (2024) is shown in Table B.1. Note that we varied the DLM architecture and size in preliminary experiments and found that the chosen architecture and size worked best in our setting. We also tested the exact same architecture as Gu et al. (2024) (16 encoder layers, 4 decoder layers, model dimension 1280; also positional encoding, feedforward style), but it performed worse in our setting.

After all our experiments (compare Table 1), we still see some gap to Gu et al. (2024) in absolute WER results. Some relevant differences which could explain this gap are the following:

- Gu et al. (2024) uses a character vocabulary, while we mostly use a subword vocabulary. See Appendix G.2 for further discussion on the difference of character and subword vocabularies.

- Another aspect is that Gu et al. (2024) uses a TTS model trained on external data, while our best result is achieved with only LibriSpeech data. But our results using the same TTS model (Appendix E.2.4) only shows small improvements. Also, Gu et al. (2024) reports that the differences between TTS models does not have such a big impact (see Table 7 in Gu et al. (2024), where the relevant WER for DSR decoding is between 3.6% and 3.8%).

- Further, Gu et al. (2024) trains for more epochs. The exact number of epochs is not stated in Gu et al. (2024) and difficult to estimate from the details that they provide, but given

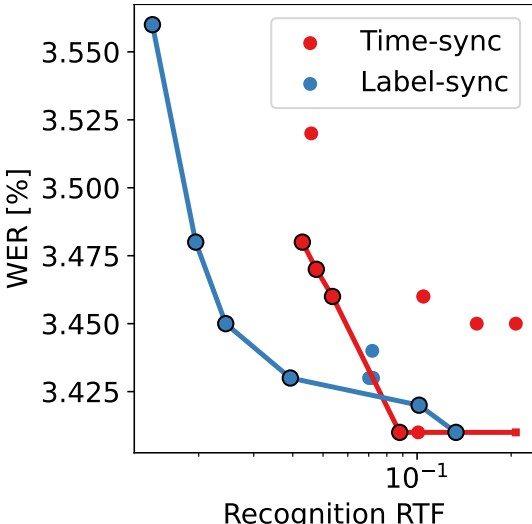

Figure C.1: Speed comparison in terms of real-time factor (RTF) of time-synchronous and label-synchronous search for ASR CTC + LM one-pass decoding.

their specified LR schedule and batch size, we estimate that they train for about 40 epochs, but this might not be accurate.

## C DECODING DETAILS

Decoding is the search procedure to find $\arg\max_{a_1^S}$ given some scoring (for example ASR + LM (Equation (8)), ASR + DLM (DSR, Equation (3)), or ASR + weighted DLM (DLM-sum, Equation (7))).

### C.1 TIME-SYNCHRONOUS SEARCH

Once an ASR model (with CTC) is involved in the score (as in most of our experiments, except DLM alone), we can use *time-synchronous search* (where the outer loop goes over time frames $t = 1, \ldots T$ (Graves, 2012; Prabhavalkar et al., 2023)). We make use of the *maximum-approximation* of CTC (Equation (10)) in this case. During search, recombine hypotheses with the same label sequence but different alignment label sequences (including blank labels) by maximum approximation.

### C.2 LABEL-SYNCHRONOUS SEARCH

In all cases, we can use *label-synchronous search* (where the outer loop goes over labels $s = 1, \ldots, S$ (Prabhavalkar et al., 2023)). Label-synchronous search also works with CTC (Hori et al., 2017). In case of label-synchronous search with CTC, we do *not* use the *maximum approximation* of CTC but instead compute the exact CTC probability (Hori et al., 2017). So, our label-synchronous search results can potentially be slightly better for CTC than our time-synchronous search results, while time-synchronous search is more standard for CTC in the ASR community. However, in our experiments, we don't really see any difference between time-synchronous and label-synchronous search for CTC+LM combination when using the same beam size.

However, we see a difference in compute time, which depends on the LM size, encoder sequence length, vocabulary size. See Figure C.1 for a speed comparison of time-synchronous and label-synchronous search for ASR CTC + LM one-pass decoding. We can see that label-synchronous search is generally faster. This is the case for our large 32 layer Transformer LM. Also, this uses CTC soft collapsing (Appendix C.8) to reduce the encoder sequence length, but consistently for both time-synchronous and label-synchronous search.

## C.3 BEAM SEARCH

In both label-synchronous search or time-synchronous search, for each step, we have a fixed number of active hypotheses, which is the *beam size*, which is just 1 in the first step, and then expands to some fixed beam size. Thus, the search is also called *beam search*. See Prabhavalkar et al. (2023) for more.

## C.4 GREEDY SEARCH

This is beam search with beam size 1. It means that in every single step, we take the $\arg\max$ over the possible next labels. This can be done with time-synchronous search or label-synchronous search.

For time-synchronous search for CTC using the maximum approximation (Equation (10)), the solution becomes trivial: We just take the most probable alignment label (including blank) at each time step $t$, and this gives us the final alignment label sequence. Then we remove blanks and merge repeated labels to get the final label sequence. This is exactly what is usually called *greedy decoding* for CTC.

## C.5 RESCORING

The search can be done first with CTC only (or with the DLM only), to generate an N-best list of hypotheses, which is then rescored with the LM or DLM. This is called *rescoring*.

## C.6 ONE-PASS SEARCH

Alternatively, the LM or DLM can be integrated into the search itself, which is called *one-pass* search. This can be done both with time-synchronous search and label-synchronous search.

Our DLM-sum decoding results are all with one-pass label-synchronous search.

The CTC+LM one-pass results use time-synchronous search, as this is more consistent to what is usually done in the ASR community. In our experiments, we don't really see any difference between time-synchronous and label-synchronous search for CTC+LM combination.

## C.7 OPTIMIZING SCALES

When we combine multiple models in the score, we usually use some scale factors (e.g. $\lambda_{\text{LM}}, \lambda_{\text{prior}}$ in Equation (8)). We usually generate an N-best list using only the CTC model (or also with the DLM, and then combine it with the CTC hypotheses), then we score those hypotheses which each model individually (rescoring), and then tune the scale factors on the N-best list on the validation set (LibriSpeech dev-other). After that, we use the tuned scale factors for the final evaluation on the other test sets.

## C.8 CTC SOFT COLLAPSING

Both time-synchronous search and label-synchronous search for CTC are linearly bound to the encoder sequence length $T$. We use a simple heuristic to reduce $T$: We merge consecutive frames of the probabilities $p_{\text{ASR},t}(y \mid x_1^T)$ where the argmax is the same and the probability is above some threshold (0.8 or 0.9 in our experiments). We merge by taking the maximum probability over the merged frames for each label $y$ and then renormalizing to a probability distribution.

This is similar to the method described in Gaido et al. (2021), but we have the additional threshold to avoid merging frames where the model is uncertain.

## C.9 DLM-SUM DECODING DETAILS

The DLM-sum decoding (Section 2.1) involves two searches: one for the hypotheses from the ASR model, and one for the final combined score (Equation (7)).

First, we apply CTC soft collapsing to the ASR model probabilities.

The first search on the ASR model is done with time-synchronous beam search to get the N-best list of ASR hypotheses. There can be potentially different alignment label sequences for the same label sequence (after merging repeated labels and removing blanks). We also tested label-synchronous search for this step, or some sampling variants, to get a more diverse set of hypotheses, but we did not see any consistent improvements.

The second search for the final combined score is done with one-pass label-synchronous beam search.

## D EXPERIMENTAL DESIGN DETAILS

Here we provide more details about our experimental design (Section 3).

### D.1 MODELS

#### D.1.1 ASR MODEL DETAILS

**Model Details.** We use a Conformer encoder (Gulati et al., 2020). The feedforward layers use ReLu squared activation function. We use a convolutional frontend consisting of three convolutional layers as with kernel sizes 3, 3, 3, strides in the time dimension 1, 3, 2 and 1, 1, 1 in the frequency dimension, and max pooling of 2 for the frequency dimension only in the first layer. For the char and spm128 models we use stride 1, 3, 1 to get less downsampling. Out dims are 32, 64, 64. We use log mel filterbank features with 80 channels, 25ms window size and 10ms step size and batch norm over the feature dimension.

This is a standard CTC ASR model (Graves et al., 2006): The model output is a linear layer to the vocabulary size plus one for the CTC blank symbol, followed by softmax, resulting in the probability distribution $p_{\mathrm{ASR},t}(y_t \mid x_1^T)$ over the vocabulary including blank at each output time step $t$. The sequence probability is the given as

$$p_{\mathrm{ASR}}(a_1^S \mid x_1^T) = \sum_{y_1^{T'} \,:\, a_1^S} \prod_{t=1}^{T'} p_{\mathrm{ASR},t}(y_t \mid x_1^T),  \tag{9}$$

where the sum is over all sequences $y_1^{T'}$ that map to $a_1^S$ after removing blanks and merging repeated symbols. Due to downsampling in the encoder (by striding in the convolutional frontend), we have $T' = \left\lceil \frac{T}{6} \right\rceil$.

In some cases, we use the maximum approximation:

$$p_{\mathrm{ASR'}}(a_1^S \mid x_1^T) = \max_{y_1^{T'} \,:\, a_1^S} \prod_{t=1}^{T'} p_{\mathrm{ASR},t}(y_t \mid x_1^T).  \tag{10}$$

**Training Details.** The TTS audio we use for ASR training is generated using our Glow-TTS model (Section 3.2) with constant noise scale 0.7 and length scale 1.0 for all of the Librispeech LM text (800M words). The noise scale parameter of the TTS data is tuned to optimize performance of ASR models trained on this data, as measured by WER on LibriSpeech dev-other. This results in about 75k hours of synthetic audio. Due to the large amount of TTS data, we disperse it over multiple epochs, such that each epoch contains a subset of $\frac{1}{75}$ of the TTS data (about 1000h) and the entire LibriSpeech ASR 960h training data. We train for 100 epochs in total, which means that we see about 96k hours of the LibriSpeech ASR 960h data and about 100k hours of TTS data in total, i.e. about 1:1 ratio of real to synthetic data, with about 200k hours of audio training data in total.

Dropout of 0.1 is applied to prevent overfitting. The models are trained with the AdamW optimizer and 1e-2 weight decay with batch size 16.6min (spm10k) or 8.3min (char, spm128) for 100 epochs, and global gradient clipping of 5.0. To improve memory efficiency, we use bfloat16 for training. We use one cycle learning rate schedule with linear increase from 1e-5 to 1e-3 lr in the first 45% of training, then linear decrease to 1e-5 at 90% of training then linear decrease to 1e-6 to the end of training. Audios that are longer than 19.5s in length are removed from the dataset. Input audio is speed perturbed with factors 0.7, 0.8, . . . , 1.1, and we use SpecAugment (Park et al., 2019)

Table D.1: Word error rates of our Conformer baseline models on LibriSpeech dev and test sets. Additionally we report the WER on TTS audios generated with our Glow-TTS system (last column) with noise scale $\sim \mathcal{U}(0.3, 0.9)$ and length scale $\sim \mathcal{U}(0.7, 1.1)$, both uniform random sampled for each sentence. The ASR model 'spm10k' is only trained on LibriSpeech ASR 960h, while all other models had additional TTS data in their training data (see text for details). Number of Parameters includes auxiliary decoder and CTC losses in the encoder.

| ASR Baselines | Number of Parameters | WER [%] | | | | |
|---|---|---|---|---|---|---|
| | | dev-clean | dev-other | test-clean | test-other | tts |
| spm10k | 471M | 2.29 | 5.02 | 2.42 | 5.33 | 8.73 |
| spm10k (TTS) | 471M | 1.75 | 4.13 | 2.03 | 4.44 | 2.17 |
| char (TTS) | 435M | 1.83 | 4.56 | 1.98 | 4.78 | 2.55 |
| spm128 (TTS) | 435M | 1.79 | 4.41 | 1.94 | 4.55 | 2.42 |

Table D.2: LM performance (perplexity (PPL) on spm10k level) on LibriSpeech dev-other for different LM architectures and number of training epochs.

| Model | Num Epochs | PPL |
|---|---|---|
| n8-d1024 | 5 | 37.2 |
| n32-d1024 | 5 | 33.9 |
| n32-d1024 | 10 | 34.4 |
| n32-d1280 | 5 | 32.9 |

as additional data augmentation. For the SentencePiece model we randomly stop tokenizing mid-word with 1% probability to stochastically generate non-deterministic subword splits. An auxiliary decoder with 6 layers and model dimension 512 is used during training as per Hentschel et al. (2024). Additionally an auxiliary CTC loss is applied at layer 4 and 8 of the encoder.

Most hyperparameters of our ASR are hand-tuned on dev-other, resulting in a slight overfit to the validation sets.

**Performance.** The performance of our ASR models and their parameter count is shown in Table D.1.

### D.1.2 LM DETAILS

The models are trained with AdamW optimizer (Loshchilov & Hutter, 2019) and 1e-2 weight decay with batch size 20k, 15k tokens for 5 epochs on LibriSpeech LM corpus, and global gradient clipping of 5.0. We use cross entropy loss, which is averaged over all sequences and tokens in a batch. The learning rate schedule is one cycle with linear increase from 1e-5 to 1e-3 lr in the first 45% of training, then linear decrease to 1e-5 at 90% of training, then linear decrease to 1e-6 to the end of training. See Table D.2 for perplexities (PPL) of our LMs on LibriSpeech dev-other.

The combined ASR + LM results are computed with shallow fusion and internal LM subtraction, and scales ($\lambda_{\text{LM}}$, $\lambda_{\text{prior}}$) are tuned on LibriSpeech dev-other. See Tables D.3, E.1 and E.2 for results.

### D.1.3 PRIOR

The prior probability $p_{\text{prior}}(a_1^S)$ is estimated from the ASR model. We take the average of $p_t(y \mid x_1^T)$ over all frames of the training dataset, i.e.

$$p_{\text{framewise-prior}}(y) = \frac{1}{\sum_{T', x_1^T \in \mathcal{D}} T'} \sum_{T', x_1^T \in \mathcal{D}} \sum_{t=1}^{T'} p_{\text{ASR},t}(y \mid x_1^T) \qquad (11)$$

following Manohar et al. (2015). This method is sometimes referred to as the *softmax average*. This is a context-independent framewise prior over the vocabulary including the CTC blank symbol.

Table D.3: Word error rates of our spm10k (TTS) ASR baseline model with external language model decoding on LibriSpeech dev and test sets. Rescoring refers to the two-pass approach where an ASR n-best list is rescored. The one-pass approach uses joint decoding with shallow fusion. `n8-d1024` refers to a Transformer decoder-only LM with 8 decoder blocks and model dimension 1024, other LMs are named accordingly. The LMs are trained for 5 epochs here. LM `n8-d1024` uses $\lambda_{\text{LM}} \approx 0.37$, $\lambda_{\text{prior}} \approx 0.27$, and `n32-d1024` uses $\lambda_{\text{LM}} \approx 0.42$, $\lambda_{\text{prior}} \approx 0.28$.

| LM | Number of Parameters | Decoding | WER [%] | | | |
|----|----------------------|----------|-----------|-----------|------------|------------|
| | | | dev-clean | dev-other | test-clean | test-other |
| None | 0 | - | 1.75 | 4.13 | 2.03 | 4.44 |
| n8-d1024 LM | 113M | rescoring | 1.63 | 3.59 | 1.83 | 3.90 |
| | | one-pass | 1.62 | 3.53 | 1.81 | 3.82 |
| n32-d1024 LM | 422M | rescoring | 1.59 | 3.57 | 1.80 | 3.84 |
| | | one-pass | 1.56 | 3.41 | 1.73 | 3.70 |

We define a context-independent labelwise prior by removing the blank symbol and renormalizing:

$$p_{\text{labelwise-prior}}(a) = \frac{p_{\text{framewise-prior}}(a)}{\sum_{a' \in \mathcal{V}} p_{\text{framewise-prior}}(a')} \tag{12}$$

for $a \in \mathcal{V}$, where $\mathcal{V}$ is the vocabulary without the CTC blank symbol.

The prior probability of a (non-blank) label sequence is then

$$p_{\text{prior}}(a_1^S) = \prod_{s=1}^{S} p_{\text{labelwise-prior}}(a_s). \tag{13}$$

### D.1.4 TTS DETAILS

**Glow-TTS.** Our Glow-TTS system is a normalizing flow-based generative model utilizing a bi-directional mapping between audio features and latent variables. The latent variable for each frame is sampled from a Gaussian distribution, parameterized by a text encoder model. For each input token, the text encoder predicts mean and variance vectors. Then the number of output frames for each input token is determined by a duration model, and the distribution parameters are repeated accordingly for each position. After sampling the latent variable for each position, the spectrograms are computed via the inverse of the normalizing flow function.

The Glow-TTS model has two straightforward ways to adjust the output of the TTS system: length scale and temperature.

The length scale is a multiplicative factor applied to the predicted durations of each input token. We observe that the TTS produces well recognizable audio for length scales between 0.6 and 2.0, with the WER increasing considerably outside this range (cf. Figure D.1). The non-TTS trained ASR model seems to be more robust to slower speech (higher length scale) than the TTS-trained ASR model, which may be due to the TTS training data having a smaller proportion of slow speech compared to the LibriSpeech ASR training data[7].

Temperature (from here on called noise scale) is a multiplicative factor applied to the standard deviation of the predicted latent distribution. Consider the sampling procedure for the latent variable $z$ in the Glow-TTS model:

$$z \sim \mathcal{N}(\mu, (\sigma\tau)^2) \tag{14}$$
$$z = \mu + \sigma\tau \cdot \epsilon \text{ with } \varepsilon \sim \mathcal{N}(0, 1) \tag{15}$$

where $z$ is the sampled latent variable, $\mu$ and $\sigma$ are the mean and standard deviation vectors of the latent distribution predicted by the text encoder model, $\tau \in \mathbb{R}^+$ is the noise scale hyperparameter (temperature), and $\epsilon$ is a vector of standard normal random variables. Increasing the noise scale $\tau$ increases the variance of the latent distribution, which leads to more diverse outputs. Noise scales

---

[7]TTS audio with Length scale = 1.0 and Noise scale = 0.7 was used for ASR model training.

Table D.4: WER of training data, ablation over TTS Length Scale uniformly distributed. TTS Noise scale $\tau \sim \mathcal{U}(0.3, 0.9)$ is used.

| TTS Length Scale over a uniform distribution | DLM Training Data WER [%] | |
|---|---|---|
| | spm10k | spm10k (TTS) |
| | Glow-TTS | Glow-TTS |
| $\mathcal{U}(1.0,\ 1.0)$ | 7.46 | 1.88 |
| $\mathcal{U}(0.7,\ 1.1)$ | 8.73 | 2.17 |
| $\mathcal{U}(0.6,\ 2.0)$ | 8.04 | 2.16 |
| $\mathcal{U}(0.5,\ 2.5)$ | 9.31 | 3.18 |
| $\mathcal{U}(0.4,\ 3.0)$ | 12.06 | 6.90 |

Table D.5: WER of training data, ablation over TTS Length Scale. TTS Noise scale $\tau \sim \mathcal{U}(0.3, 0.9)$ is used.

| TTS Length Scale | DLM Training Data WER [%] | |
|---|---|---|
| | spm10k | spm10k (TTS) |
| | Glow-TTS | Glow-TTS |
| 0.2 | 96.09 | 96.47 |
| 0.4 | 70.33 | 54.31 |
| 0.5 | 36.28 | 14.64 |
| 0.6 | 19.34 | 5.17 |
| 0.8 | 9.83 | 2.39 |
| 1.0 | 7.46 | 1.88 |
| 1.2 | 6.72 | 1.74 |
| 1.4 | 6.58 | 1.76 |
| 2.0 | 7.85 | 2.72 |
| 2.5 | 11.43 | 8.85 |
| 3.0 | 18.41 | 26.85 |
| 3.5 | 28.63 | 51.29 |
| 4.0 | 40.12 | 72.52 |

up to 0.8 seem to produce well recognizable audio for the TTS-trained ASR model, while the non-TTS trained ASR model has already doubled its WER at that point (cf. Figure D.2). Again, the TTS-trained ASR model performs better than the non-TTS trained model on the noise parameters it has seen during training.

We generate TTS audio with different length scales and measure the WER of the resulting ASR hypotheses, shown in Figure D.1. WER for different levels of noise scale is shown in Figure D.2.

**YourTTS.** We download the YourTTS model from the public Coqui-ai Github repository[8] and use it in its default configuration (unless otherwise stated). The YourTTS model has been trained on the LibriTTS (Zen et al., 2019) and CML-TTS (Oliveira et al., 2023) datasets.

Similar to the Glow-TTS system, YourTTS also has length and noise scale parameters that can be adjusted during inference. We keep length scale $\in [1.0, 1.5]$ and noise scale $\tau = 0.3$ unless otherwise stated. The default values for these parameters are 1.5 and 0.3 respectively, so we believe that our results are comparable to that of Gu et al. (2024).

**Generated Data WERs** WER results for hypotheses generated with Glow-TTS and YourTTS on different ASR models are shown in Table D.8. WER for spm10k is about the same as for the Glow-TTS system, and the relative change from non-TTS to the TTS-trained ASR models is similar to that of the LibriSpeech validation and test sets (cf. Table D.1). This indicates that the YourTTS audios are meaningfully different from the Glow-TTS audios, and should introduce additional diversity into the training data when both systems combined.

---

[8] https://github.com/coqui-ai/TTS

Table D.6: TTS Noise Scale Uniform.

| TTS Noise Scale Uniform | DLM Training Data WER [%] | |
| | spm10k | spm10k (TTS) |
| | Glow-TTS | Glow-TTS |
|---|---|---|
| $\mathcal{U}(0.3,\ 0.4)$ | 5.12 | 1.55 |
| $\mathcal{U}(0.3,\ 0.5)$ | 5.31 | 1.55 |
| $\mathcal{U}(0.3,\ 0.6)$ | 5.59 | 1.58 |
| $\mathcal{U}(0.3,\ 0.7)$ | 6.03 | 1.63 |
| $\mathcal{U}(0.3,\ 0.8)$ | 6.62 | 1.72 |
| $\mathcal{U}(0.3,\ 0.9)$ | 7.46 | 1.88 |
| $\mathcal{U}(0.3,\ 1.0)$ | 8.61 | 2.15 |
| $\mathcal{U}(0.3,\ 1.1)$ | 10.18 | 2.64 |
| $\mathcal{U}(0.3,\ 1.2)$ | 12.19 | 4.11 |
| $\mathcal{U}(0.3,\ 1.3)$ | 14.69 | 6.66 |
| $\mathcal{U}(0.3,\ 1.4)$ | 17.70 | 9.89 |
| $\mathcal{U}(0.3,\ 1.5)$ | 20.98 | 13.57 |

Table D.7: Ablation over TTS Noise Scale. Constant length scale of $1.0$ is used.

| TTS Noise Scale $\tau$ | DLM Training Data WER [%] | |
| | spm10k | spm10k (TTS) |
| | Glow-TTS | Glow-TTS |
|---|---|---|
| 0.0 | 4.89 | 1.64 |
| 0.2 | 4.92 | 1.59 |
| 0.4 | 5.26 | 1.55 |
| 0.6 | 6.64 | 1.69 |
| 0.8 | 10.12 | 2.33 |
| 1.0 | 17.88 | 4.57 |
| 1.2 | 32.43 | 23.27 |
| 1.4 | 52.36 | 47.74 |

Table D.8: Recognition performance of ASR baselines on Glow-TTS and YourTTS audio data. Glow-TTS audio data is generated with noise scale $\tau \sim \mathcal{U}(0.7, 1.1)$ and length scale $\sim \mathcal{U}(0.7, 1.1)$, YourTTS audios use noise scale $\tau = 0.3$, length scale $\sim \mathcal{U}(1.0, 1.5)$, all uniformly sampled for each sentence.

| ASR Baselines | WER [%] | |
| | Glow-TTS | YourTTS |
|---|---|---|
| spm10k | 8.73 | 8.47 |
| spm10k TTS | 2.17 | 6.11 |
| char TTS | 2.55 | 6.29 |
| spm128 TTS | 2.42 | 6.10 |

### D.1.5 DLM DETAILS

We follow the design principles of Llama (Touvron et al., 2023) like with our LMs.

The input hypotheses for the encoder are postfixed with a special end-of-sequence token from the vocabulary.

We use label smoothing of 0.1 for the cross entropy loss. We train most DLMs for 5 epochs, where we will use the one-cycle learning rate schedule as with our LMs, but with the learning rate halved at every step. So the learning rate increases linearly from 5e-6 to 5e-4 in the first 45% of training, then decreases linearly to 5e-6 at 90% of training, then decreases linearly to 5e-7 at the end of training. For the experiments with our best results we train for 10 epochs, where we use the exact same learning rate schedule without halving the learning rate at every step.

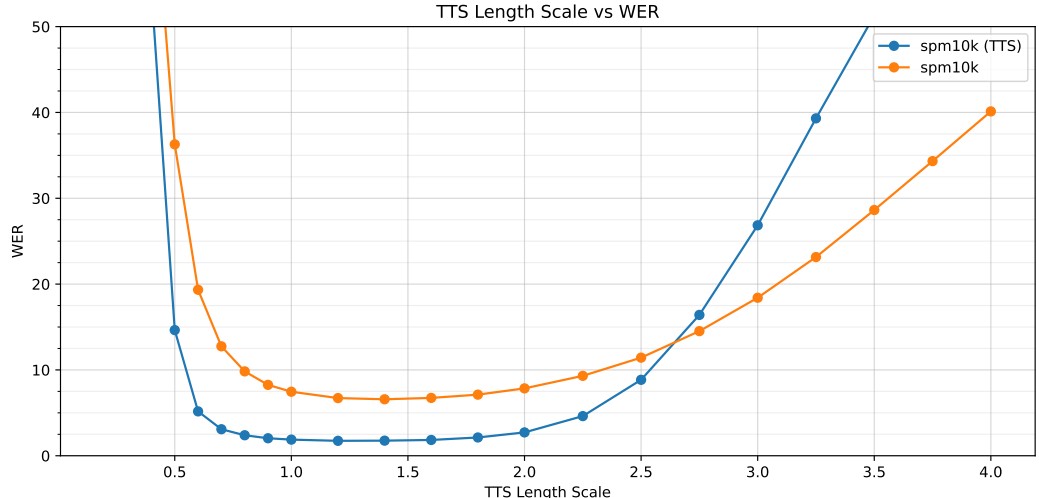

Figure D.1: Length scale of TTS vs. WER of the ASR hypotheses. Noise scale $\tau \sim \mathcal{U}(0.3, 0.9)$. See Table D.5 for raw data.

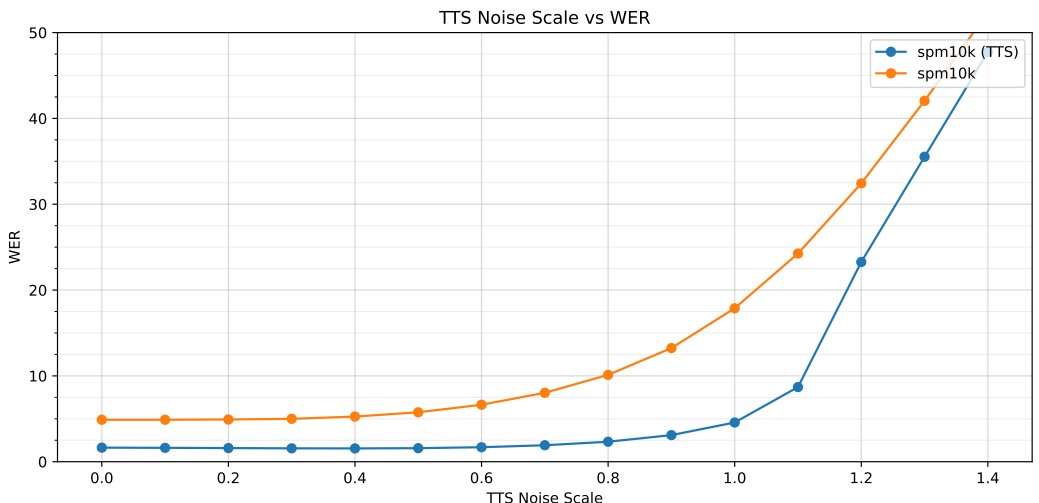

Figure D.2: Noise scale of TTS vs WER of the ASR hypotheses. Length scale $=$ 1.0 was used. See Table D.7 for raw data.

## D.2 DATA AUGMENTATION STRATEGIES

The goal is to find training data that leads to the best DLM performance.

Unless otherwise stated, we apply each technique to the LibriSpeech ASR validation sets with real audio, and to Glow-TTS audio from a random subset of 53890 sentences with total 1068186 words, about 0.13% of the LM corpus. A 0.01% change in WER on this set corresponds to about 106 word errors. Given the great cost of generating the full training data, this is a reasonable compromise to get an estimate of the impact of each technique. We use the spm10k Conformer ASR models, with and without TTS training data, to measure the WER of the generated hypotheses. Since there is no indication that the data augmentation methods should behave differently across different vocabularies, we chose not to conduct additional experiments with the *spm128* and *char* vocabularies.

Table D.9: WER on early ASR checkpoints. The ASR model here is 'spm10k (TTS)'.

| ASR checkpoint (out of 100) | DLM Training Data WER [%] | | |
|---|---|---|---|
| | dev-clean | dev-other | Glow-TTS |
| Epoch 10 | 5.24 | 12.39 | 7.24 |
| Epoch 20 | 3.76 | 9.14 | 5.53 |
| Epoch 40 | 3.19 | 8.37 | 4.89 |
| Epoch 80 | 2.08 | 5.00 | 2.76 |
| Epoch 100 | 1.75 | 4.13 | 2.17 |

Table D.10: Training data WER for different SpecAugment settings.

| SpecAugment | DLM Training Data WER [%] | | | | | |
|---|---|---|---|---|---|---|
| | spm10k | | | spm10k (TTS) | | |
| | dev-clean | dev-other | Glow-TTS | dev-clean | dev-other | Glow-TTS |
| Off | 2.29 | 5.02 | 8.73 | 1.75 | 4.13 | 2.17 |
| On (only frequency masking) | 2.46 | 5.95 | 9.79 | 1.90 | 4.92 | 2.54 |
| On (only time masking) | 3.94 | 8.38 | 16.79 | 3.27 | 7.01 | 6.54 |
| On (time + frequency) | 4.88 | 10.21 | 19.45 | 3.82 | 8.99 | 8.02 |

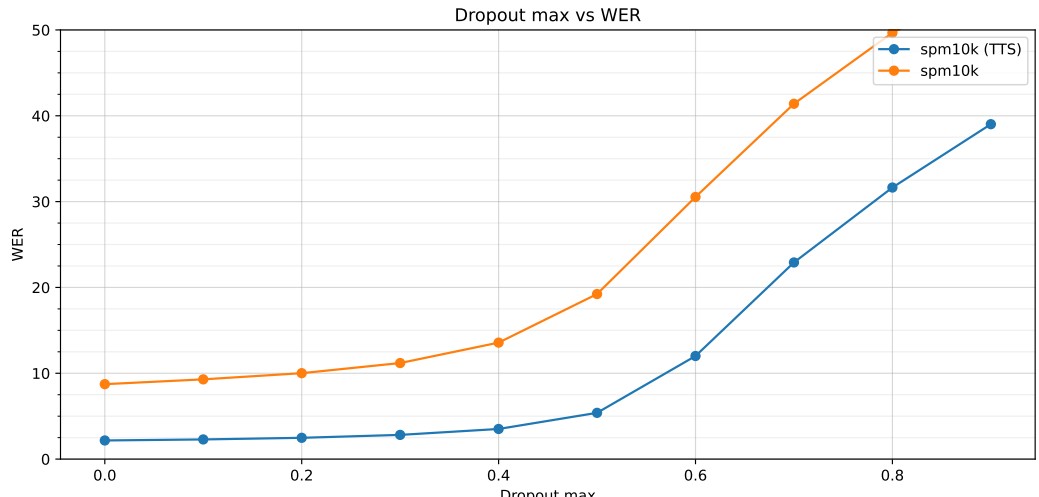

Figure D.3: Dropout is sampled uniformly from $\mathcal{U}(0.0, p_{\max})$ for each sequence. $p_{\max}$ vs. WER of the ASR hypotheses. Data shown is TTS audio from Glow-TTS.

### D.2.1 EARLY ASR CHECKPOINTS

Generated training data statistics for different checkpoints are shown in Table D.9.

### D.2.2 SPECAUGMENT

We use a variant of SpecAugment (Park et al., 2019): we do frequency masking with 2 to 5 masks of max size 16, and time masking with 2 to $\frac{\text{len}}{25}$ masks with max size 20 where len is the length of the time dimension. We do not apply time warping. We test whether only time masking, only frequency masking, or both combined lead to the best DLM performance.

Generated training data statistics using SpecAugment are shown in Table D.10.

Table D.11: Training data WER for different dropout percentages.

| $(p_{\min}, p_{\max})$ | DLM Training Data WER [%] | | | | | |
|---|---|---|---|---|---|---|
| | spm10k | | | spm10k (TTS) | | |
| | dev-clean | dev-other | Glow-TTS | dev-clean | dev-other | Glow-TTS |
| (0.0, 0.0) | 2.29 | 5.02 | 8.73 | 1.75 | 4.13 | 2.17 |
| (0.0, 0.1) | 2.36 | 5.41 | 9.29 | 1.83 | 4.36 | 2.29 |
| (0.0, 0.2) | 2.62 | 5.97 | 10.01 | 1.99 | 4.67 | 2.48 |
| (0.0, 0.3) | 3.00 | 6.77 | 11.19 | 2.35 | 5.44 | 2.82 |
| (0.0, 0.4) | 3.79 | 8.52 | 13.57 | 2.99 | 7.02 | 3.51 |
| (0.0, 0.5) | 6.15 | 12.82 | 19.23 | 4.89 | 11.22 | 5.39 |
| (0.0, 0.6) | 14.67 | 22.84 | 30.54 | 13.96 | 22.69 | 12.01 |
| (0.0, 0.7) | 28.14 | 34.00 | 41.40 | 28.16 | 35.56 | 22.91 |
| (0.0, 0.8) | 37.91 | 42.16 | 49.67 | 37.71 | 44.51 | 31.64 |
| (0.0, 0.9) | 43.93 | 47.96 | 55.64 | 43.96 | 50.32 | 39.02 |
| (0.1, 0.5) | 7.36 | 14.68 | 21.60 | 5.88 | 13.00 | 6.29 |
| (0.2, 0.5) | 9.06 | 17.79 | 24.98 | 7.10 | 15.65 | 7.58 |
| (0.5, 0.5) | 34.17 | 51.87 | 57.28 | 26.36 | 46.43 | 24.64 |

Table D.12: Training data WER for different token substitution percentages. A percentage is uniformly sampled for each sentence, and that percentage of tokens in the sentence are randomly replaced with other tokens.

| Token substitution | DLM Training Data WER [%] | | | | | |
|---|---|---|---|---|---|---|
| | spm10k | | | spm10k (TTS) | | |
| | dev-clean | dev-other | Glow-TTS | dev-clean | dev-other | Glow-TTS |
| 0% to 0% | 2.29 | 5.02 | 8.73 | 1.75 | 4.13 | 2.17 |
| 5% to 5% | 8.64 | 11.22 | 14.59 | 8.19 | 10.50 | 8.58 |
| 10% to 10% | 14.98 | 16.95 | 20.30 | 14.33 | 16.23 | 14.96 |
| 20% to 20% | 26.76 | 28.36 | 31.46 | 26.91 | 27.73 | 27.11 |
| 30% to 30% | 38.78 | 39.86 | 42.21 | 38.28 | 39.26 | 38.91 |
| 0% to 30% | 20.95 | 22.45 | 25.91 | 20.13 | 21.98 | 20.96 |
| 0% to 40% | 26.59 | 28.50 | 31.16 | 26.34 | 27.93 | 26.88 |
| 0% to 90% | 54.20 | 55.33 | 56.69 | 53.79 | 54.13 | 54.47 |

### D.2.3 DROPOUT

We use dropout (Srivastava et al., 2014) at inference time to generate diverse hypotheses. See Figure D.3 for dropout rate vs. WER. See Table D.11 for generated training data statistics using different dropout rates.

### D.2.4 TOKEN SUBSTITUTION

We expect that the WER of the hypotheses will approximately increase by $\frac{\min + \max}{2}$ over the whole corpus. The actual WER increase is typically higher than this, due to two reasons: First, the substitution probability is per token, not per word. Because a word can consist of more than one token, the probability of a word having at least one substitution is higher than the probability of a single token having a substitution. Secondly, a substitution may result in a word being split into two smaller words because of a mid-word token replaced with one that begins with a space. Such a substitution produces two errors for a single token substitution (an insertion error and a substitution error).

Training data WER for different Token substitution rates is shown in Table D.12.

### D.2.5 MIXUP

Mixup (Zhang et al., 2017) is a data augmentation technique that enforces the model to learn a linear relationship between any two training examples:

$$\tilde{x} = \lambda x_i + (1 - \lambda)x_j, \qquad \text{with } x_i, x_j \text{ input vectors} \qquad (16)$$
$$\tilde{y} = \lambda y_i + (1 - \lambda)y_j, \qquad \text{with } y_i, y_j \text{ output vectors} \qquad (17)$$
$$\text{or: } \mathcal{L}_{\text{mixspeech}} = \lambda \mathcal{L}(\tilde{x}, y_i) + (1 - \lambda)\mathcal{L}(\tilde{x}, y_j), \qquad \text{where } \mathcal{L} \text{ is the original loss} \qquad (18)$$
$$\text{with } \lambda \in [0, 1] \qquad (19)$$

It has been shown that this leads to improvements when applied to ASR model training (Meng et al., 2021a).

Because we only want to generate DLM training data, and are not interested training ASR models, we only need equation equation 16, where we mix the spectograms of multiple audio sequences together. We extend the mixup equation to more than two sequences as follows:

$$\tilde{x} = (1 - \lambda)x_i + \lambda x_{\text{sum}} \qquad (20)$$

$$\text{with } x_{\text{sum}} = \sum_j^n \alpha_j x_j \text{ and } \sum_j^n \alpha_j = 1 \qquad (21)$$

$$\text{where } x_j \text{ are randomly chosen input features of other sequences}$$
$$\text{and } \alpha_j \text{ are random weights unique to an } x_i$$

Rather than linearization, this approach can be best understood as adding some background noise to an audio sequence. For consistency to other training data ablations, we have inverted the usage of $\lambda$ to be the amount of noise added instead of the amount of original signal kept. This way $\lambda = 0$ means no noise, and $\lambda = 1$ means 100% noise. At inference time, audio sequence spectrograms are continuously appended to a buffer, and for mixup we randomly pick $n$ offsets from which we copy the input spectrograms. This sometimes results in multiple adjacent spectrograms from the buffer being mixed into the current sequence. We randomly pick $n \in \{1, 2\}$ for every sequence and set $\lambda$ to a constant value in Figure D.4 and Table D.13.

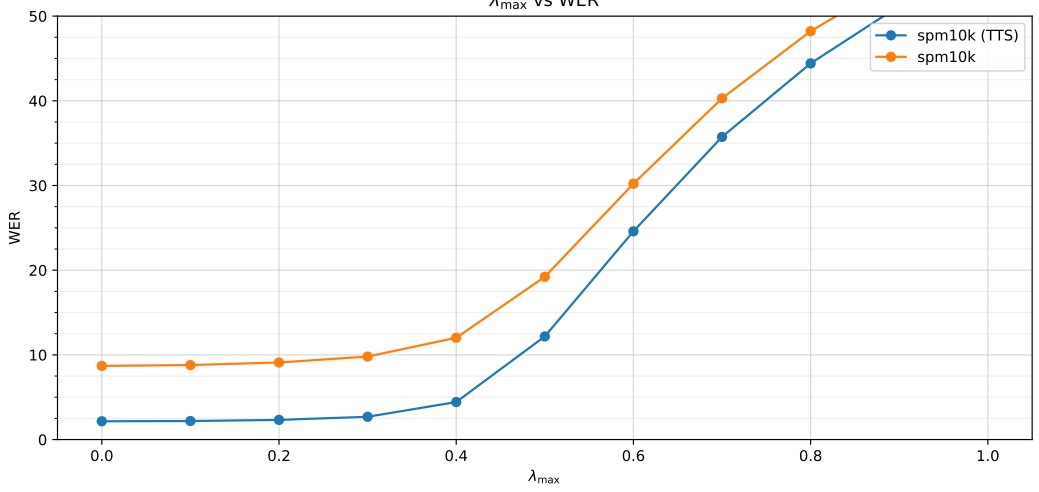

Figure D.4: Mixup $\lambda_{\text{max}}$ vs. WER of the ASR hypotheses.

As expected, the WER increases sharply near $\lambda \approx 0.5$, where the noise begins to dominate the original signal. Before this point, the WER remains remarkably stable. For our training experiments we pick $n \in \{1, 2\}$ randomly, but sample $\lambda \sim \mathcal{U}(0, \lambda_{\text{max}})$ for every sequence with some constant $\lambda_{\text{max}}$.

Table D.13: Training data WER for different Mixup $\lambda$ values. The $\lambda$ value decides how much noise from other audios is added to each sample.

| Mixup $\lambda$ | DLM Training Data WER [%] | | | | | |
| | spm10k | | | spm10k (TTS) | | |
| | dev-clean | dev-other | Glow-TTS | dev-clean | dev-other | Glow-TTS |
|---|---|---|---|---|---|---|
| 0.0 | 2.29 | 5.02 | 8.70 | 1.75 | 4.13 | 2.16 |
| 0.1 | 2.33 | 5.20 | 8.97 | 1.76 | 4.23 | 2.26 |
| 0.2 | 2.51 | 5.96 | 9.94 | 1.89 | 4.88 | 2.71 |
| 0.3 | 3.22 | 9.55 | 13.17 | 2.62 | 7.97 | 4.69 |
| 0.4 | 11.01 | 26.00 | 27.66 | 9.37 | 23.34 | 19.39 |
| 0.5 | 57.24 | 67.04 | 68.51 | 54.93 | 65.13 | 67.99 |
| 0.6 | 95.23 | 96.00 | 95.63 | 95.05 | 96.39 | 98.12 |
| 0.7 | 102.87 | 102.51 | 101.94 | 103.23 | 103.35 | 103.95 |

Table D.14: Training data WER for different Mixup $\lambda_{max}$ values.

| Mixup $\lambda_{max}$ | DLM Training Data WER [%] | | | | | |
| | spm10k | | | spm10k (TTS) | | |
| | dev-clean | dev-other | Glow-TTS | dev-clean | dev-other | Glow-TTS |
|---|---|---|---|---|---|---|
| 0.0 | 2.29 | 5.02 | 8.70 | 1.75 | 4.13 | 2.16 |
| 0.1 | 2.31 | 5.05 | 8.80 | 1.76 | 4.11 | 2.19 |
| 0.2 | 2.35 | 5.22 | 9.10 | 1.79 | 4.34 | 2.32 |
| 0.3 | 2.51 | 5.94 | 9.80 | 1.94 | 4.89 | 2.69 |
| 0.4 | 3.26 | 8.43 | 12.02 | 2.55 | 6.95 | 4.43 |
| 0.5 | 9.25 | 15.82 | 19.22 | 7.49 | 13.86 | 12.18 |
| 0.6 | 21.02 | 26.79 | 30.21 | 18.30 | 25.04 | 24.59 |
| 0.7 | 32.24 | 37.82 | 40.29 | 29.32 | 34.84 | 35.74 |
| 0.8 | 41.49 | 45.97 | 48.22 | 37.81 | 43.03 | 44.42 |
| 0.9 | 48.33 | 52.33 | 54.39 | 44.70 | 49.60 | 51.20 |
| 1.0 | 54.18 | 57.71 | 59.41 | 50.38 | 55.22 | 56.72 |

### D.2.6 SAMPLING FROM ASR MODEL

A good approximation for the top hypothesis of a CTC-based ASR model is to take the most probable token at each frame, and then collapse the resulting sequence by removing blanks and repeated tokens. This approach is called greedy decoding and it is what we use for our training data generation. At recognition time however, we use an n-best list of ASR hypotheses in DLM-sum Decoding to improve performance (see Section 2.1). To reflect this, we can generate training data that contains more suboptimal hypotheses to prepare the DLM for this scenario. To do this, we first collapse the frames as we would do for greedy decoding, i.e. remove blanks and repeated tokens according to the most probable label at each frame. But instead of taking the most probable label, we collect the top $k$ token labels for each collapsed frame, normalize their probabilities back to 1, and then sample from this distribution at every label position. A similar sampling approach for language modeling is described in Fan et al. (2018). This approach is negligible in terms of additional compute resources[9] as the operation is parallelizable over all frames and can be done in a single pass. The impact on the WER of the hypotheses is shown in Table D.15.

We see diminishing changes after $k \geq 8$, indicating that most of the probability mass is concentrated on the top few tokens.

Through our approach, we disregard a significant amount of the frames of the input sequence which may contain additional, possibly important, information. One could instead take the average or the maximum over all collapsed frames and store this information in the resulting combined frame, but we leave this for future work. Nucleus sampling is also an interesting alternative to explore.

---

[9]Baseline: 61.6min, Top-k 16: 62.9min

Table D.15: Training data WER for different Top-k sampling $k$ values. $k = 1$ corresponds to the baseline (no sampling).

| $k$ | DLM Training Data WER [%] | | | | | |
| | spm10k | | | spm10k (TTS) | | |
| | dev-clean | dev-other | Glow-TTS | dev-clean | dev-other | Glow-TTS |
|---|---|---|---|---|---|---|
| 1 | 2.29 | 5.02 | 8.77 | 1.75 | 4.13 | 2.16 |
| 2 | 4.51 | 7.34 | 11.29 | 4.01 | 6.37 | 4.45 |
| 4 | 6.38 | 9.28 | 13.19 | 5.69 | 8.08 | 6.21 |
| 8 | 6.68 | 9.88 | 13.59 | 5.99 | 8.56 | 6.53 |
| 16 | 6.73 | 9.51 | 13.62 | 6.08 | 8.68 | 6.60 |
| 32 | 6.61 | 9.65 | 13.68 | 6.04 | 8.64 | 6.62 |

Table D.16: Combined data augmentation configurations. All configs use TTS length scale $\sim \mathcal{U}(0.7, 1.1)$. TTS noise scale is sampled from $\mathcal{U}(0.3, \tau_{max})$ and mixup $\lambda \sim \mathcal{U}(0.0, \lambda_{max})$ uniformly for every sequence. Configs are sorted according to WER on hypotheses from TTS audio, see Table D.17 for hypotheses WERs.

| Name | TTS Noise $\tau_{max}$ | SpecAugment | Dropout | Token Substitution | Mixup $\lambda_{max}$ |
|---|---|---|---|---|---|
| baseline | | - | (0.0, 0.0) | 0% | 0.0 |
| verylow | | | (0.0, 0.1) | 0% to 10% | 0.2 |
| stdPerturb | | | (0.1, 0.5) | 10% | 0.0 |
| low | 0.9 | Time | (0.0, 0.2) | | |
| very low (ASR ep. 40) | | | (0.0, 0.1) | 0% to 10% | 0.2 |
| low$^+$ | | Time+Freq | (0.0, 0.2) | 10% | |
| lowmedium | 1.2 | | (0.0, 0.0) | 0% to 20% | 0.4 |
| high (no ASR augment) | 1.5 | - | | 20% | 0.0 |
| medium | 1.2 | Time+Freq | (0.0, 0.5) | 10% | 0.4 |
| high | 1.5 | | (0.1, 0.5) | 20% | 0.6 |

### D.2.7 OTHER DATA AUGMENTATIONS

We tried Generalized SpecAugment (Soni et al., 2024) which, instead of masking blocks in the spectogram to zero, replaces them with white noise. Training data generation with this method yielded very broken sequences ($\approx 50\%$ WER) and we did not run DLM training on this data. To lower the WER, we tried Generalized SpecAugment during ASR training, but this led to a significantly worse ASR model, so we excluded this data augmentation method from further testing.

A max-pooling-like data augmentation method is proposed in Tóth et al. (2018), where only the loudest parts of the spectogram are kept, and the rest is masked to zero. The intuition behind this approach is that the features most robust to noise are usually the loudest ones, and that only these are needed to understand speech. We chose not to implement it for DLM training data generation due to similarity to SpecAugment and dropout, but it could be an interesting data augmentation method to try in future work.

### D.2.8 COMBINING MULTIPLE DATA AUGMENTATION TECHNIQUES

Parameters for all configurations are shown in Table D.16, and the respective hypotheses WERs of the training datasets are shown in Table D.17. The degradation caused by the data augmentation methods is quite severe, especially for the `high` dataset. Interestingly, for some of the configurations the WER of the TTS hypotheses is now higher than for dev-other, even though it was lower in almost all individual data augmentation experiments.

### D.3 ALTERNATIVE TRAINING INPUTS: DENSE k-PROBABILITY

Instead of having the DLM guess a correction for some label sequence, we can give it more information from the ASR model about which labels are considered probable alternatives.

Table D.17: Putting it all together: Training data (hypotheses) WER for different data augmentation configurations from Table D.16.

| Data Augmentation | DLM Training Data WER [%] | | |
|---|---|---|---|
| Configuration | dev-clean | dev-other | Glow-TTS |
| baseline | 1.75 | 4.13 | 2.17 |
| very low | 9.90 | 13.85 | 13.73 |
| stdPerturb | 18.89 | 24.93 | 19.55 |
| low | 16.45 | 20.02 | 20.15 |
| very low (ASR ep. 40) | 13.35 | 21.00 | 20.63 |
| low$^+$ | 17.06 | 22.24 | 21.87 |
| low medium | 18.38 | 26.06 | 31.98 |
| high (no ASR augment) | 26.91 | 27.73 | 36.65 |
| medium | 27.05 | 37.97 | 41.63 |
| high | 52.69 | 60.55 | 69.86 |

A conservative approximation to store the full probability distribution of the needed storage space to save the probability distributions for the entire LibriSpeech LM corpus reveals that this is not feasible:

$$\text{Dataset Size} \approx \text{num\_frames} \times \text{vocab\_size} \times \text{float\_size} \tag{22}$$

$$\geq \text{num\_words} \times \text{vocab\_size} \times \text{float\_size} \tag{23}$$

$$\approx 800 \text{ Million} \times 10000 \times 4 \tag{24}$$

$$\approx 32\text{TB} \tag{25}$$

We propose a more efficient approach:

1. Only store the top $k$ probabilities and their corresponding token indices

2. Instead of storing the probability distribution at every audio frame, do label synchronous search and store the label probabilities at every step

With this approach, we can reduce the storage requirements for a full generation of the LibriSpeech LM corpus to about 38GB with $k = 5$. Instead of storing the data as text lines like we do in our other experiments, we store this data using the HDF5 format[10]. Token substitution is adjusted to work with this data format to allow for any of the top-$k$ tokens from the beam to be substituted.

The input to the DLM encoder is then computed as a weighted sum of the top-$k$ token embeddings at every label position:

$$e_i = \sum_{j=1}^{k} p_{i,j} \cdot \text{emb}(t_{i,j}) \tag{26}$$

$$\text{with } p_{i,j} \text{ the probability of the } j\text{-th most probable token at position } i \tag{27}$$

$$\text{and } t_{i,j} \text{ the corresponding token index}$$

# E  RESULTS AND ANALYSIS DETAILS

We train DLMs using the training data augmentation techniques described in Section 3 and evaluate their performance on the LibriSpeech validation and test sets. Unless stated otherwise, all DLMs in this Chapter are trained for 5 epochs and contain 10x LibriSpeech ASR data. Sometimes we run experiments with the `stdPerturb` configuration, which is a combination of multiple data augmentation techniques that produces well-performing DLMs. Its exact parameters are described in Section 4.2. This configuration is not necessarily optimal, but we found it quite early in our research and therefore used it for many experiments. Sometimes we mention that an experiment is run without additional data augmentation, or do not explicitly state any data augmentation. In

---

[10]https://www.hdfgroup.org/solutions/hdf5/

Table E.1: Traditional LM vs DLM performance comparison. The DLM uses data augmentation configuration `low`. DLM rescore for the DLMs is rescore using only ASR hypotheses and thus compares to LM rescoring. The ASR model was trained on TTS data.

| LM | Num. Param. | Num. Epochs | Decoding | WER [%] | | | |
|---|---|---|---|---|---|---|---|
| | | | | dev-clean | dev-other | test-clean | test-other |
| None | 0 | 0 | greedy | 1.75 | 4.13 | 2.03 | 4.44 |
| n8-d1024 LM | 113M | 5 | rescoring | 1.63 | 3.59 | 1.83 | 3.90 |
| | | | one-pass | 1.62 | 3.53 | 1.81 | 3.82 |
| n32-d1024 LM | 422M | 5 | rescoring | 1.59 | 3.57 | 1.80 | 3.84 |
| | | | one-pass | 1.56 | 3.41 | 1.73 | 3.70 |
| | | 10 | rescoring | 1.61 | 3.54 | 1.81 | 3.83 |
| | | | one-pass | 1.60 | 3.45 | 1.74 | 3.72 |
| n32-d1280 LM | 663M | 5 | rescoring | 1.58 | 3.57 | 1.80 | 3.83 |
| | | | one-pass | 1.58 | 3.46 | 1.77 | 3.73 |
| DLM | 466M | 5 | greedy | 2.10 | 4.12 | 2.25 | 4.56 |
| | | | DLM rescore | 1.58 | 3.72 | 1.84 | 3.95 |
| | | | DSR | 1.53 | 3.50 | 1.76 | 3.81 |
| | | | DLM-sum | 1.51 | 3.40 | 1.74 | 3.66 |
| | | 10 | greedy | 2.31 | 4.06 | 2.32 | 4.57 |
| | | | DLM rescore | 1.59 | 3.68 | 1.85 | 3.88 |
| | | | DSR | 1.49 | 3.43 | 1.79 | 3.70 |
| | | | DLM-sum | 1.49 | 3.29 | 1.72 | 3.53 |

Table E.2: Traditional LM vs DLM performance comparison. The DLM uses data augmentation configuration `low`. DLM rescore for the DLMs is rescore using only ASR hypotheses and thus compares to LM rescoring. The ASR model was trained without TTS data, only on LibriSpeech 960h. This is the only difference to Table E.1.

| LM | Num. Param. | Num. Epochs | Decoding | WER [%] | | | |
|---|---|---|---|---|---|---|---|
| | | | | dev-clean | dev-other | test-clean | test-other |
| None | 0 | 0 | greedy | 2.29 | 5.02 | 2.42 | 5.33 |
| n32-d1024 LM | 422M | 5 | rescoring | 1.93 | 4.18 | 2.09 | 4.50 |
| | | | one-pass | 1.85 | 3.93 | 2.00 | 4.27 |
| | | 10 | rescoring | 1.95 | 4.22 | 2.09 | 4.56 |
| | | | one-pass | 1.90 | 3.94 | 2.00 | 4.27 |
| DLM | 466M | 10 | greedy | 2.49 | 4.63 | 2.41 | 5.19 |
| | | | DLM rescore | 1.98 | 4.32 | 2.11 | 4.70 |
| | | | DSR | 1.76 | 3.95 | 1.91 | 4.37 |
| | | | DLM-sum | 1.68 | 3.70 | 1.83 | 4.15 |

that case we still typically use TTS noise and length scales with noise scale $\sim \mathcal{U}(0.3, 0.9)$ and length scale $\sim \mathcal{U}(0.7, 1.1)$, as these are the default parameters of our training data generation process.

## E.1 DLM vs. Standard LM

We compare the DLMs (presented in Section 3.2) with these standard LMs (presented in Section 3.2): `n32-d1024` has exactly the same layers as the encoder+decoder of our DLMs with the same model dimension, and `n8-d1024` is exactly the same as the DLM decoder. Both models still have less parameters than our DLM because of the lack of cross attention and embedding in the encoder, thus we also include an even bigger LM with model dimension 1280. DSR decoding includes both the ASR and DLM beams for rescoring, while LM rescoring only includes the ASR beam. Thus, we also include DLM rescoring, which only rescores the ASR hypotheses. LMs and DLM performance is compared in Tables E.1 and E.2. DLMs and LMs trained with 5 epochs have matching performance, while with 10 epochs of training, DLMs surpass LMs. So the longer training is crucial to see the benefits of DLMs.

Table E.3: Putting it all together.

| Data Augmentation Configuration | Decoding | DLM Performance: WER [%] | | | |
|---|---|---|---|---|---|
| | | dev-clean | dev-other | test-clean | test-other |
| baseline | greedy | 1.82 | 3.98 | 2.05 | 4.49 |
| | DSR | 1.56 | 3.78 | 1.89 | 4.08 |
| | DLM-sum | 1.55 | 3.67 | 1.84 | 4.00 |
| very low | greedy | 1.87 | 4.09 | 2.12 | 4.47 |
| | DSR | 1.51 | 3.54 | 1.76 | 3.81 |
| | DLM-sum | 1.52 | 3.43 | 1.74 | 3.71 |
| stdPerturb | greedy | 1.95 | 4.05 | 2.25 | 4.60 |
| | DSR | 1.54 | 3.54 | 1.77 | 3.84 |
| | DLM-sum | 1.45 | 3.45 | 1.76 | 3.69 |
| low | greedy | 2.10 | 4.12 | 2.25 | 4.56 |
| | DSR | 1.53 | 3.50 | 1.76 | 3.81 |
| | DLM-sum | 1.51 | 3.40 | 1.74 | 3.66 |
| very low (ASR ep. 40) | greedy | 2.10 | 4.15 | 2.43 | 4.81 |
| | DSR | 1.56 | 3.50 | 1.79 | 3.82 |
| | DLM-sum | 1.50 | 3.41 | 1.72 | 3.64 |
| low$^+$ | greedy | 2.12 | 4.06 | 2.32 | 4.42 |
| | DSR | 1.51 | 3.47 | 1.79 | 3.82 |
| | DLM-sum | 1.49 | 3.40 | 1.75 | 3.60 |
| low medium | greedy | 2.45 | 4.25 | 2.60 | 5.05 |
| | DSR | 1.52 | 3.50 | 1.79 | 3.84 |
| | DLM-sum | 1.51 | 3.44 | 1.73 | 3.76 |
| high (no ASR augment) | greedy | 2.03 | 4.14 | 2.45 | 4.76 |
| | DSR | 1.57 | 3.58 | 1.79 | 3.92 |
| | DLM-sum | 1.50 | 3.49 | 1.74 | 3.79 |
| medium | greedy | 2.37 | 4.37 | 2.66 | 5.14 |
| | DSR | 1.60 | 3.52 | 1.80 | 3.84 |
| | DLM-sum | 1.56 | 3.42 | 1.75 | 3.72 |
| high | greedy | 4.71 | 5.66 | 4.50 | 6.58 |
| | DSR | 1.61 | 3.60 | 1.82 | 3.95 |
| | DLM-sum | 1.67 | 3.56 | 1.81 | 3.89 |

### E.2 IMPACT OF TRAINING DATA GENERATION STRATEGIES

#### E.2.1 COMBINING DATA AUGMENTATION TECHNIQUES

We investigate how combining multiple data augmentation techniques (see Appendix D.2.8) affects DLM performance. The different configurations are described in Appendix D.2.8. Results of DLM training are shown in Table E.3. Notably, the `high (no ASR augment)` configuration performs worse than the similar in WER configurations `low medium` and `medium`, indicating that artificial augmentations for the ASR model are essential to getting good DLM performance.

#### E.2.2 TRAINING LONGER

For this ablation, we extend the learning rate schedule by stretching it out over more epochs, i.e. if we double the number of epochs, the learning rate increases and decreases twice as slowly but we keep the same minimum and maximum learning rate. Learning rate is halved for the 5 epoch configuration, but for the higher epoch configurations we keep the same peak learning rate. All other hyperparameters are kept the same, and we use the `stdPerturb` data augmentation configuration for all models. The results are shown in Table E.4. Going from 5 to 10 epochs helps, but we see diminishing returns after that. We hypothesize this may be caused by the learning rate schedule not being optimal for longer training, or the DLM reaching its plateau. To test the first hypothesis, we run an additional experiment with a step decay learning rate schedule, inspired by the one used in

Table E.4: DLMs trained for different numbers of epochs. All models use data augmentation configuration `stdPerturb`. The first model is trained for 5 epochs with a lower base learning rate of 0.5 instead of 1.0.

| DLM Training Epochs | Decoding | DLM Performance: WER [%] | | | |
|---|---|---|---|---|---|
| | | dev-clean | dev-other | test-clean | test-other |
| 5 | greedy | 1.95 | 4.05 | 2.25 | 4.60 |
| | DSR | 1.54 | 3.54 | 1.77 | 3.84 |
| | DLM-sum | 1.45 | 3.45 | 1.76 | 3.69 |
| 10 | greedy | 2.26 | 3.96 | 2.47 | 4.70 |
| | DSR | 1.51 | 3.46 | 1.78 | 3.73 |
| | DLM-sum | 1.50 | 3.32 | 1.80 | 3.57 |
| 15 | greedy | 2.60 | 4.00 | 2.60 | 5.11 |
| | DSR | 1.51 | 3.43 | 1.72 | 3.73 |
| | DLM-sum | 1.49 | 3.34 | 1.71 | 3.61 |
| 20 | greedy | 2.40 | 4.03 | 2.35 | 4.70 |
| | DSR | 1.47 | 3.44 | 1.77 | 3.68 |
| | DLM-sum | 1.47 | 3.34 | 1.74 | 3.53 |

Table E.5: A single DLM trained with step decay learning rate schedule and evaluated at different epochs. Model uses data augmentation configuration `stdPerturb`.

| Step decay results per Epoch | Decoding | DLM Performance: WER [%] | | | |
|---|---|---|---|---|---|
| | | dev-clean | dev-other | test-clean | test-other |
| 4 | greedy | 2.24 | 4.17 | 2.69 | 4.75 |
| | DSR | 1.56 | 3.62 | 1.84 | 3.92 |
| | DLM-sum | 1.53 | 3.51 | 1.76 | 3.79 |
| 8 | greedy | 2.61 | 4.02 | 2.59 | 5.11 |
| | DSR | 1.54 | 3.55 | 1.81 | 3.84 |
| | DLM-sum | 1.54 | 3.46 | 1.77 | 3.73 |
| 16 | greedy | 3.15 | 4.46 | 3.27 | 5.40 |
| | DSR | 1.47 | 3.50 | 1.78 | 3.80 |
| | DLM-sum | 1.51 | 3.36 | 1.78 | 3.73 |
| 25 | greedy | 2.89 | 4.48 | 3.16 | 5.57 |
| | DSR | 1.51 | 3.44 | 1.71 | 3.77 |
| | DLM-sum | 1.50 | 3.34 | 1.70 | 3.62 |
| 32 | greedy | 2.99 | 4.45 | 3.45 | 5.51 |
| | DSR | 1.55 | 3.42 | 1.74 | 3.70 |
| | DLM-sum | 1.55 | 3.33 | 1.79 | 3.65 |

Gu et al. (2024): warmup for 2.17 epochs to 0.0005, then constant for another 10.2 epochs, then decay by a factor of 0.5 every 6.8 epochs and stop training after a total of 32 epochs[11]. Results for different checkpoints of this training run are shown in Table E.5. It appears that we reach a similar performance ceiling with the step decay learning rate schedule.

We also train a DLM with the `low` data augmentation configuration for 10 epochs with our usual learning rate schedule, which results in our best DLM, see Table E.6.

### E.2.3  TTS NOISE & TTS LENGTH

We sample the noise level $T$ uniformly for every sequence, and fix a minimum noise level of $\tau_{\min} = 0.3$ and vary $\tau_{\max} \in \{0.6, 0.9, 1.2, 1.5\}$. Results are shown in Table E.7. Performance seems to increase with rising noise scale, and we see consistent gains even up to $\tau_{\max} = 1.5$.

---

[11]Gu et al. (2024) use peak lr 0.001. Through preliminary experiments we found that this was unstable, so we reduced lr by factor 0.5. We believe this is necessary because our batch size is 8x smaller.

Table E.6: DLMs trained with data augmentation configuration `low`.

| DLM Training Epochs | Decoding | DLM Performance: WER [%] | | | |
|---|---|---|---|---|---|
| | | dev-clean | dev-other | test-clean | test-other |
| 5 epochs | greedy | 2.10 | 4.12 | 2.25 | 4.56 |
| | DSR | 1.53 | 3.50 | 1.76 | 3.81 |
| | DLM-sum | 1.51 | 3.40 | 1.74 | 3.66 |
| 10 epochs | greedy | 2.31 | 4.06 | 2.32 | 4.57 |
| | DSR | 1.49 | 3.43 | 1.79 | 3.70 |
| | DLM-sum | 1.49 | 3.29 | 1.72 | 3.53 |

Table E.7: TTS Noise Scale Ablation Experiment. For each sequence, a noise scale is uniformly sampled from $\mathcal{U}(\tau_{\min}, \tau_{\max})$.

| $(\tau_{\min}, \tau_{\max})$ | Decoding | DLM Performance: WER [%] | | | |
|---|---|---|---|---|---|
| | | dev-clean | dev-other | test-clean | test-other |
| (0.3,0.6) | greedy | 1.69 | 4.00 | 2.03 | 4.39 |
| | DSR | 1.59 | 3.81 | 1.89 | 4.12 |
| | DLM-sum | 1.55 | 3.73 | 1.84 | 4.00 |
| (0.3,0.9) | greedy | 1.83 | 4.15 | 2.04 | 4.50 |
| | DSR | 1.57 | 3.78 | 1.89 | 4.12 |
| | DLM-sum | 1.54 | 3.68 | 1.80 | 3.97 |
| (0.3,1.2) | greedy | 1.86 | 4.01 | 2.12 | 4.59 |
| | DSR | 1.60 | 3.70 | 1.86 | 4.04 |
| | DLM-sum | 1.55 | 3.62 | 1.82 | 3.92 |
| (0.3,1.5) | greedy | 1.96 | 4.23 | 2.24 | 4.79 |
| | DSR | 1.54 | 3.67 | 1.82 | 3.97 |
| | DLM-sum | 1.51 | 3.61 | 1.76 | 3.83 |

Table E.8: TTS Length Scale Ablation Experiment. For each sequence, a length scale is uniformly sampled from $\mathcal{U}(d_{\min}, d_{\max})$.

| $(d_{\min}, d_{\max})$ | Decoding | DLM Performance: WER [%] | | | |
|---|---|---|---|---|---|
| | | dev-clean | dev-other | test-clean | test-other |
| (1.0, 1.0) | greedy | 1.83 | 4.15 | 2.04 | 4.50 |
| | DSR | 1.57 | 3.78 | 1.89 | 4.12 |
| | DLM-sum | 1.54 | 3.68 | 1.80 | 3.97 |
| (0.6, 2.0) | greedy | 1.66 | 4.00 | 2.03 | 4.37 |
| | DSR | 1.62 | 3.77 | 2.10 | 4.10 |
| | DLM-sum | 1.54 | 3.71 | 1.82 | 3.97 |
| (0.5, 2.5) | greedy | 1.73 | 3.99 | 2.01 | 4.42 |
| | DSR | 1.55 | 3.75 | 1.87 | 4.06 |
| | DLM-sum | 1.51 | 3.68 | 1.81 | 3.92 |
| (0.4, 3.0) | greedy | 1.69 | 3.99 | 2.04 | 4.34 |
| | DSR | 1.58 | 3.69 | 1.85 | 4.02 |
| | DLM-sum | 1.56 | 3.68 | 1.83 | 3.95 |

We choose multiple minimum and maximum length scales such that the WER at the minimum and maximum length scales is roughly equal. Then, we sample the length scale uniformly between the minimum and maximum for every sequence. Training data WER for these configurations is shown in Table D.4. Results are shown in Table E.8. There does not seem to be any noticeable difference between the different length scales we test. With increasing length scale, the audio data increases in length and thus training data generation becomes slower, so we stick to more moderate length scales $\sim \mathcal{U}(0.7, 1.1)$ for all other experiments.

Table E.9: Different TTS systems and their combinations used for generating hypotheses. First three use no data augmentation, last three use data augmentation configuration `stdPerturb`. Trained for 5 epochs.

| TTS System | Decoding | DLM Performance: WER [%] | | | |
|---|---|---|---|---|---|
| | | dev-clean | dev-other | test-clean | test-other |
| Glow-TTS | greedy | 1.82 | 3.98 | 2.05 | 4.49 |
| | DSR | 1.56 | 3.78 | 1.89 | 4.08 |
| | DLM-sum | 1.55 | 3.67 | 1.84 | 4.00 |
| YourTTS | greedy | 2.31 | 4.53 | 2.46 | 4.93 |
| | DSR | 1.60 | 3.74 | 1.83 | 3.96 |
| | DLM-sum | 1.58 | 3.70 | 1.81 | 3.88 |
| 50% Glow-TTS, 50% YourTTS | greedy | 1.84 | 4.16 | 2.11 | 4.38 |
| | DSR | 1.58 | 3.70 | 1.83 | 3.96 |
| | DLM-sum | 1.57 | 3.64 | 1.82 | 3.91 |
| Glow-TTS (stdPerturb) | greedy | 1.95 | 4.05 | 2.25 | 4.60 |
| | DSR | 1.54 | 3.54 | 1.77 | 3.84 |
| | DLM-sum | 1.45 | 3.45 | 1.76 | 3.69 |
| YourTTS (stdPerturb) | greedy | 2.58 | 4.69 | 2.88 | 5.51 |
| | DSR | 1.58 | 3.58 | 1.79 | 3.84 |
| | DLM-sum | 1.55 | 3.51 | 1.74 | 3.76 |
| 50% Glow-TTS, 50% YourTTS (both stdPerturb) | greedy | 2.19 | 4.02 | 2.26 | 4.66 |
| | DSR | 1.55 | 3.53 | 1.80 | 3.82 |
| | DLM-sum | 1.49 | 3.48 | 1.73 | 3.79 |

It may be necessary to choose a different random distribution that favors extreme values ($\rightsquigarrow$ Beta distribution with $\alpha = \beta$) because most values in the middle do not seem to affect the ASR system performance much, but we leave this to future work.

### E.2.4 COMBINING TTS SYSTEMS

We combine the two TTS systems to generate more diverse synthetic audio data. First we train baseline DLMs using each TTS system individually, then we combine the two systems by passing half the text data through one system and the other half through the second system. We run the same experiments again using the `stdPerturb` data augmentation configuration. Results are shown in Table E.9.

It is difficult to determine which TTS system or combination performs best, and it seems that the data augmentation configuration has a far greater impact on DLM performance than the choice of TTS system. If there is a positive effect of combining multiple TTS systems as reported by Gu et al. (2024), it is too small for us to measure or our system of choice (YourTTS) does not differ enough from the Glow-TTS system. It is also possible that all data has to be generated by both TTS systems to get more variations of the same text, but the results from our ablation 'More training data' in Appendix E.2.14 do not indicate that significant gains should be expected from this approach.

### E.2.5 EARLY ASR CHECKPOINTS

We choose epochs 10, 40 and 100 (final) of our spm10k (TTS) ASR model to generate DLM training data. Results are shown in Table E.10. While greedy performance decreases with earlier ASR checkpoints, DSR and DLM-sum performance improves. Going from epoch 100 to 40 yields gains, but going from 40 to 10 brings little improvement. even though the difference in WER of the training data is quite significant. This suggest that this early ASR checkpoint may already be too different from the final ASR model and the additional WER in the training data does not seem to help the DLM.

Table E.10: DLM ablation results with training data from early ASR checkpoints. Trained for 5 epochs. See Table D.9 about the training data (hypotheses) WERs.

| ASR checkpoint (of 100) | Decoding | DLM Performance: WER [%] | | | |
|---|---|---|---|---|---|
| | | dev-clean | dev-other | test-clean | test-other |
| Epoch 10 | greedy | 2.06 | 4.17 | 2.23 | 4.78 |
| | DSR | 1.59 | 3.66 | 1.80 | 3.87 |
| | DLM-sum | 1.56 | 3.55 | 1.77 | 3.87 |
| Epoch 40 | greedy | 1.86 | 4.07 | 2.16 | 4.70 |
| | DSR | 1.58 | 3.70 | 1.82 | 3.94 |
| | DLM-sum | 1.52 | 3.57 | 1.76 | 3.88 |
| Epoch 100 | greedy | 1.82 | 3.98 | 2.05 | 4.49 |
| | DSR | 1.56 | 3.78 | 1.89 | 4.08 |
| | DLM-sum | 1.55 | 3.67 | 1.84 | 4.00 |

Table E.11: SpecAugment ablation experiment for DLM training. Trained for 5 epochs. See Section 3.3 for the DLM training data WERs for each case.

| SpecAugment | Decoding | DLM Performance: WER [%] | | | |
|---|---|---|---|---|---|
| | | dev-clean | dev-other | test-clean | test-other |
| Off | greedy | 1.82 | 3.98 | 2.05 | 4.49 |
| | DSR | 1.56 | 3.78 | 1.89 | 4.08 |
| | DLM-sum | 1.55 | 3.67 | 1.84 | 4.00 |
| On (only frequency masking) | greedy | 1.94 | 3.99 | 2.02 | 4.42 |
| | DSR | 1.57 | 3.77 | 1.89 | 4.01 |
| | DLM-sum | 1.58 | 3.67 | 1.81 | 3.92 |
| On (only time masking) | greedy | 1.87 | 4.05 | 2.10 | 4.49 |
| | DSR | 1.54 | 3.62 | 1.80 | 3.91 |
| | DLM-sum | 1.52 | 3.53 | 1.78 | 3.83 |
| On (time + frequency) | greedy | 1.89 | 4.03 | 2.17 | 4.50 |
| | DSR | 1.54 | 3.62 | 1.79 | 3.87 |
| | DLM-sum | 1.49 | 3.47 | 1.76 | 3.76 |

### E.2.6   SPECAUGMENT

We test all configurations described in Section 3.3. For this ablation we resplit subwords (cf. Appendix E.2.11) and only use 1x LibriSpeech ASR data due to an oversight in parameter selection. Results are shown in Table E.11. We see improving performance with increasing WER of the training data, and the best performance for SpecAugment with time and frequency masking. Even with the small difference with only frequency masking from the baseline configuration, we already see an improvement for $\{dev, test\}$-other of $0.1\%$. Performance of the highest configuration roughly matches those of the dropout ablation with similar training data WER.

### E.2.7   DROPOUT

For each sentence, the dropout in the ASR model is sampled $p \sim \mathcal{U}(p_{\min}, p_{\max})$. We test multiple configurations where we keep $p_{\min} = 0$ but vary $p_{\max} \in \{0.0, 0.1, 0.2, 0.5, 0.9\}$. We also test whether grounding $p_{\min}$ in zero is necessary and vary $p_{\min} \in \{0.1, 0.2\}$ while we fix $p_{\max} = 0.5$. For this ablation we resplit subwords (cf. Appendix E.2.11) and only use 1x LibriSpeech ASR data due to an oversight in parameter selection. Results are shown in Table E.12.

It seems that $p \sim \mathcal{U}(0.0, 0.5)$ is the sweet spot for $p_{\min} = 0$, and performance decreases in either direction. Surprisingly, we get even better results if we also increase $p_{\min}$, and we get our best results with $p \sim \mathcal{U}(0.1, 0.5)$ using DLM-sum decoding on test-other. This is unexpected, as this configuration moves it away from the test-time data distribution, as the training data will always have at least $10\%$ dropout applied. One may hypothesize that there could be some optimal WER (or similar metric) which best trains the DLM, and with the more varied dropouts we move our train-

Table E.12: Dropout Ablation Experiment. Trained for 5 epochs.

| $(p_{\min}, p_{\max})$ | Decoding | DLM Performance: WER [%] | | | |
|---|---|---|---|---|---|
| | | dev-clean | dev-other | test-clean | test-other |
| (0.0,0.0) | greedy | 1.82 | 3.98 | 2.05 | 4.49 |
| | DSR | 1.56 | 3.78 | 1.89 | 4.08 |
| | DLM-sum | 1.55 | 3.67 | 1.84 | 4.00 |
| (0.0,0.1) | greedy | 1.70 | 3.99 | 2.00 | 4.46 |
| | DSR | 1.61 | 3.80 | 1.89 | 4.09 |
| | DLM-sum | 1.57 | 3.69 | 1.81 | 3.91 |
| (0.0,0.2) | greedy | 1.94 | 4.00 | 2.15 | 4.40 |
| | DSR | 1.61 | 3.75 | 1.84 | 4.07 |
| | DLM-sum | 1.54 | 3.64 | 1.83 | 3.91 |
| (0.0,0.5) | greedy | 1.90 | 3.94 | 1.99 | 4.54 |
| | DSR | 1.54 | 3.63 | 1.82 | 3.92 |
| | DLM-sum | 1.49 | 3.52 | 1.81 | 3.85 |
| (0.0,0.9) | greedy | 2.07 | 4.17 | 2.09 | 5.03 |
| | DSR | 1.62 | 3.66 | 1.82 | 4.05 |
| | DLM-sum | 1.59 | 3.55 | 1.83 | 3.95 |
| (0.1,0.5) | greedy | 1.89 | 3.97 | 2.06 | 4.52 |
| | DSR | 1.55 | 3.64 | 1.82 | 3.92 |
| | DLM-sum | 1.49 | 3.54 | 1.74 | 3.84 |
| (0.2,0.5) | greedy | 1.98 | 3.96 | 2.09 | 4.45 |
| | DSR | 1.54 | 3.57 | 1.78 | 3.83 |
| | DLM-sum | 1.52 | 3.52 | 1.76 | 3.78 |

ing data distribution approximately in the right direction but with high variance, while the dropout configuration with higher $p_{\min}$ also approaches this optimal WER but with less variance.

It is also interesting to see that the $\mathcal{U}(0.0, 0.9)$ configuration still performs reasonably well, even though a large portion of the hypotheses provide very little useful information.

### E.2.8 TOKEN SUBSTITUTION

We test constant token substitution for $p \in \{0.0, 0.05, 0.1, 0.2, 0.3\}$ and varying token substitution with $p_{\min} = 0.0$ and $p_{\max} \in \{0.4, 0.9\}$. Results are shown in Table E.13.

It appears that, as long as there is any level of token substitution, we get consistent gains of about 0.2% for {dev, test}-other. Only with $p_{\max} = 0.9$ do we start to see a significant degradation, as the input data becomes very unreliable for the DLM.

### E.2.9 MIXUP

We sample the mixing factor $\lambda$ from a uniform random distribution $\mathcal{U}(0, \lambda_{\max})$ with $\lambda_{\max} \in \{0.0, 0.2, 0.4, 0.6, 0.8\}$. When $\lambda > 0.5$, the noise audio from other sequences starts to dominate, and the audio becomes almost unrecognizable for the ASR system, as shown in Figure D.4. Results for different $\lambda_{\max}$ are shown in Table E.14.

We see gains until $\lambda_{\max} = 0.6$, after which the greedy performance degrades significantly.

### E.2.10 SAMPLING FROM ASR MODEL

We use the sampling procedure as described in Section 3.3 with $k \in \{1, 16\}$. Results are shown in Table E.15.

We see less gains as expected for the increase in WER, and conclude that Top-k Sampling does not meaningfully help DLM performance.

Table E.13: Token Substitution Ablation Experiment. Trained for 5 epochs.

| Token substitution | Decoding | DLM Performance: WER [%] | | | |
|---|---|---|---|---|---|
| | | dev-clean | dev-other | test-clean | test-other |
| 0% to 0% | greedy | 1.82 | 3.98 | 2.05 | 4.49 |
| | DSR | 1.56 | 3.78 | 1.89 | 4.08 |
| | DLM-sum | 1.55 | 3.67 | 1.84 | 4.00 |
| 5% to 5% | greedy | 1.71 | 3.98 | 2.07 | 4.47 |
| | DSR | 1.53 | 3.62 | 1.80 | 3.94 |
| | DLM-sum | 1.50 | 3.53 | 1.77 | 3.83 |
| 10% to 10% | greedy | 1.82 | 4.00 | 2.16 | 4.40 |
| | DSR | 1.57 | 3.62 | 1.81 | 3.93 |
| | DLM-sum | 1.52 | 3.53 | 1.76 | 3.79 |
| 20% to 20% | greedy | 1.90 | 4.11 | 2.35 | 4.49 |
| | DSR | 1.56 | 3.58 | 1.83 | 3.92 |
| | DLM-sum | 1.50 | 3.47 | 1.80 | 3.79 |
| 30% to 30% | greedy | 2.15 | 4.21 | 2.44 | 4.72 |
| | DSR | 1.57 | 3.62 | 1.83 | 4.00 |
| | DLM-sum | 1.54 | 3.50 | 1.80 | 3.82 |
| 0% to 40% | greedy | 1.79 | 3.98 | 2.18 | 4.41 |
| | DSR | 1.55 | 3.58 | 1.83 | 3.92 |
| | DLM-sum | 1.50 | 3.49 | 1.75 | 3.78 |
| 0% to 90% | greedy | 2.06 | 4.01 | 2.27 | 4.65 |
| | DSR | 1.57 | 3.67 | 1.85 | 3.99 |
| | DLM-sum | 1.55 | 3.58 | 1.78 | 3.94 |

Table E.14: Mixup Ablation Experiment. $\lambda \sim \mathcal{U}(0, \lambda_{\max})$. Trained for 5 epochs.

| Mixup $\lambda_{\max}$ | Decoding | DLM Performance: WER [%] | | | |
|---|---|---|---|---|---|
| | | dev-clean | dev-other | test-clean | test-other |
| 0.0 | greedy | 1.82 | 3.98 | 2.05 | 4.49 |
| | DSR | 1.56 | 3.78 | 1.89 | 4.08 |
| | DLM-sum | 1.55 | 3.67 | 1.84 | 4.00 |
| 0.2 | greedy | 1.91 | 4.01 | 2.07 | 4.69 |
| | DSR | 1.60 | 3.75 | 1.89 | 4.04 |
| | DLM-sum | 1.54 | 3.65 | 1.84 | 3.96 |
| 0.4 | greedy | 1.80 | 3.99 | 2.03 | 4.43 |
| | DSR | 1.56 | 3.67 | 1.85 | 3.98 |
| | DLM-sum | 1.51 | 3.58 | 1.77 | 3.87 |
| 0.6 | greedy | 3.03 | 4.05 | 2.70 | 4.93 |
| | DSR | 1.59 | 3.65 | 1.85 | 4.00 |
| | DLM-sum | 1.55 | 3.55 | 1.79 | 3.87 |
| 0.8 | greedy | 4.39 | 5.25 | 4.38 | 6.53 |
| | DSR | 1.58 | 3.67 | 1.79 | 4.00 |
| | DLM-sum | 1.99 | 3.55 | 1.82 | 4.00 |

One could try to use a different sampling procedure, for example with label-synchronous search as we do in Section 3.4 or increasing diversity with softmax temperature, but we leave these ideas for future work.

### E.2.11 RESPLIT SUBWORDS

In this work, both the ASR model and the DLM model use the same vocabulary, which for `spm10k` and `spm128` is based on subword units. For vocabularies like this, it is possible to have the same word represented by multiple different subword sequences, such as "example" being represented as

Table E.15: Top-k Sampling Ablation Experiment. Trained for 5 epochs.

| $k$ | Decoding | DLM Performance: WER [%] | | | |
|---|---|---|---|---|---|
| | | dev-clean | dev-other | test-clean | test-other |
| 1 (baseline) | greedy | 1.82 | 3.98 | 2.05 | 4.49 |
| | DSR | 1.56 | 3.78 | 1.89 | 4.08 |
| | DLM-sum | 1.55 | 3.67 | 1.84 | 4.00 |
| 16 | greedy | 1.71 | 3.98 | 2.05 | 4.56 |
| | DSR | 1.53 | 3.73 | 1.87 | 4.00 |
| | DLM-sum | 1.55 | 3.67 | 1.83 | 3.95 |

Table E.16: We test whether ASR hypotheses should be normalized by resplitting the subwords before being passed to the DLM. This experiment uses 1x LibriSpeech ASR data, and DLMs are trained for 5 epochs.

| Resplit subwords | Decoding | DLM Performance: WER [%] | | | |
|---|---|---|---|---|---|
| | | dev-clean | dev-other | test-clean | test-other |
| keep ASR subwords | greedy | 1.74 | 3.97 | 2.02 | 4.44 |
| | DSR | 1.62 | 3.74 | 1.90 | 4.11 |
| | DLM-sum | 1.53 | 3.67 | 1.83 | 3.98 |
| resplit subwords | greedy | 1.78 | 4.03 | 2.08 | 4.36 |
| | DSR | 1.59 | 3.79 | 1.89 | 4.12 |
| | DLM-sum | 1.63 | 3.81 | 1.95 | 4.13 |

"exam" + "ple" or "ex" + "ample". The tokenizer implements a deterministic mapping of words to subword tokens, but it is not guaranteed that the models will always output sequences that match the tokenizers' mapping. This is especially relevant for DLMs, because the ASR hypotheses are fed directly into the DLM as token sequences. This brings up the question if there is a benefit to merging the subwords back to words and then deterministically re-splitting them into subwords, and thus guaranteeing a consistent word-to-token mapping before feeding them into the DLM. We test both configurations, and show results in Table E.16.

We observe that re-splitting the subwords into words and back does not lead to a significant change for greedy or DSR decoding, and even leads to a slight degradation in DLM-sum decoding.

### E.2.12 ADDITIONAL PHONEME REPRESENTATIONS

During our research, we noticed a bug in the training data generation process, where the TTS system was given phoneme representations which it was not trained on (but which are otherwise valid). This leads to a noticeable degradation in the WER of the ASR hypotheses, from which we conclude that the TTS system is not able to generate accurate audio for these phoneme sequences. Regardless, one may interpret this as a form of data augmentation for the TTS system, and we compare the performance of this "bad" training data with the "good" training data, which uses the phoneme representations that the TTS system is trained on. Results are shown in Table E.17.

The results show that the "bad" phoneme representations are not a good form of data augmentation and even lead to a slight performance drop. Unless otherwise stated, we use the "good" phoneme representations for all other experiments in this work.

### E.2.13 ASR TRAINED WITH AND WITHOUT TTS DATA

As mentioned in Section 3.2, we have two different ASR models available with the SentencePiece 10k subword vocabulary: one trained just on LibriSpeech ASR data, and one additionally trained with TTS data generated from the LibriSpeech LM corpus. The latter model significantly outperforms the former in terms of WER, but the question remains which one is better suited for generating hypotheses for DLM training. For both models we generate a full DLM training dataset with the LibriSpeech LM corpus without any data augmentation (only TTS noise and length sampling) and train DLMs for 10 epochs. Results are shown in Table E.18.

Table E.17: A bug in our implementation used additional phoneme representations that the TTS was not trained on. We test whether these additional phoneme representations are a good form of data augmentation. The first two models are trained without additional data augmentation, the last two with data augmentation configuration `stdPerturb`. All trained for 10 epochs.

| Phoneme Representation | Decoding | DLM Performance: WER [%] | | | |
|---|---|---|---|---|---|
| | | dev-clean | dev-other | test-clean | test-other |
| Additional phoneme representations | greedy | 2.14 | 4.42 | 2.38 | 4.76 |
| | DSR | 1.57 | 3.72 | 1.87 | 3.95 |
| | DLM-sum | 1.59 | 3.68 | 1.84 | 3.99 |
| Compatible to TTS | greedy | 1.96 | 4.04 | 2.19 | 4.66 |
| | DSR | 1.58 | 3.71 | 1.87 | 4.02 |
| | DLM-sum | 1.53 | 3.61 | 1.81 | 3.89 |
| Additional phoneme representations (stdPerturb) | greedy | 2.71 | 4.21 | 2.74 | 5.25 |
| | DSR | 1.56 | 3.50 | 1.82 | 3.75 |
| | DLM-sum | 1.49 | 3.39 | 1.78 | 3.68 |
| Compatible to TTS (stdPerturb) | greedy | 2.26 | 3.96 | 2.47 | 4.70 |
| | DSR | 1.51 | 3.46 | 1.78 | 3.73 |
| | DLM-sum | 1.50 | 3.32 | 1.80 | 3.57 |

Table E.18: Comparison of DLMs trained from hypotheses of an ASR model trained with and without TTS data. All models trained for 10 epochs and without additional data augmentation.

| ASR TTS vs non-TTS | Decoding | DLM Performance: WER [%] | | | |
|---|---|---|---|---|---|
| | | dev-clean | dev-other | test-clean | test-other |
| spm10k (TTS) | greedy | 1.96 | 4.04 | 2.19 | 4.66 |
| | DSR | 1.58 | 3.71 | 1.87 | 4.02 |
| | DLM-sum | 1.53 | 3.61 | 1.81 | 3.89 |
| spm10k | greedy | 2.63 | 5.09 | 2.67 | 5.88 |
| | DSR | 1.69 | 4.03 | 1.89 | 4.41 |
| | DLM-sum | 1.68 | 3.88 | 1.89 | 4.29 |
| spm10k, eval with spm10k (TTS) | greedy | 2.53 | 4.54 | 2.61 | 5.24 |
| | DSR | 1.53 | 3.57 | 1.80 | 3.87 |
| | DLM-sum | 1.51 | 3.44 | 1.75 | 3.79 |

Initially, it seems that the DLM trained with the non-TTS ASR model is worse, but when evaluated with the TTS ASR model it actually outperforms the TTS ASR DLM. This is quite a surprising result, as one would expect that the best DLMs for a particular ASR model would be the ones trained with hypotheses from that same ASR model. We run another experiment where we generate data with data augmentation parameters that are known to produce a good DLM (`stdPerturb` from Section 4.2) and train another two DLMs. Results are shown in Table E.19. While the performance of both DLMs has improved, the gains for the non-TTS ASR DLM are smaller and the TTS ASR DLM is now slightly better. It appears that either the DLMs have hit some ceiling on performance gains, the `stdPerturb` configuration favors the TTS-trained ASR model, or that the gap between the worse hypotheses from the non-TTS trained ASR model was closed by the error-inducing data augmentation configuration. In Section 4.3 we pursue this theory, and test which underlying factors of the training data lead to better gains in DLM performance.

### E.2.14   MORE TRAINING DATA

Our training data generation process with data augmentation is not inherently deterministic. Augmentations such as dropout, mixup and SpecAugment depend on random sampling, and we explicitly add noise to the latent vector as part of the TTS audio generation process. We enforce some level of determinism by setting a fixed random seed at the beginning every training data generation run, but this does not guarantee that the same training data will be generated when some augmentations are flipped on or off. But one may argue that this source of randomness is beneficial to DLM training, as it ensures that the DLM is trained on a more diverse set of hypotheses. Therefore we

Table E.19: Comparison of DLMs trained from hypotheses of an ASR model trained with and without TTS data. All models trained for 10 epochs with data augmentation configuration `stdPerturb`.

| ASR TTS vs non-TTS (stdPerturb) | Decoding | DLM Performance: WER [%] | | | |
|---|---|---|---|---|---|
| | | dev-clean | dev-other | test-clean | test-other |
| spm10k (TTS) | greedy | 2.26 | 3.96 | 2.47 | 4.70 |
| | DSR | 1.51 | 3.46 | 1.78 | 3.73 |
| | DLM-sum | 1.50 | 3.32 | 1.80 | 3.57 |
| spm10k | greedy | 3.39 | 5.19 | 3.04 | 6.11 |
| | DSR | 1.78 | 3.87 | 1.87 | 4.29 |
| | DLM-sum | 1.71 | 3.76 | 1.90 | 4.27 |
| spm10k, eval with spm10k (TTS) | greedy | 3.46 | 4.55 | 3.01 | 5.46 |
| | DSR | 1.63 | 3.46 | 1.75 | 3.77 |
| | DLM-sum | 1.56 | 3.35 | 1.77 | 3.68 |

Table E.20: DLM ablation to test whether using additional data from the same text helps. Additional data is made using a different random seed. Trained for 10 epochs, with data augmentation configuration `stdPerturb`.

| Num hypotheses | Decoding | DLM Performance: WER [%] | | | |
|---|---|---|---|---|---|
| | | dev-clean | dev-other | test-clean | test-other |
| 1 | greedy | 2.71 | 4.21 | 2.74 | 5.25 |
| | DSR | 1.56 | 3.50 | 1.82 | 3.75 |
| | DLM-sum | 1.49 | 3.39 | 1.78 | 3.68 |
| 5 | greedy | 2.71 | 4.18 | 2.78 | 5.01 |
| | DSR | 1.56 | 3.44 | 1.79 | 3.74 |
| | DLM-sum | 1.51 | 3.35 | 1.75 | 3.66 |

experiment with generating a unique training dataset for more DLM epochs instead of reusing the same generated data every DLM epoch. We expect additional regularisation effects due to the DLM seeing different variations of the same text during training, which may improve generalisation. Results are shown in Table E.20. We observe very minor improvements in WER, which we deem not statistically significant. We present some plausible explanations for this:

- Token substitution is implemented as part of DLM training, thus it already reduces overfitting by making the inputs more diverse in each epoch.
- The training dataset is so large that there is little need for additional regularization.
- The differences between the training data of different random seeds is too small to have a significant impact.
- We hit some performance ceiling, either in model capacity or the quality of the training data.

We can not conclusively determine the reason for the lack of improvement, but we can confidently say that the additional effort of generating a new training dataset for every epoch is not worth it[12].

This experiment used phoneme representations of the input text which were different from those used in TTS training, thus the training data produced a slightly worse DLM (cf. Appendix E.2.12).

E.2.15   LIBRISPEECH ASR TRAINING DATA

We use both the synthetic TTS data and the real audio from the LibriSpeech ASR training dataset to generate hypotheses for DLM training. In their work, Gu et al. (2024) determine that a ratio of 1:1 between TTS and real audio training data is optimal for ASR model training, but it remains an open question what the optimal ratio between real and synthetic audio is for DLM training.

---

[12]Generating a single epoch of training data for the DLM is roughly the same computational effort as training a DLM for 5.8 epochs.

Table E.21: We vary the amount of LibriSpeech ASR training data used. 0x means no LibriSpeech ASR data is used, 40x means 40 times the standard amount (38400 hours). TTS audio of LibriSpeech LM corpus is about 75000 hours. Trained for 5 epochs.

| LibriSpeech ASR data | Decoding | DLM Performance: WER [%] | | | |
|---|---|---|---|---|---|
| | | dev-clean | dev-other | test-clean | test-other |
| 0x | greedy | 1.86 | 4.10 | 2.24 | 4.56 |
| | DSR | 1.63 | 3.80 | 1.95 | 4.16 |
| | DLM-sum | 1.64 | 3.72 | 1.98 | 4.17 |
| 1x | greedy | 1.74 | 3.97 | 2.02 | 4.44 |
| | DSR | 1.62 | 3.74 | 1.90 | 4.11 |
| | DLM-sum | 1.53 | 3.67 | 1.83 | 3.98 |
| 10x | greedy | 1.82 | 3.98 | 2.05 | 4.49 |
| | DSR | 1.56 | 3.78 | 1.89 | 4.08 |
| | DLM-sum | 1.55 | 3.67 | 1.84 | 4.00 |
| 20x | greedy | 1.77 | 4.01 | 1.99 | 4.41 |
| | DSR | 1.59 | 3.77 | 1.91 | 4.08 |
| | DLM-sum | 1.53 | 3.70 | 1.85 | 3.98 |
| 40x | greedy | 1.82 | 4.02 | 2.07 | 4.52 |
| | DSR | 1.59 | 3.76 | 1.87 | 4.05 |
| | DLM-sum | 1.56 | 3.71 | 1.85 | 3.95 |

We generate DLM training data with increasing amounts of LibriSpeech ASR training data, and leave the amount of synthetic TTS data constant. Therefore in this setup the total amount of data seen during training (and total training time) increases with the amount of ASR data. LibriSpeech ASR training data has approximately 50M characters, while the LM corpus has approximately 4.3B characters. Therefore with a 40x multiplier on the LibriSpeech ASR data, we have a 1:2 proportion of real to synthetic data. Our results are shown in Table E.21.

There may be a slight improvement going from zero to one instance of LibriSpeech ASR training data, but the statistical significance is questionable. Surprisingly, further increases in the proportion of LibriSpeech ASR training data do not seem to meaningfully change the performance of the DLM, and the results are almost indistinguishable from the random seed baseline in Table E.38. We conclude that the DLM may benefit from more variation through the use of additional training data, but seeing a higher ratio of data from real audio is not important for DLM training.

### E.2.16  ONLY LIBRISPEECH ASR TRAINING DATA

A significant portion of this work is dedicated to generating training data for the DLM from the text-only LibriSpeech LM corpus by using TTS and various data augmentation techniques. Naturally one may wonder whether the additional data is needed, given that our non-TTS ASR model already performs quite well even though the training dataset is much smaller. We therefore generate a DLM training dataset using only the LibriSpeech ASR training data and no TTS data at all. For every epoch we duplicate LibriSpeech ASR data 85x to match the amount of data seen during typical DLM training. We also run this experiment with `stdPerturb` data augmentation configuration, and each of the 85 duplicates is generated with a different random seed to increase diversity. The results are shown in Table E.22.

Both DLMs overfit to the training data, which is not surprising given the small effective size of the training dataset. The training loss plot can be seen in Figure E.1. What is suprising though, is that the overfitting effect is much more pronounced in the DLM trained with the `stdPerturb` config-uration, which has more data augmentation. In the DLM with `stdPerturb` data augmentation we see that the tuned scales for DSR and DLM-sum decoding have a very low value for $\lambda_{\text{DLM}}$, mostly ignoring the DLM output.

We conclude that the LibriSpeech ASR 960h training dataset alone is not sufficient for DLM training, even with data augmentation.

Table E.22: DLMs trained with only LibriSpeech ASR data (no TTS data) and without additional data augmentation. The first model is a baseline with only TTS audio (no real audio). The second model is trained with only real audio. The third model has data augmentation configuration `stdPerturb` applied to real audio. All models trained for 5 epochs.

| Only LibriSpeech ASR data | Decoding | DLM Performance: WER [%] | | | |
|---|---|---|---|---|---|
| | | dev-clean | dev-other | test-clean | test-other |
| only TTS audio | greedy | 1.86 | 4.10 | 2.24 | 4.56 |
| | DSR | 1.63 | 3.80 | 1.95 | 4.16 |
| | DLM-sum | 1.64 | 3.72 | 1.98 | 4.17 |
| only real audio | greedy | 4.19 | 4.84 | 3.11 | 5.22 |
| | DSR | 1.77 | 4.12 | 2.09 | 4.46 |
| | DLM-sum | 2.56 | 4.36 | 2.53 | 5.11 |
| only real audio (stdPerturb) | greedy | 94.02 | 97.80 | 93.69 | 97.53 |
| | DSR | 1.72 | 3.96 | 1.95 | 4.30 |
| | DLM-sum | 1.78 | 3.98 | 1.99 | 4.37 |

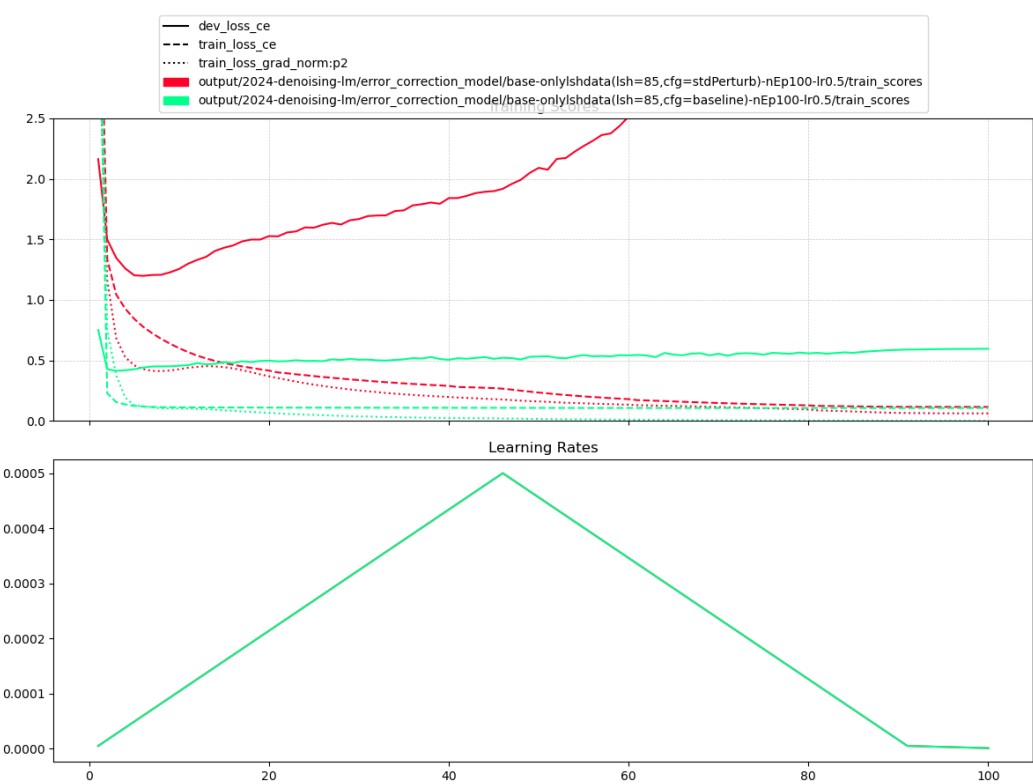

Figure E.1: Only LibriSpeech ASR training plots. A DLM epoch corresponds to 20 sub-epochs on the x-axis.

### E.2.17 RELEVANCE OF TTS-ASR DATA

We can make use of the Librispeech text-only corpus but still avoid the TTS model by generating training data via heuristic error generation methods. Previous work (Hrinchuk et al., 2020; Dutta et al., 2022; Ma et al., 2023a) used BERT, BART (Lewis et al., 2020) or T5 (Raffel et al., 2020) pretraining objectives, where text is corrupted by masking or random token substitution. Gu et al. (2024) also experimented with such heuristics, although no numbers were reported. Some preliminary results for this approach are shown in Table E.23. For the heuristics, here we use random token substitution with a rate that it uniformly sampled between 10% and 50% per sequence. The DLM

Table E.23: DLM performance when trained on data generated via TTS-ASR versus heuristic error generation. The DLM trained on errors via such heuristics is a standard error correction model baseline.

| Model | DLM Training Data Generation Method | WER [%] | | | |
|---|---|---|---|---|---|
| | | dev-clean | dev-other | test-clean | test-other |
| ASR only | - | 2.29 | 5.02 | 2.42 | 5.33 |
| ASR + LM | | 1.85 | 3.93 | 2.00 | 4.27 |
| ASR + DLM | TTS-ASR | 1.68 | 3.70 | 1.83 | 4.15 |
| | Heuristics | 2.11 | 4.60 | 2.28 | 5.06 |

trained on these data is significantly worse, and also worse than the standard LM. This is consistent to findings from Gu et al. (2024). However, we note that the error patterns can be improved a lot, and this can be finetuned or mixed with real ASR hypotheses, as done in previous work (Dutta et al., 2022; Ma et al., 2023a).

### E.3    ANALYSIS OF INFERENCE AND MODEL BEHAVIOR

#### E.3.1    DECODING METHODS

A comparison of our different decoding methods (DLM greedy, DSR decoding, DLM-sum) is shown in Tables 1, E.1 and E.2. We note that the DLM greedy WER is sometimes even worse than the ASR baseline (without LM). This is different to Gu et al. (2024), where the DLM greedy decoding clearly outperforms the ASR baseline. We assume that the different vocabulary (subwords vs. characters) is an important contributing factor to this difference (Appendix G.2). The DSR decoding method typically already outperforms a standard LM in both rescoring and first-pass decoding, and the DLM-sum decoding consistently achieves the best performance, surpassing all other methods.

A grid search over different $\lambda_{\text{DLM}}$, $\lambda_{\text{prior}}$ scales for DSR and DLM-sum rescoring is shown in Figures E.2 and E.3. ASR and DLM n-best lists are generated with scores once, and grid search is performed offline. We see that the optimal prior scale is quite low, and with no prior, we get only small degradations in WER. We provide a scale tuning plot for the n32-d1024 LM from Table D.2 in Figure E.4 for comparison.

We evaluate different beam sizes for their impact on the WER for all three decoding methods. The DLM is trained with the stdPerturb data augmentation configuration for 10 epochs. The method "beam search" refers to DLM greedy, but with beam size $k > 1$. DSR w/o concat is rescoring only DLM hypotheses, DSR with concat uses $\frac{\text{beam\_size}}{2}$ DLM beam size, $\frac{\text{beam\_size}}{2}$ ASR beam size and combines both beams for rescoring. DLM rescore is only rescoring ASR hypotheses. Results for these methods are shown in Figure E.5. We see that "beam search" slowly approaches ASR baseline performance from above, and is thus not very useful. DSR without concat and DLM rescore converge to a similar WER, while DLM rescore is slightly better for smaller beam sizes. DSR with concat is the best method for all beam sizes.

We plot the performance of DLM-sum decoding in Figure E.6 for different beam sizes and number of ASR hypotheses. Both increasing the beam size and number of ASR hypotheses helps. With one ASR hypothesis we roughly match DSR decoding performance with concatenation, and surpass that with a higher number of ASR hypotheses. For beam size 1, scale tuning is impossible, so we re-use scales from DSR decoding with concatenation, beam size 64 and recover usable results. DSR decoding defaults to only ASR scoring for beam size 1.

#### E.3.2    SEARCH AND MODEL ERRORS

Due to the autoregressive nature of our decoding methods, the most probable output sequence is not always found. This happens when a correct label is discarded at an early step in the decoding process due to beam size pruning, but this label would lead to a better overall sequence if it were included. We only count search errors where the ground truth sequence is not selected but had a

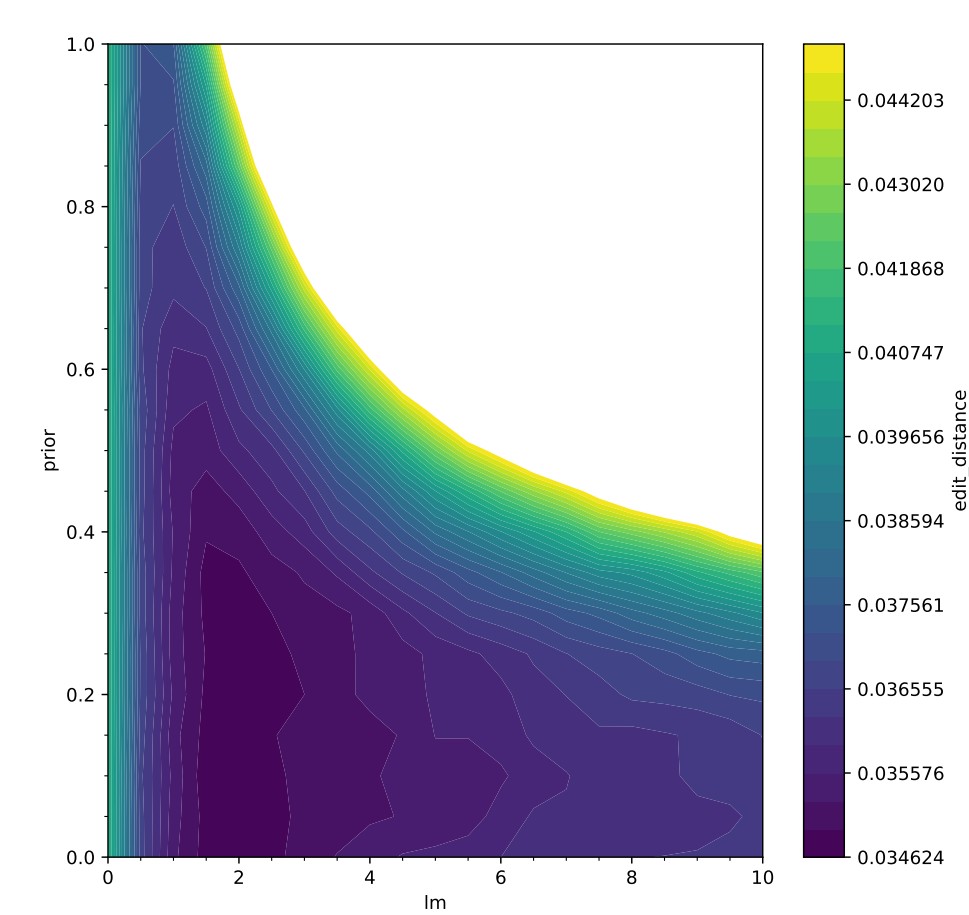

Figure E.2: Plot of different scales for DLM and prior for **DSR decoding** on dev-other. The DLM is trained on `stdPerturb` data augmentation for 10 epochs. The prior scale is relative to the DLM scale, i.e. when the DLM scale is 0.1 and the prior scale is 2.0, the actual prior scale is 0.2. ASR score scale is kept at a constant 1.0.

higher predicted probability than the output sequence found during decoding[13]. In contrast to that, model errors are the percentage of sentences where the ground truth sequence is not the most probable sequence according to the model. We also calculate the Oracle WER by picking the hypothesis with the lowest WER from the beam. Results are shown in Table E.26. Counted search errors are $\leq 1\%$ across the board, while model error rates are significantly higher. This suggests that our search methods are quite effective at finding the most probable output sequence according to the model, and that the main limitation of our decoding methods is the model itself. For comparison we also give results for LM rescoring, where the DLM rescores only the ASR n-best list. Through the Oracle WER we see that the DLM is limited by the quality of the ASR hypotheses in LM rescoring, and that integration of DLM hypotheses into the decoding process as in DSR decoding is essential for further improvements over the ASR model.

---

[13]This does not count all search errors. We cannot count all search errors, as we never know what would be the highest possible probability for some given input.

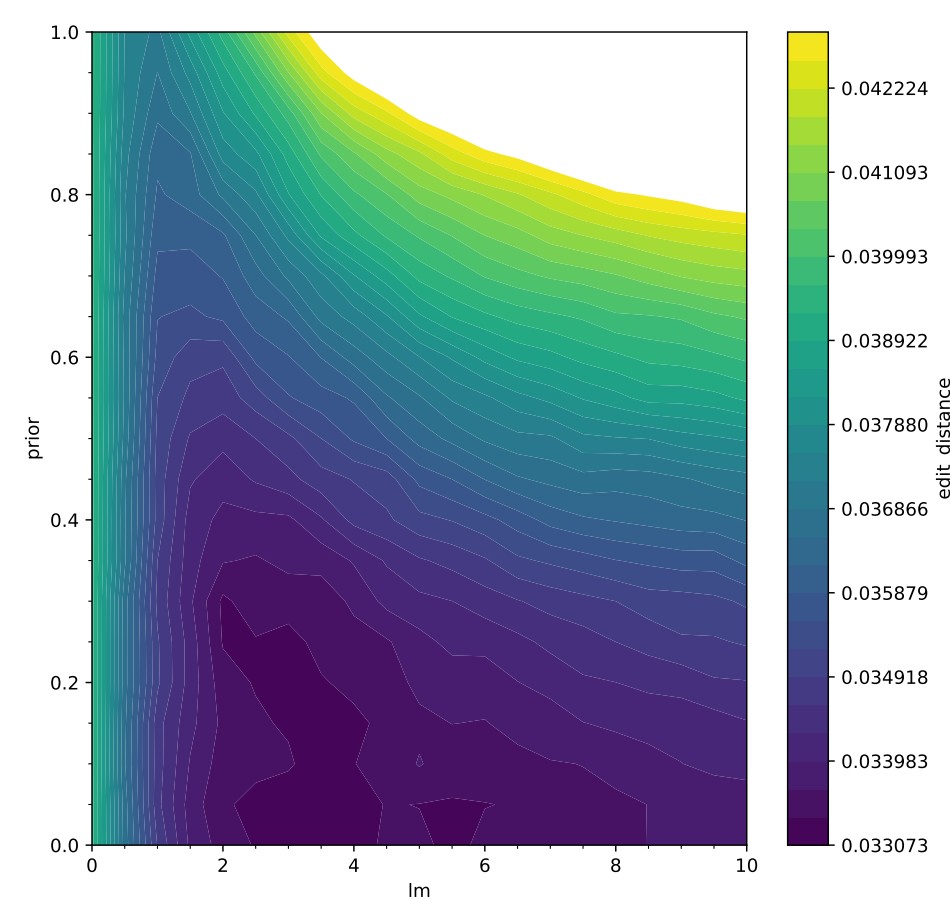

Figure E.3: Plot of different scales for DLM and prior for **DLM-sum decoding** on dev-other. The DLM is trained on `stdPerturb` data augmentation for 10 epochs. The prior scale is relative to the DLM scale, i.e. when the DLM scale is 0.1 and the prior scale is 2.0, the actual prior scale is 0.2. ASR score scale is kept at a constant 1.0.

### E.3.3 EVALUATION OVER THE COURSE OF TRAINING

We evaluate the DLM performance over the course of training. During DLM training, we keep checkpoints at epochs 1, 2, 4, 8 and 10. At every checkpoint, we run scale tuning as described in Section 2.1 and evaluate the WER on test-other. Here we look at a DLM trained with the `stdPerturb` data augmentation configuration for a total of 10 epochs. The results are shown in Figure E.8.

Greedy decoding improvement looks somewhat unstable, while DSR and DLM-sum decoding steadily approach their final WER. The unstable trend of greedy decoding could be explained by our findings in Section 4.3.

### E.3.4 WER DISTRIBUTION

Different data augmentation techniques lead to different WER of the training data, but the WER of the dataset alone does not reveal any information about the variance or distribution of error rates on the sentence-level. Here we look at the WER of individual sentences in the training data, and their length.

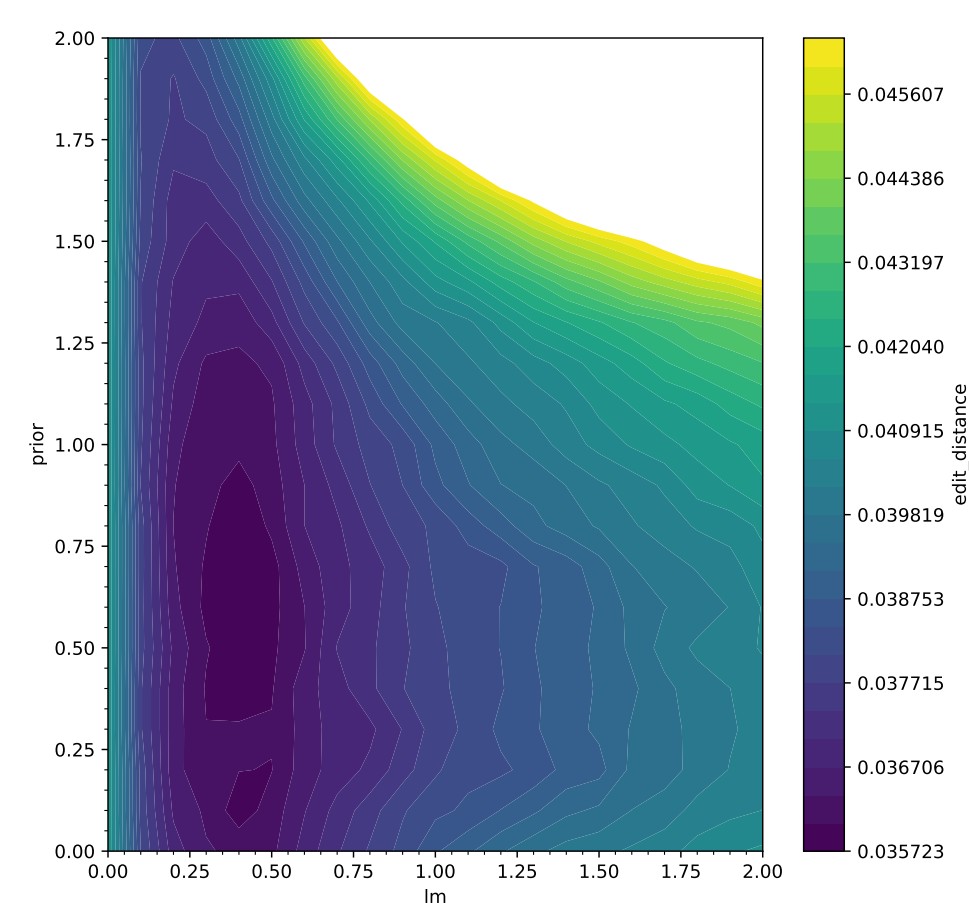

Figure E.4: Plot of different scales for LM and prior for **LM rescoring** on dev-other. The LM is n32-d1024 trained for 5 epochs from Table D.2. The prior scale is relative to the LM scale, i.e. when the LM scale is 0.1 and the prior scale is 2.0, the actual prior scale is 0.2. ASR score scale is kept at a constant 1.0.

Figure E.9 shows the distribution of WER among sentences in the training data for no data augmentation, and with stdPerturb data augmentation. Note that the y-axis is logarithmic to better visualize the distribution, and each sentence is weighted by its reference length similar to how WER is calculated over a dataset. We see that the shape of the distribution changes significantly, from a monotonic decay in the baseline case to a more hill-like shape with a peak around 10% WER with stdPerturb. All three datasets seem to behave similarly, though the dev-other set is biased toward slightly higher WER, which is expected as it is a more difficult validation set.

Figure E.10 shows the distribution of sentence lengths in the training data. The two data augmentation configurations seem to have a very similar length distribution, with an average of around 17-20 words per sentence.

We show WER and length distributions for all combined data augmentation techniques in Figures E.11 to E.19.

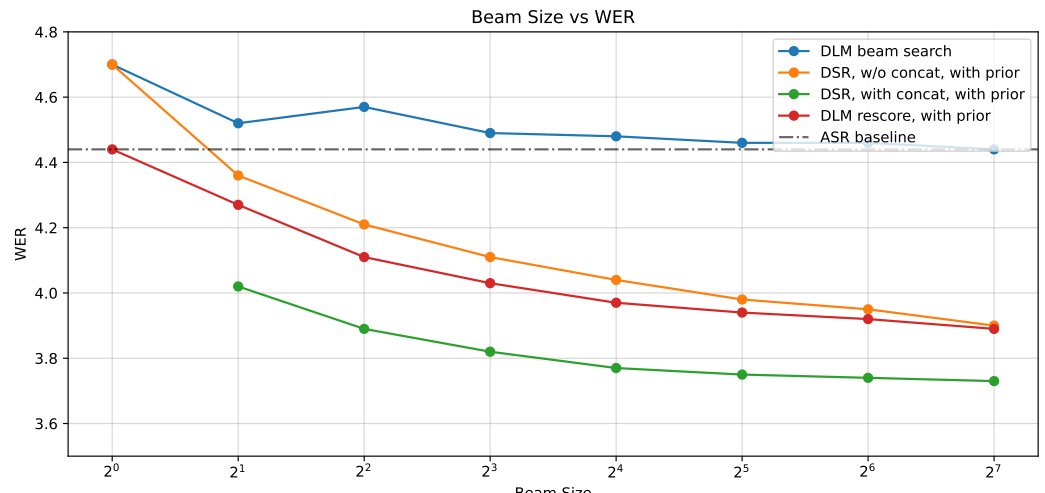

Figure E.5: Performance of DLM greedy and DSR decoding over different beam sizes on test-other. DSR with concat uses $\frac{\text{beam\_size}}{2}$ for DLM and ASR beams and combines both beams for rescoring. DLM is trained on `stdPerturb` data augmentation for 10 epochs. Raw data can be found in Table E.24.

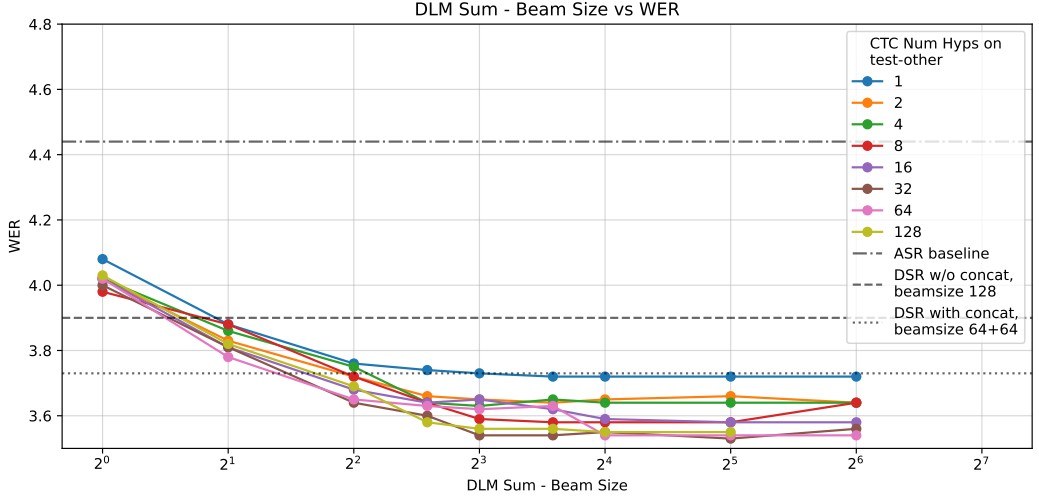

Figure E.6: Performance of DLM-sum decoding over different beam sizes and different number of hypotheses from ASR model. DLM is trained on `stdPerturb` data augmentation for 10 epochs. Same y and x axis as Figure E.5. Raw data can be found in Table E.25.

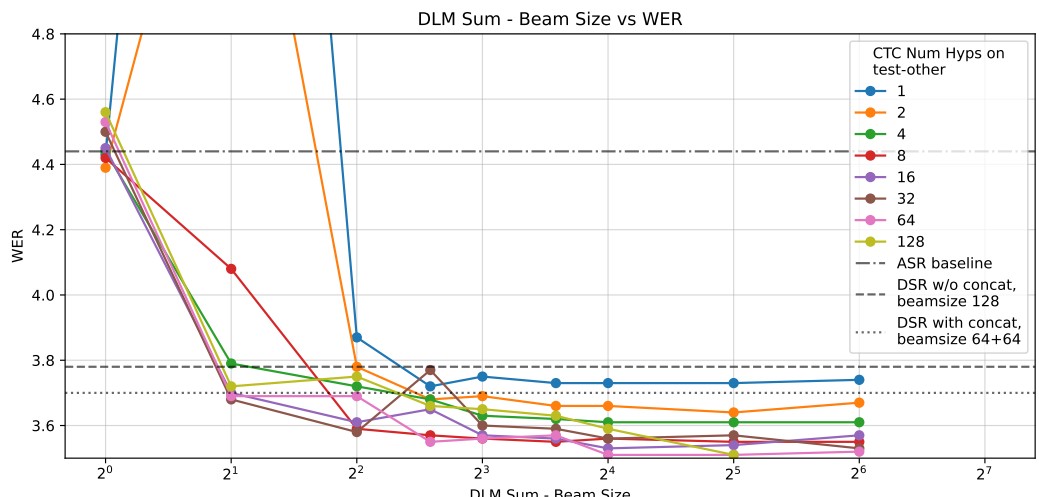

Figure E.7: Performance of DLM-sum decoding over different beam sizes and different number of hypotheses from ASR model. DLM is trained on `low` data augmentation for 10 epochs. Same y and x axis as Figure E.5 and Figure E.6. Performance breaks down for beam size 2 due to poor scale tuning.

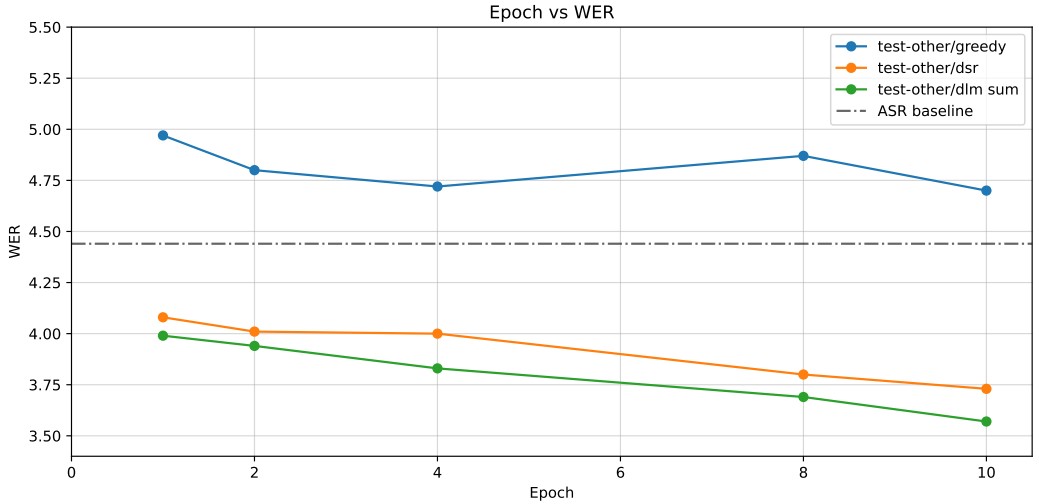

Figure E.8: Performance changes over the course of Standard DLM training. Shown are different checkpoints of the same DLM training run. The DLM is trained with the `stdPerturb` data augmentation configuration for 10 epochs.

Table E.24: Decoding Test: Beam Size.

| Decoding Test: Beam Size | Decoding | DLM Performance: WER [%] | | | |
|---|---|---|---|---|---|
| | | dev-clean | dev-other | test-clean | test-other |
| 1 | greedy | 2.26 | 3.96 | 2.47 | 4.70 |
| | DSR | 2.26 | 3.96 | 2.47 | 4.70 |
| 2 | greedy | 2.22 | 3.90 | 2.28 | 4.52 |
| | DSR | 2.04 | 3.74 | 2.13 | 4.36 |
| 4 | greedy | 2.14 | 3.89 | 2.28 | 4.57 |
| | DSR | 1.87 | 3.67 | 2.08 | 4.21 |
| 8 | greedy | 2.09 | 3.87 | 2.21 | 4.49 |
| | DSR | 1.75 | 3.59 | 1.93 | 4.11 |
| 16 | greedy | 2.09 | 3.87 | 2.21 | 4.48 |
| | DSR | 1.73 | 3.54 | 1.91 | 4.04 |
| 32 | greedy | 2.09 | 3.87 | 2.21 | 4.46 |
| | DSR | 1.71 | 3.50 | 1.88 | 3.98 |
| 64 | greedy | 2.09 | 3.87 | 2.21 | 4.46 |
| | DSR | 1.69 | 3.50 | 1.85 | 3.95 |
| 128 | greedy | 2.09 | 3.86 | 2.21 | 4.44 |
| | DSR | 1.65 | 3.47 | 1.86 | 3.90 |
| 1 (with concat 1 ASR hyps) | greedy | 2.07 | 3.94 | 2.39 | 4.56 |
| | DSR | 1.57 | 3.70 | 1.85 | 4.02 |
| 2 (with concat 2 ASR hyps) | greedy | 2.21 | 3.90 | 2.23 | 4.44 |
| | DSR | 1.51 | 3.59 | 1.88 | 3.89 |
| 4 (with concat 4 ASR hyps) | greedy | 2.13 | 3.88 | 2.22 | 4.47 |
| | DSR | 1.49 | 3.55 | 1.77 | 3.82 |
| 8 (with concat 8 ASR hyps) | greedy | 2.09 | 3.87 | 2.21 | 4.49 |
| | DSR | 1.48 | 3.50 | 1.76 | 3.77 |
| 16 (with concat 16 ASR hyps) | greedy | 2.09 | 3.86 | 2.21 | 4.47 |
| | DSR | 1.49 | 3.49 | 1.76 | 3.75 |
| 32 (with concat 32 ASR hyps) | greedy | 2.09 | 3.86 | 2.21 | 4.46 |
| | DSR | 1.51 | 3.46 | 1.79 | 3.74 |
| 64 (with concat 64 ASR hyps) | greedy | 2.09 | 3.86 | 2.21 | 4.46 |
| | DSR | 1.51 | 3.46 | 1.78 | 3.73 |

Table E.25: Decoding test: DLM-sum beam size.

| DLM-sum beam size | ASR num_hyps | DLM Performance: WER [%] | | | |
|---|---|---|---|---|---|
| | | dev-clean | dev-other | test-clean | test-other |
| 2 | 1 | 1.53 | 3.48 | 1.86 | 3.88 |
| | 2 | 1.60 | 3.46 | 1.85 | 3.83 |
| | 4 | 1.67 | 3.46 | 1.89 | 3.86 |
| | 8 | 1.70 | 3.45 | 1.94 | 3.88 |
| | 16 | 1.65 | 3.39 | 1.86 | 3.81 |
| | 32 | 1.68 | 3.40 | 1.88 | 3.81 |
| 4 | 1 | 1.50 | 3.49 | 1.84 | 3.76 |
| | 2 | 1.52 | 3.44 | 1.83 | 3.72 |
| | 4 | 1.53 | 3.40 | 1.80 | 3.75 |
| | 8 | 1.50 | 3.36 | 1.81 | 3.72 |
| | 16 | 1.49 | 3.34 | 1.80 | 3.68 |
| | 32 | 1.51 | 3.33 | 1.79 | 3.64 |
| 6 | 1 | 1.50 | 3.50 | 1.84 | 3.74 |
| | 2 | 1.53 | 3.40 | 1.84 | 3.66 |
| | 4 | 1.51 | 3.38 | 1.82 | 3.64 |
| | 8 | 1.51 | 3.36 | 1.82 | 3.64 |
| | 16 | 1.51 | 3.34 | 1.80 | 3.64 |
| | 32 | 1.51 | 3.35 | 1.77 | 3.60 |
| 8 | 1 | 1.51 | 3.48 | 1.84 | 3.73 |
| | 2 | 1.52 | 3.40 | 1.84 | 3.65 |
| | 4 | 1.52 | 3.38 | 1.82 | 3.63 |
| | 8 | 1.50 | 3.38 | 1.80 | 3.59 |
| | 16 | 1.52 | 3.35 | 1.80 | 3.65 |
| | 32 | 1.47 | 3.33 | 1.78 | 3.54 |
| 12 | 1 | 1.51 | 3.47 | 1.84 | 3.72 |
| | 2 | 1.53 | 3.41 | 1.83 | 3.64 |
| | 4 | 1.51 | 3.38 | 1.81 | 3.65 |
| | 8 | 1.50 | 3.35 | 1.79 | 3.58 |
| | 16 | 1.48 | 3.34 | 1.79 | 3.62 |
| | 32 | 1.51 | 3.31 | 1.80 | 3.54 |
| 16 | 1 | 1.51 | 3.47 | 1.84 | 3.72 |
| | 2 | 1.54 | 3.39 | 1.83 | 3.65 |
| | 4 | 1.51 | 3.38 | 1.80 | 3.64 |
| | 8 | 1.51 | 3.35 | 1.79 | 3.58 |
| | 16 | 1.51 | 3.32 | 1.80 | 3.59 |
| | 32 | 1.49 | 3.31 | 1.79 | 3.55 |
| 32 | 1 | 1.52 | 3.47 | 1.83 | 3.72 |
| | 2 | 1.52 | 3.39 | 1.83 | 3.66 |
| | 4 | 1.51 | 3.38 | 1.80 | 3.64 |
| | 8 | 1.50 | 3.35 | 1.79 | 3.58 |
| | 16 | 1.50 | 3.31 | 1.80 | 3.58 |
| | 32 | 1.51 | 3.31 | 1.80 | 3.53 |

Table E.26: Search/Model errors and Oracle WER for different decoding methods. DLM trained on `stdPerturb` data augmentation configuration for 10 epochs. DLM rescore uses the ASR n-best list, DSR uses ASR and DLM beams. Respective beam sizes are shown in parentheses.

| Decoding | Search Error % | | | |
|---|---|---|---|---|
| | dev-clean | dev-other | test-clean | test-other |
| DLM greedy (1) | 0.89 | 0.77 | 0.69 | 0.88 |
| DLM beam search (64) | 0.07 | 0.14 | 0.00 | 0.07 |
| DLM rescore (64) | 0.37 | 0.84 | 0.27 | 0.88 |
| DSR (32+32) | 0.92 | 0.28 | 0.27 | 0.68 |

| Decoding | Model Error % | | | |
|---|---|---|---|---|
| | dev-clean | dev-other | test-clean | test-other |
| DLM greedy (1) | 24.20 | 36.17 | 25.95 | 39.61 |
| DLM beam search (64) | 24.60 | 36.45 | 26.49 | 40.12 |
| DLM rescore (64) | 20.87 | 34.29 | 22.52 | 36.54 |
| DSR (32+32) | 19.35 | 33.07 | 22.44 | 35.79 |

| Decoding | Oracle WER % | | | |
|---|---|---|---|---|
| | dev-clean | dev-other | test-clean | test-other |
| DLM greedy (1) | 2.26 | 3.96 | 2.47 | 4.70 |
| DLM beam search (64) | 0.66 | 1.73 | 0.71 | 1.95 |
| DLM rescore (64) | 0.73 | 2.17 | 0.89 | 2.17 |
| DSR (32+32) | 0.41 | 1.54 | 0.54 | 1.58 |

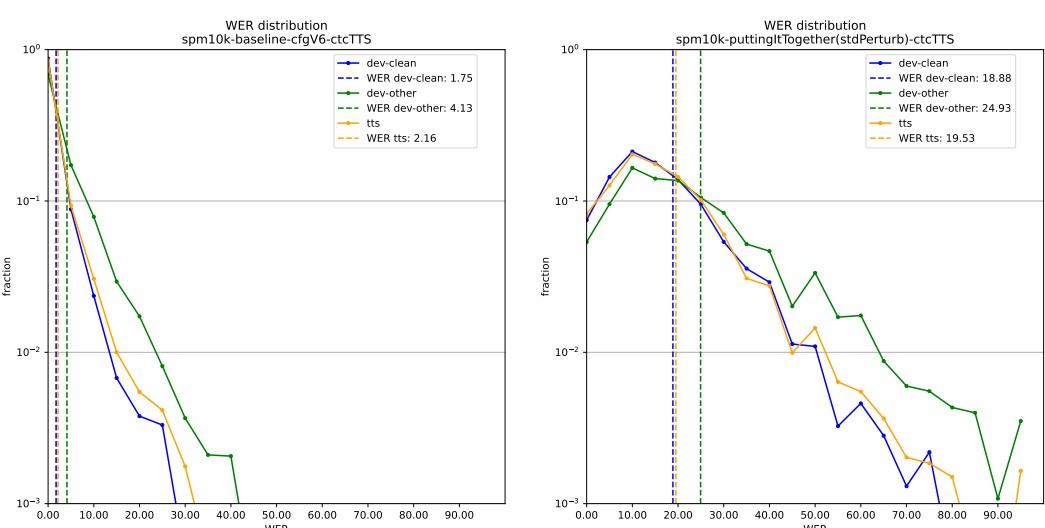

(a) Baseline training data without additional data augmentation.

(b) Training data with `stdPerturb` data augmentation.

Figure E.9: WER distribution of training data. Both generated with spm10k (TTS) ASR model.

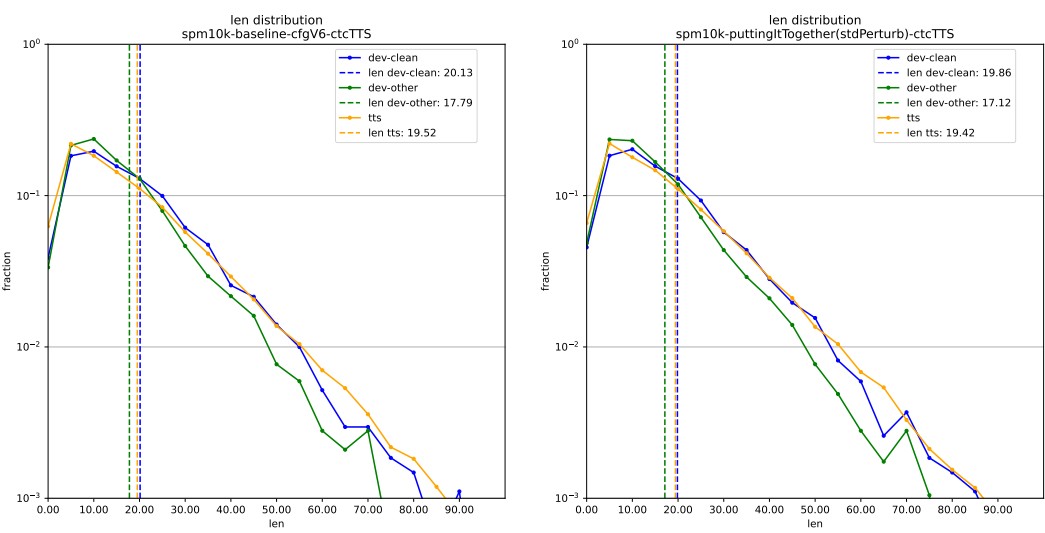

(a) Baseline training data without additional data augmentation.

(b) Training data with `stdPerturb` data augmentation.

Figure E.10: Length distribution of training data. Both generated with spm10k (TTS) ASR model.

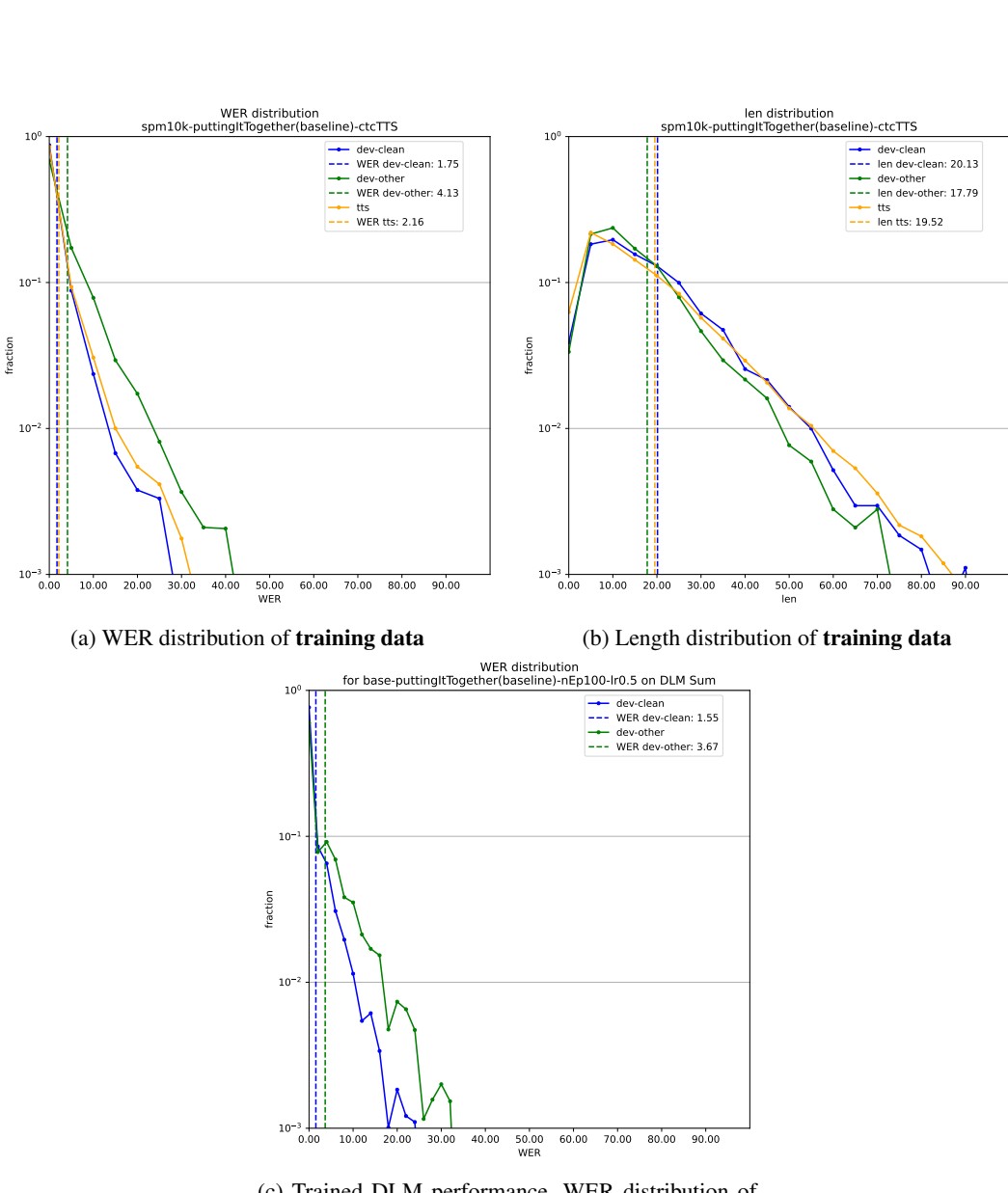

(a) WER distribution of **training data**

(b) Length distribution of **training data**

(c) Trained DLM performance, WER distribution of DLM-sum output **in recognition**

Figure E.11: **Baseline** data augmentation for DLM training. Top: length distribution and DLM-sum output WER distribution of **training data**, Bottom: WER distribution of DLM-sum output **in recognition**.

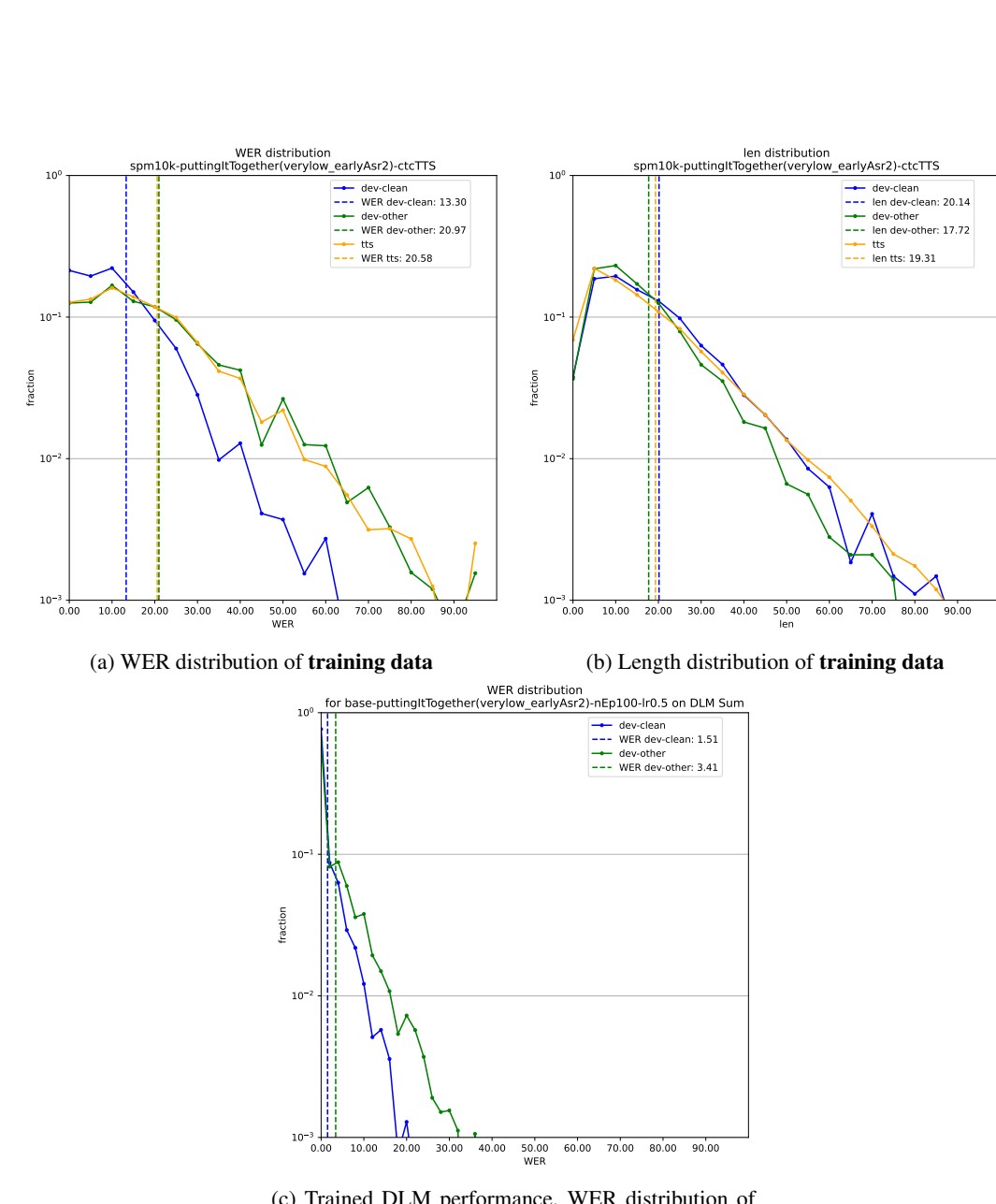

(a) WER distribution of **training data**

(b) Length distribution of **training data**

(c) Trained DLM performance, WER distribution of
DLM-sum output **in recognition**

Figure E.12: **Verylow-earlyAsr2** data augmentation for DLM training. Top: length distribution
and DLM-sum output WER distribution of **training data**, Bottom: WER distribution of DLM-sum
output **in recognition**.

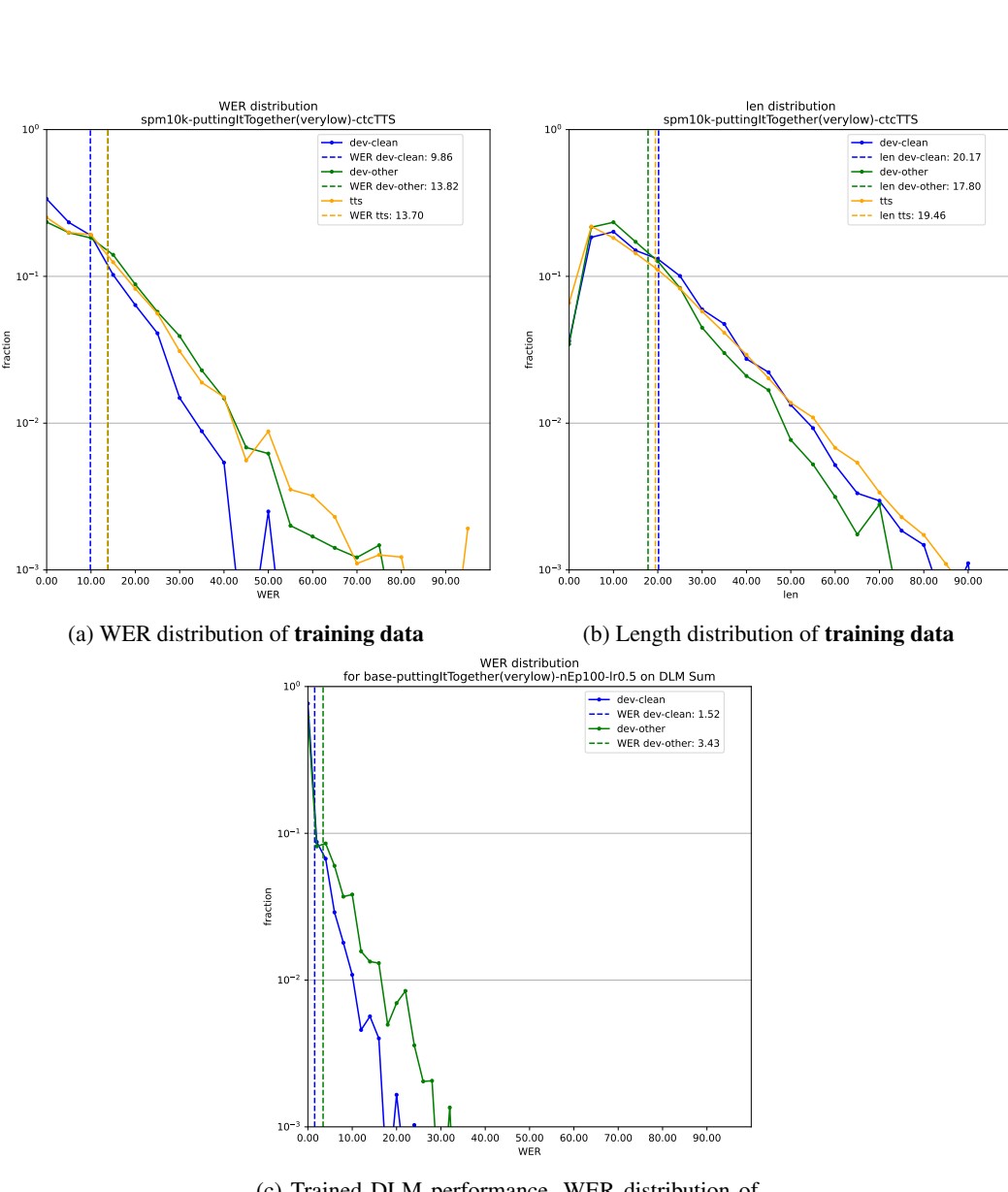

(a) WER distribution of **training data**

(b) Length distribution of **training data**

(c) Trained DLM performance, WER distribution of
DLM-sum output **in recognition**

Figure E.13: **Verylow** data augmentation for DLM training. Top: length distribution and DLM-sum output WER distribution of **training data**, Bottom: WER distribution of DLM-sum output **in recognition**.

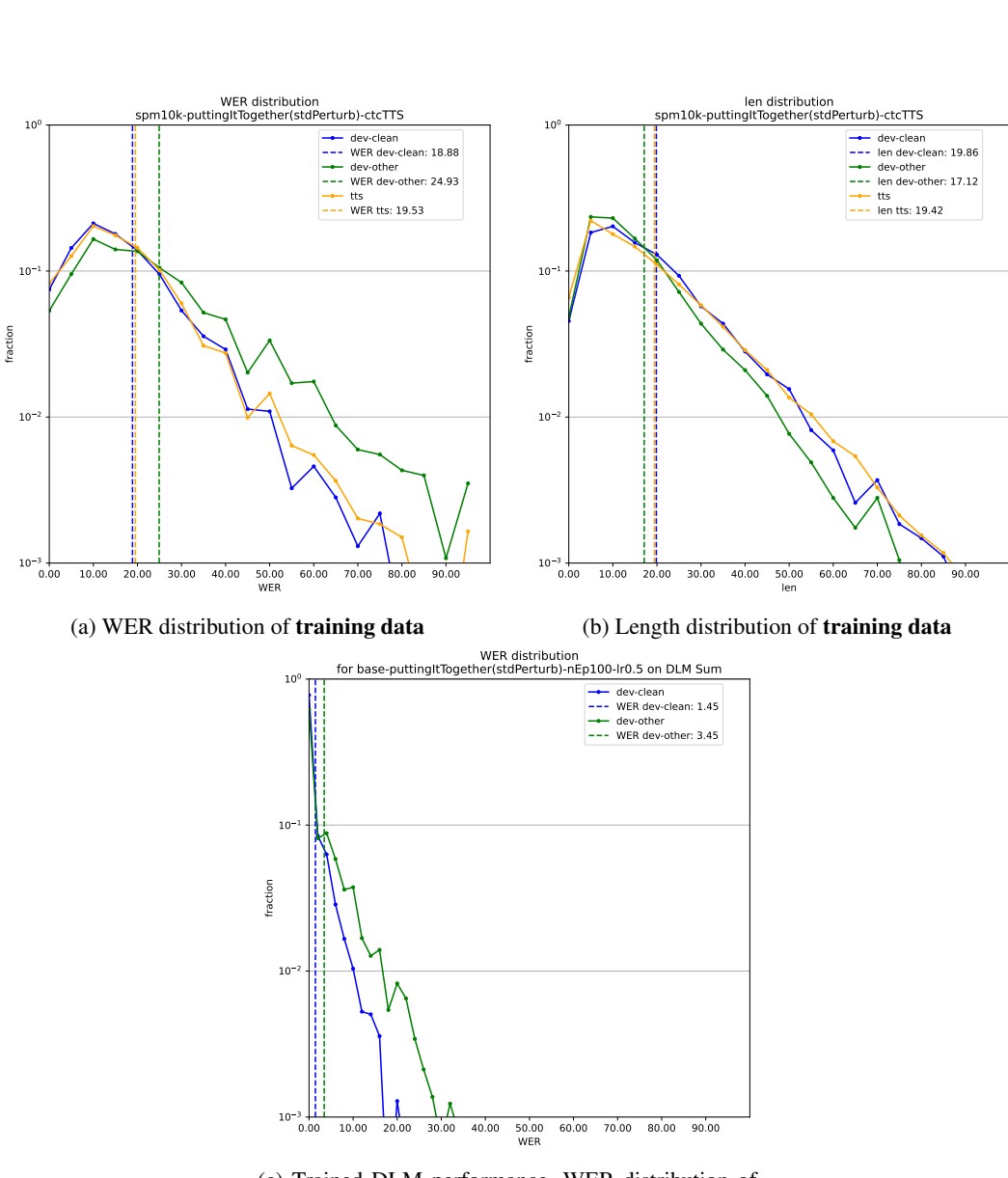

(a) WER distribution of **training data**

(b) Length distribution of **training data**

(c) Trained DLM performance, WER distribution of
DLM-sum output **in recognition**

Figure E.14: **stdPerturb** data augmentation for DLM training. Top: length distribution and DLM-sum output WER distribution of **training data**, Bottom: WER distribution of DLM-sum output **in recognition**.

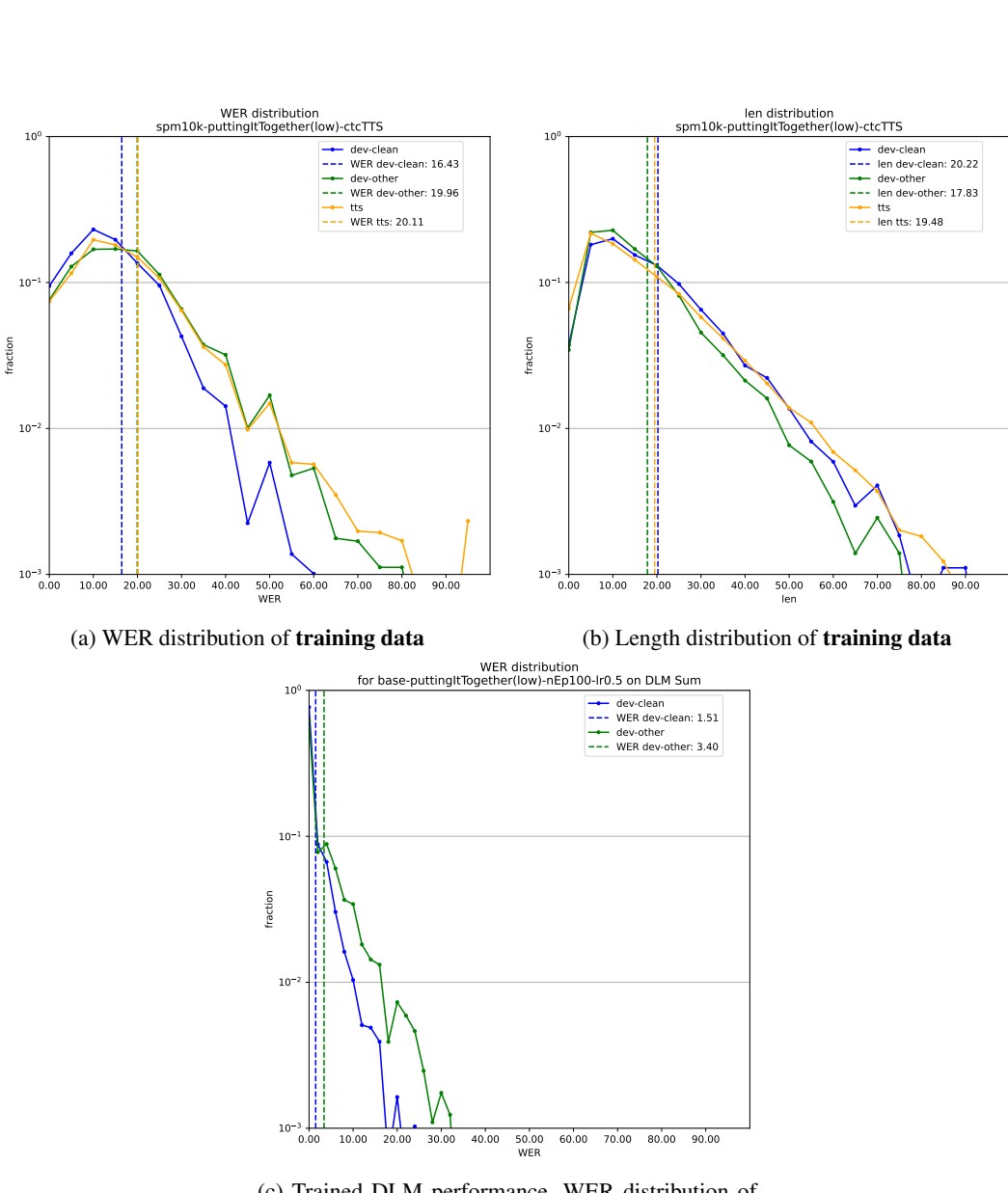

(a) WER distribution of **training data**

(b) Length distribution of **training data**

(c) Trained DLM performance, WER distribution of
DLM-sum output **in recognition**

Figure E.15: **Low** data augmentation for DLM training. Top: length distribution and DLM-sum output WER distribution of **training data**, Bottom: WER distribution of DLM-sum output **in recognition**.

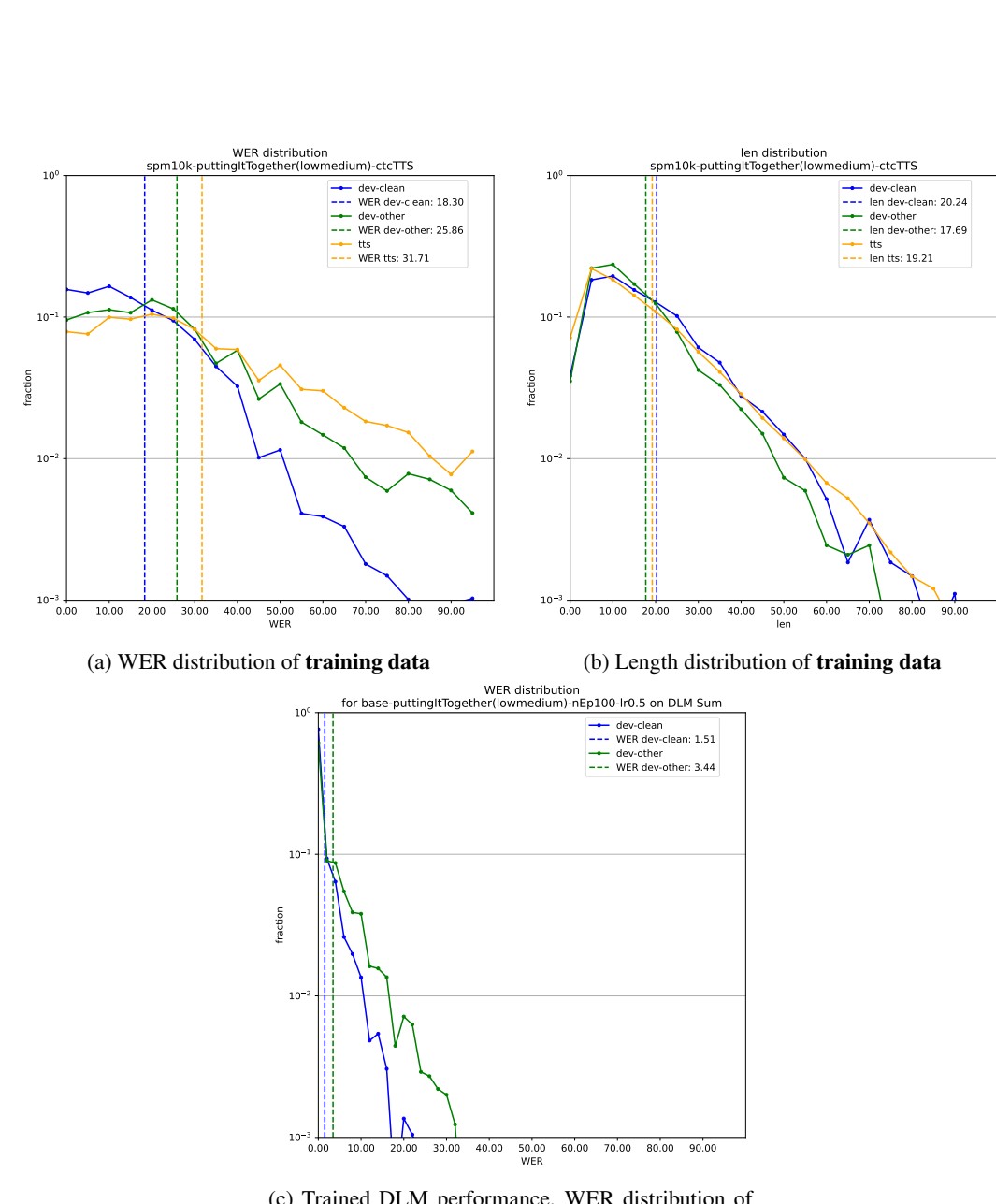

(a) WER distribution of **training data**

(b) Length distribution of **training data**

(c) Trained DLM performance, WER distribution of DLM-sum output **in recognition**

Figure E.16: **Lowmedium** data augmentation for DLM training. Top: length distribution and DLM-sum output WER distribution of **training data**, Bottom: WER distribution of DLM-sum output **in recognition**.

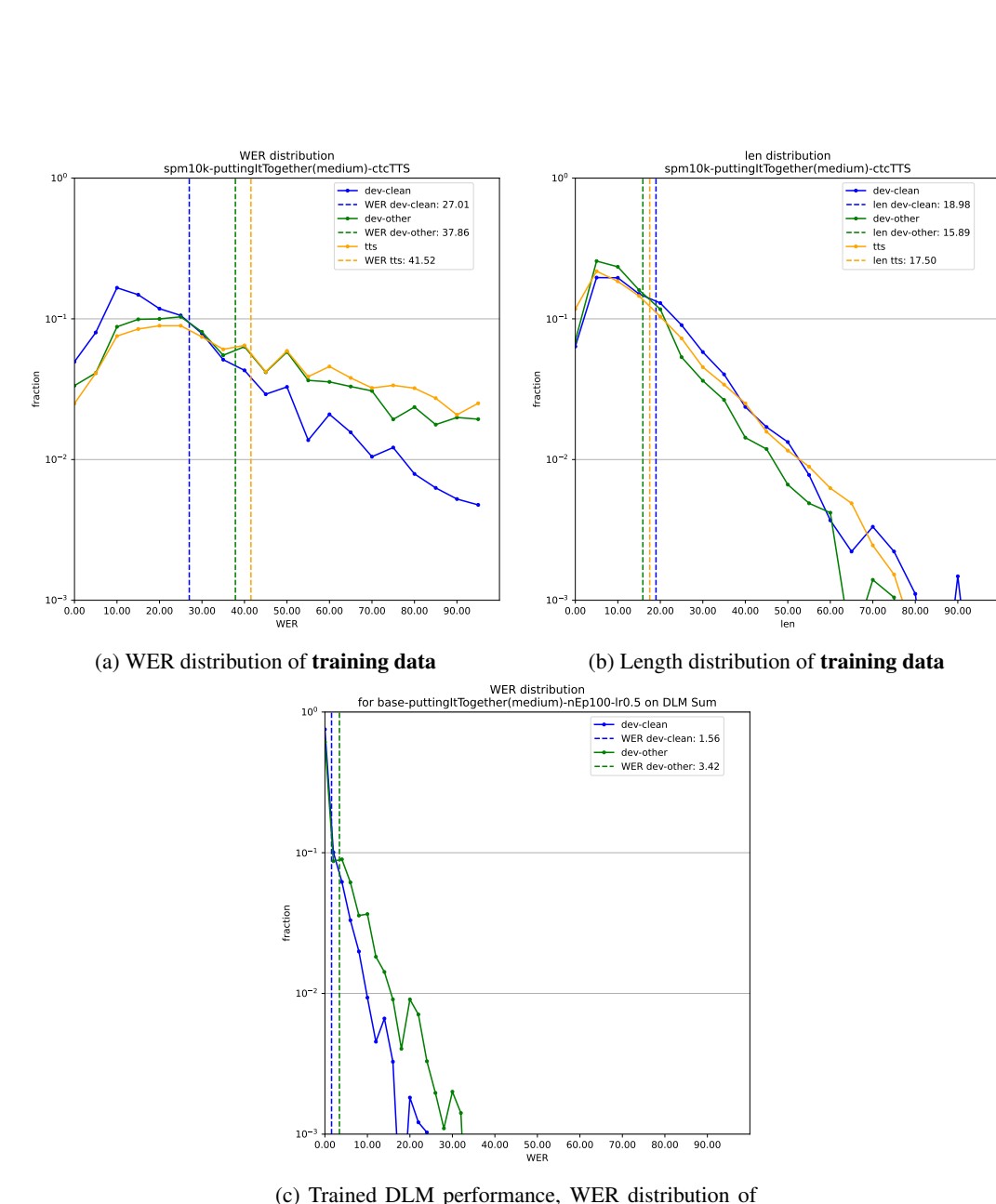

(a) WER distribution of **training data**

(b) Length distribution of **training data**

(c) Trained DLM performance, WER distribution of
DLM-sum output **in recognition**

Figure E.17: **Medium** data augmentation for DLM training. Top: length distribution and DLM-sum output WER distribution of **training data**, Bottom: WER distribution of DLM-sum output **in recognition**.

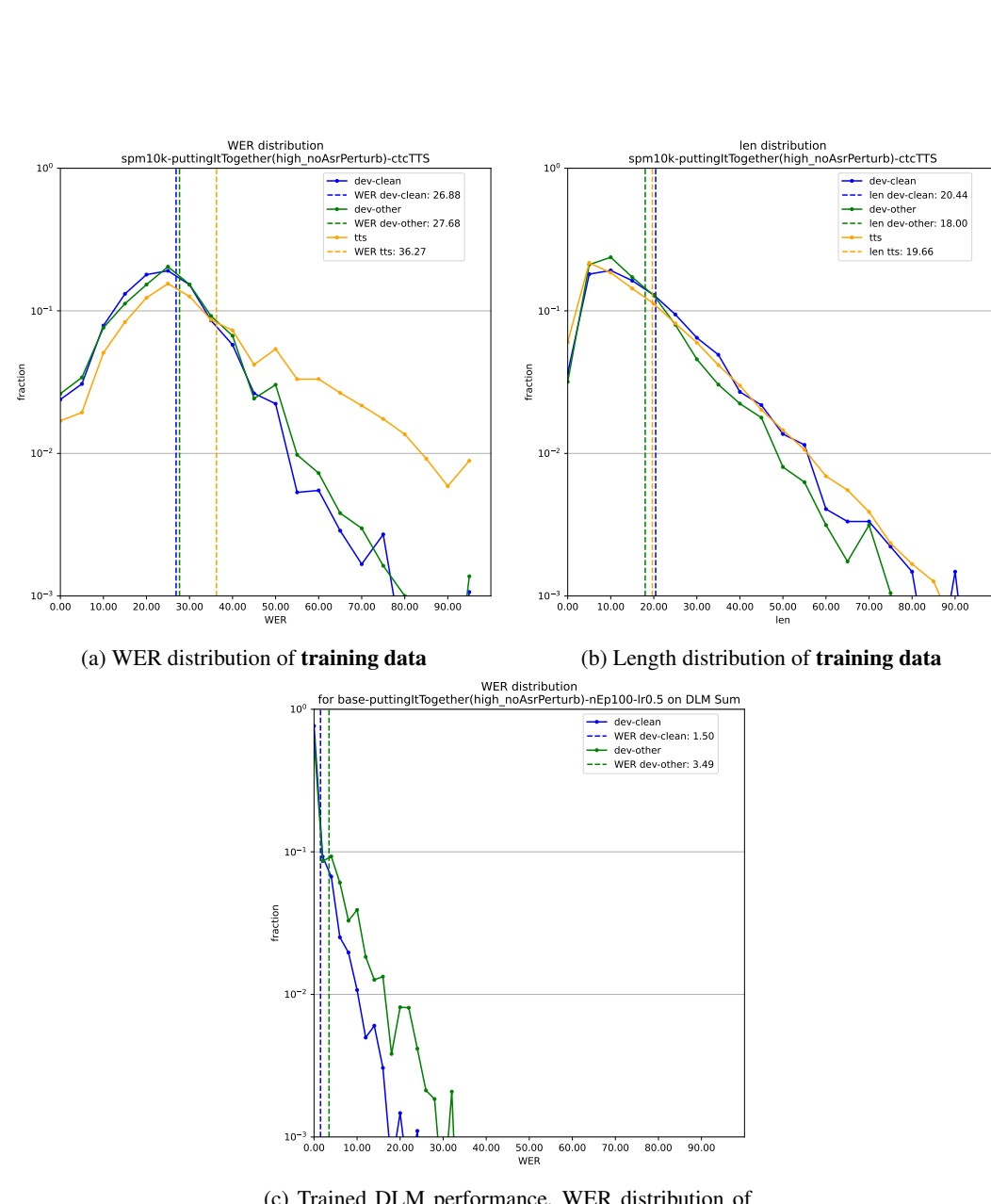

(a) WER distribution of **training data**

(b) Length distribution of **training data**

(c) Trained DLM performance, WER distribution of
DLM-sum output **in recognition**

Figure E.18: **High-noAsrPerturb** data augmentation for DLM training. Top: length distribution and DLM-sum output WER distribution of **training data**, Bottom: WER distribution of DLM-sum output **in recognition**.

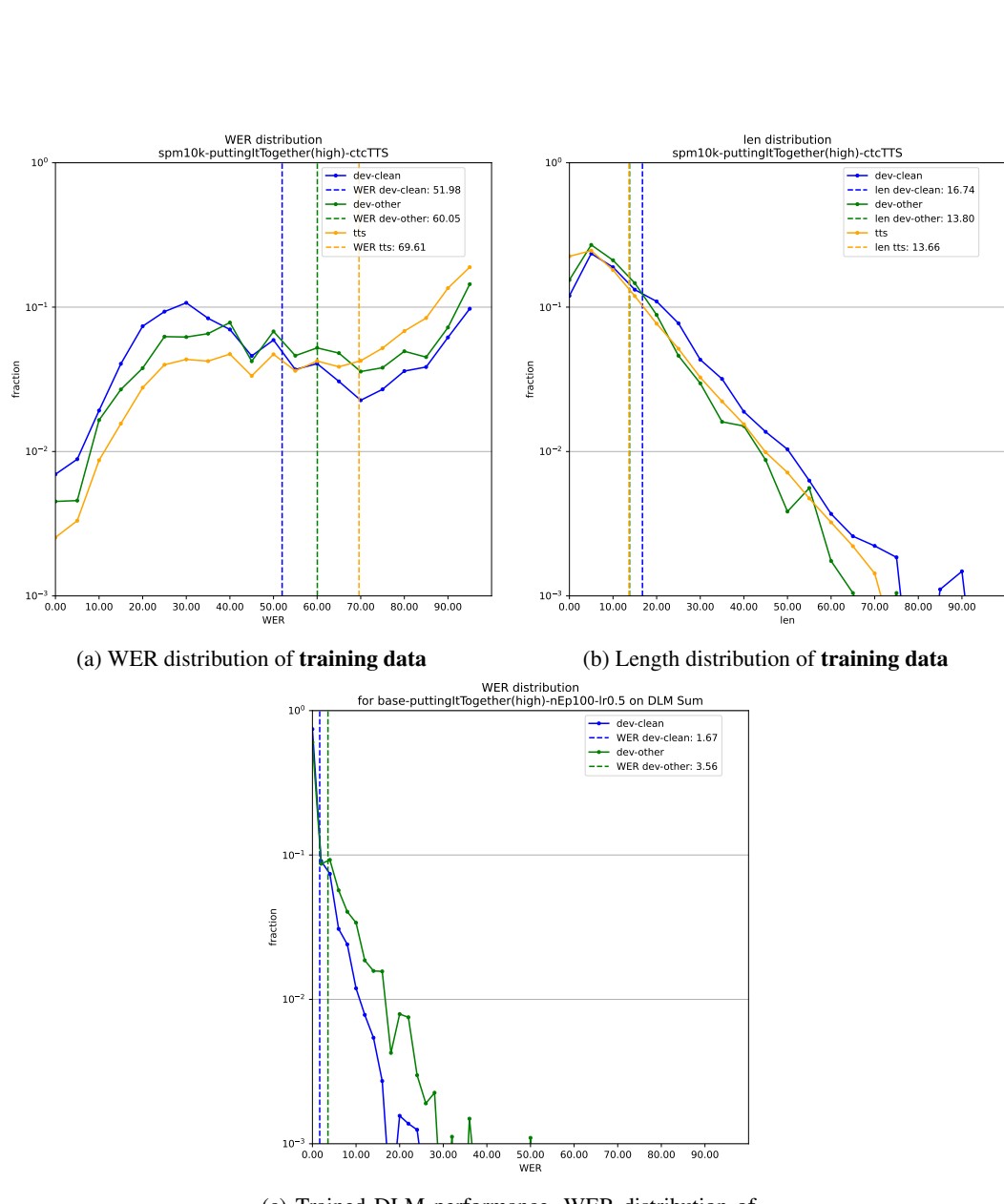

(a) WER distribution of **training data**

(b) Length distribution of **training data**

(c) Trained DLM performance, WER distribution of DLM-sum output **in recognition**

Figure E.19: **High** data augmentation for DLM training. Top: length distribution and DLM-sum output WER distribution of **training data**, Bottom: WER distribution of DLM-sum output **in recognition**.

### E.3.5 CORRELATIONS

Our experiments show that some data augmentation techniques lead to great improvements in DLM performance, while others have little effect. We want to understand why this is the case, and if there are any metrics that can predict the effectiveness of a data augmentation technique before a DLM is trained. To do this, we plot various metrics against each other to check for correlations, which may give us further insights into the effects of data augmentation. The data points consist of all the data augmentation techniques we have evaluated in this work, including the experiments where we combine multiple techniques. Due to space constraints, we only show the data augmentation category each point corresponds to, but not its exact parameters. To make these experiments feasible, we only compute and aggregate these metrics over a subset of about 52k sentences from the TTS-generated training data, instead of the full dataset of about 40M sentences.

Before we start with the training data correlations, we want to see how well the different decoding methods correlate with each other. We plot the different decoding methods against each other in Figure E.20.

It is apparent that there is hardly any correlation between greedy and DSR decoding, but DSR and DLM-sum decoding are highly correlated. There are even some outliers in plot (a) for DSR WER around 4.0% where the greedy WER varies between 4.48% and 6.53%. This is an indication that greedy decoding performance is not a good predictor for performance in model combination methods, as in DSR or DLM-sum decoding. Given that our best results are achieved with DSR and DLM-sum decoding, we will therefore not focus as much on greedy decoding but we still report it for completeness. We do not know the reason for the bad correlation between greedy and DSR, and we cannot rule out that this is not caused by a bug in our implementation[14]. A possible explanation is that with more data augmentation the DLM learns to become less peaky or confident in its predictions, which may improve model combination performance but hurt DLM greedy decoding. We explore this possibility in Section 4.3 by artificially adjusting the peakiness of the DLM output distribution through softmax temperature scaling. Another explanation can be given by the presence of unpredictable DLM hallucinations, which we show in Section 4.3.

We plot the WER of the TTS-generated hypotheses from the training data against DLM greedy and DSR decoding in Figure E.21.

While it appears that increasing the training data WER generally leads to better performance, it is not a strong guarantee. Consider plot (b) where the training data WER is around 13.7% but the DLM DSR WER varies between 3.81% (our best result) and 3.97%.

Another metric to measure the quality of text is by scoring it with a language model. Those texts that are good representations of the text distribution in the LibriSpeech LM dataset are scored high, and we expect that the sentences with more errors are scored lower. We use the language model

---

[14]The mismatch between greedy and DSR performance does not seem to occur in the results published by Gu et al. (2024), though the sample size is not very large

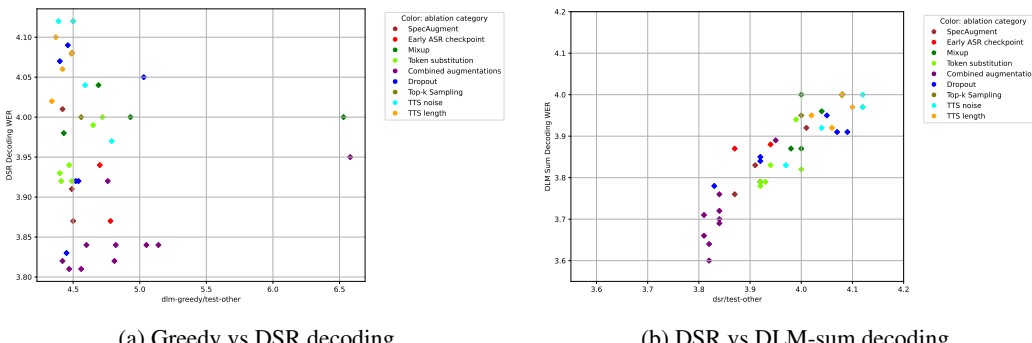

| (a) Greedy vs DSR decoding | (b) DSR vs DLM-sum decoding |

Figure E.20: We compare different decoding methods against each other. Every point corresponds to a different data augmentation technique. The purple data points correspond to our data augmentation combination experiments in Section 4.2.

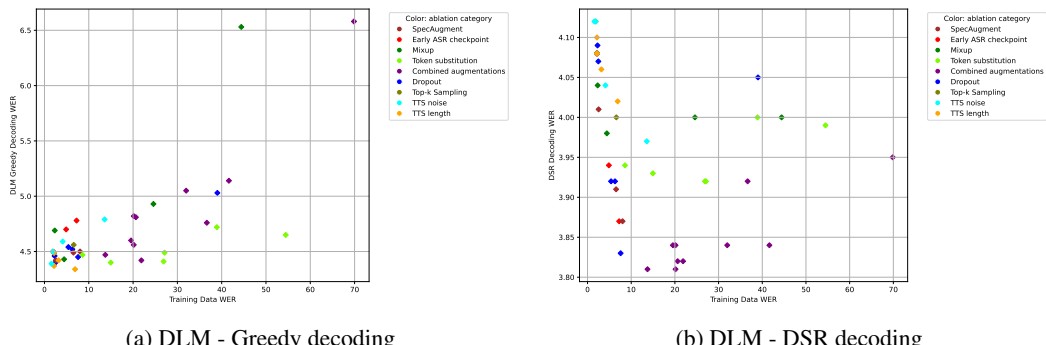

(a) DLM - Greedy decoding

(b) DLM - DSR decoding

Figure E.21: WER of training data hypotheses plotted against DLM performance on test-other.

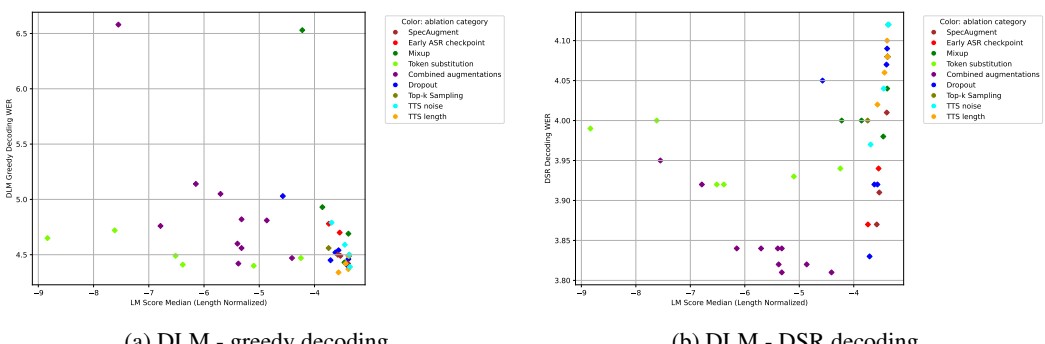

(a) DLM - greedy decoding

(b) DLM - DSR decoding

Figure E.22: Median LM log-probability of training data hypotheses plotted against DLM performance on test-other. Scores were performed with the `n8-d1024` LM from Section 3.2 and length-normalized by dividing by the number of tokens in the hypothesis.

`n8 d1024` from Section 3.2 to assign every hypothesis a probability $p_{\text{LM}}(a_1^S)$, and then length normalize it by dividing the log-probability with the number of tokens in the hypothesis (including end-of-sentence token). This reduces outliers where very short hypotheses are assigned a very high probability[15]. Then we take the median of the log-probabilities over all hypotheses to get a single score for the whole dataset. The median length-normalized LM log-probability is plotted against DLM performance in Figure E.22.

The greedy decoding plot (a) again looks somewhat random, but there seems to be a nice correlation in the DSR decoding plot (b). The best performing DLMs seem to have training data with a median length-normalized LM log-probability between -4.5 and -5.5. But again this does not guarantee good performance, as can be seen by the data points in that range that go up to 4.03% WER.

Motivated by the observation that greedy and DSR decoding seem to be somewhat negatively correlated, meaning that as greedy WER goes up, DSR WER goes down, we hypothesize that the DLM learns some characteristic that makes it less confident in its own predictions but better tolerates model combination. We therefore compute the entropy of the DLM output distribution for every output token and average it over all tokens and all hypotheses to get a single score for the whole

---

[15]Hypothesis shrinking only existed for high dropout values. Other data points moved only by a small amount after applying length normalization.

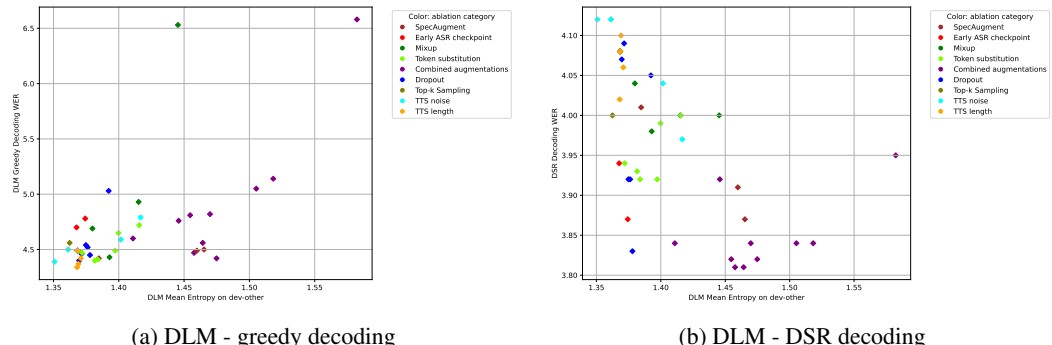

(a) DLM - greedy decoding          (b) DLM - DSR decoding

Figure E.23: Mean label-wise entropy of DLM output distribution plotted against DLM performance on test-other.

dataset. The entropy for a single output token is computed as follows:

$$H(j, p_{\text{DLM}}) = -\sum_{i=1}^{|V|} p_i \log p_i \tag{28}$$

$$\text{where } p_i := p_{\text{DLM}}(a_j = v_i | a_1^{j-1}, \tilde{a}_1^{\tilde{S}})$$

$$\text{and } j \text{ is the position of the output token}$$

$$\text{and } v_i \in V \text{ is the i-th token in the vocabulary}$$

Results are shown in Figure E.23.

This partially confirms our hypothesis, Greedy WER appears to increase with higher entropy, while DSR WER decreases. But again, the correlation is not perfect, and significant outliers exist.

Another related metric is the Expected Calibration Error (ECE) (Lee & Chang, 2021), which measures how well the predicted probabilities of a model reflect the true accuracy. Intuitively, it is a measure of whether the model is over/underconfident or well-calibrated. When we plot ECE against DLM performance, we arrive at an almost exact replica of Figure E.23, and it turns out that ECE and entropy are highly correlated in our case. We therefore do not show the ECE plots here, but rather the correlation between ECE and entropy in Figure E.24. Further investigation is needed to understand why ECE and entropy are so closely related in our case.

A reliability diagram as per Lee & Chang (2021) is shown in Figure E.25.

For completeness, we also show the correlation between training data WER and entropy in Figure E.26b and between training data WER and LM score in Figure E.26a.

### E.3.6 SOFTMAX TEMPERATURE

As motivated by the previous section, we want to see if we can improve DLM performance by adjusting the peakiness of the DLM output distribution. Softmax Temperature scaling is defined as follows:

$$p_i = \frac{\exp(z_i/T)}{\sum_{j=1}^{|V|} \exp(z_j/T)} \tag{29}$$

$$\text{where } z_i \text{ are the logits of the DLM output distribution}$$

where $T$ is the temperature. A temperature of $T = 1.0$ corresponds to the original distribution, $T < 1.0$ makes the distribution peakier, and $T > 1.0$ makes it softer ($\rightsquigarrow$ higher entropy). We take a baseline DLM trained without data augmentation, and apply temperature scaling to the output distribution during decoding. We do not use a better DLM with data augmentation because those already have a higher mean entropy, and we want to see if we can recover similar improvements by artificially increasing the entropy. The baseline DLM has a mean entropy of 1.34. If there is a causal relationship between entropy and DLM performance, we expect to see a performance improvement

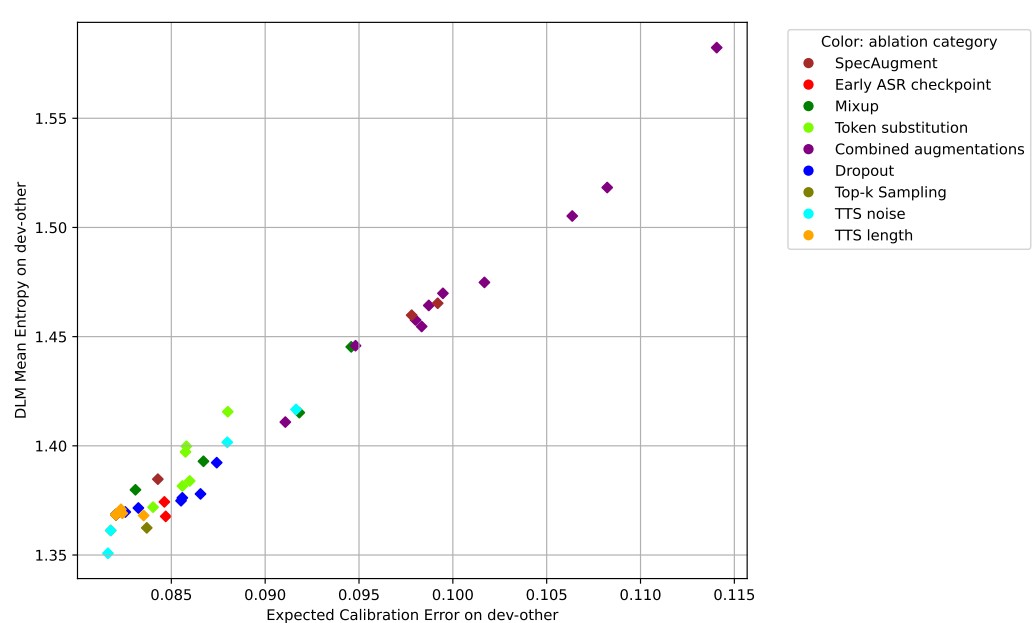

Figure E.24: Expected Calibration Error plotted against Mean Entropy, both on dev-other.".

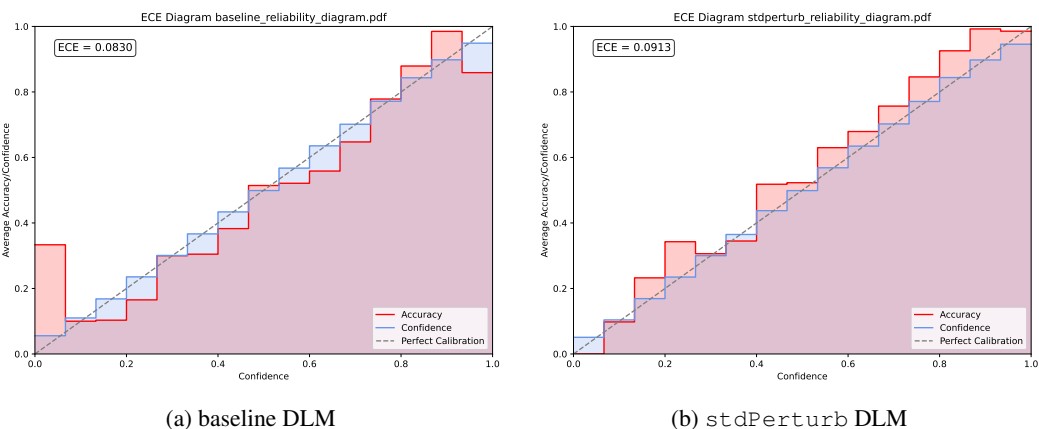

(a) baseline DLM

(b) stdPerturb DLM

Figure E.25: DLM reliability diagram, as per Lee & Chang (2021). Both Models match their predicted confidence and true accuracy quite well. Confidence is the maximum probability of the model output distribution. Accuracy is the % of how often the argmax of the probability distribution matches the ground truth. Values are binned according to confidence.

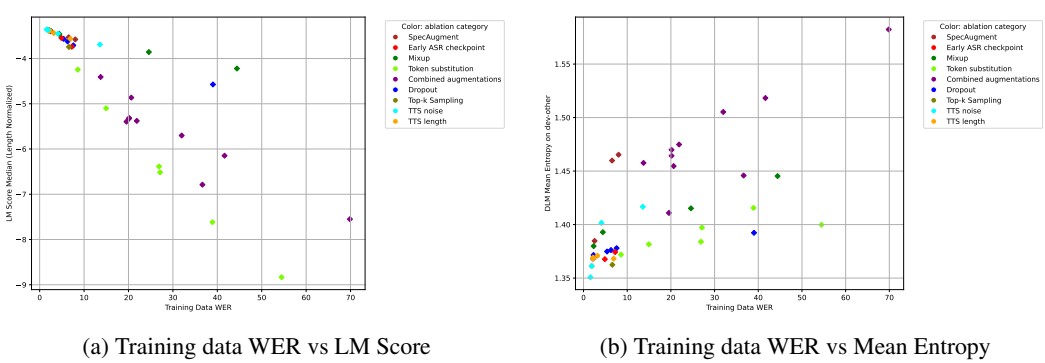

(a) Training data WER vs LM Score

(b) Training data WER vs Mean Entropy

Figure E.26: Correlation between different metrics

Table E.27: Softmax Temperature.

| Softmax Temperature | Decoding | DLM Performance: WER [%] | | | |
|---|---|---|---|---|---|
| | | dev-clean | dev-other | test-clean | test-other |
| 0.5 | DSR | 1.58 | 3.73 | 1.87 | 4.17 |
| | DLM-sum | 1.58 | 3.65 | 1.85 | 3.97 |
| 0.6 | DSR | 1.62 | 3.73 | 1.89 | 4.15 |
| | DLM-sum | 1.60 | 3.69 | 1.85 | 3.96 |
| 0.7 | DSR | 1.59 | 3.73 | 1.90 | 4.11 |
| | DLM-sum | 1.56 | 3.63 | 1.85 | 3.93 |
| 0.8 | DSR | 1.59 | 3.73 | 1.89 | 4.10 |
| | DLM-sum | 1.58 | 3.65 | 1.82 | 3.91 |
| 0.9 | DSR | 1.59 | 3.72 | 1.87 | 4.01 |
| | DLM-sum | 1.55 | 3.64 | 1.83 | 3.95 |
| 0.95 | DSR | 1.59 | 3.73 | 1.86 | 4.00 |
| | DLM-sum | 1.53 | 3.63 | 1.82 | 3.91 |
| 0.96 | DSR | 1.59 | 3.72 | 1.87 | 4.02 |
| | DLM-sum | 1.53 | 3.62 | 1.81 | 3.91 |
| 0.97 | DSR | 1.59 | 3.71 | 1.86 | 4.00 |
| | DLM-sum | 1.53 | 3.64 | 1.82 | 3.91 |
| 0.98 | DSR | 1.59 | 3.71 | 1.86 | 4.01 |
| | DLM-sum | 1.53 | 3.62 | 1.82 | 3.91 |
| 0.99 | DSR | 1.59 | 3.71 | 1.86 | 4.01 |
| | DLM-sum | 1.53 | 3.62 | 1.81 | 3.91 |
| 1.0 | DSR | 1.58 | 3.71 | 1.87 | 4.02 |
| | DLM-sum | 1.53 | 3.61 | 1.81 | 3.89 |
| 1.01 | DSR | 1.58 | 3.71 | 1.86 | 4.02 |
| | DLM-sum | 1.53 | 3.61 | 1.81 | 3.91 |
| 1.02 | DSR | 1.58 | 3.71 | 1.86 | 4.02 |
| | DLM-sum | 1.53 | 3.61 | 1.81 | 3.90 |
| 1.03 | DSR | 1.58 | 3.71 | 1.86 | 4.02 |
| | DLM-sum | 1.53 | 3.61 | 1.81 | 3.91 |
| 1.04 | DSR | 1.58 | 3.71 | 1.86 | 4.01 |
| | DLM-sum | 1.53 | 3.62 | 1.81 | 3.92 |
| 1.05 | DSR | 1.58 | 3.71 | 1.86 | 4.03 |
| | DLM-sum | 1.54 | 3.62 | 1.83 | 3.90 |
| 1.1 | DSR | 1.60 | 3.71 | 1.89 | 4.04 |
| | DLM-sum | 1.57 | 3.64 | 1.83 | 3.91 |
| 1.15 | DSR | 1.59 | 3.71 | 1.87 | 4.00 |
| | DLM-sum | 1.57 | 3.63 | 1.85 | 3.94 |
| 1.2 | DSR | 1.59 | 3.71 | 1.88 | 3.99 |
| | DLM-sum | 1.54 | 3.64 | 1.83 | 3.87 |
| 1.3 | DSR | 1.55 | 3.72 | 1.87 | 3.99 |
| | DLM-sum | 1.64 | 3.74 | 1.86 | 4.01 |
| 1.4 | DSR | 1.56 | 3.73 | 1.87 | 3.99 |
| | DLM-sum | 1.59 | 3.70 | 1.85 | 3.93 |
| 1.5 | DSR | 1.61 | 3.73 | 1.89 | 4.01 |
| | DLM-sum | 1.67 | 3.78 | 1.89 | 3.99 |
| 1.7 | DSR | 1.58 | 3.75 | 1.89 | 4.01 |
| | DLM-sum | 1.72 | 3.83 | 1.92 | 4.11 |
| 2.0 | DSR | 1.62 | 3.78 | 1.90 | 4.03 |
| | DLM-sum | 1.73 | 3.90 | 1.90 | 4.09 |
| 2.3 | DSR | 1.67 | 3.83 | 1.86 | 4.06 |
| | DLM-sum | 1.72 | 3.92 | 1.92 | 4.14 |
| 2.66 | DSR | 1.67 | 3.87 | 1.91 | 4.10 |
| | DLM-sum | 1.72 | 3.93 | 1.96 | 4.22 |
| 3.0 | DSR | 1.63 | 3.89 | 1.90 | 4.08 |
| | DLM-sum | 1.69 | 3.96 | 1.92 | 4.20 |
| 3.5 | DSR | 1.65 | 3.92 | 1.91 | 4.14 |
| | DLM-sum | 1.67 | 3.93 | 1.93 | 4.24 |

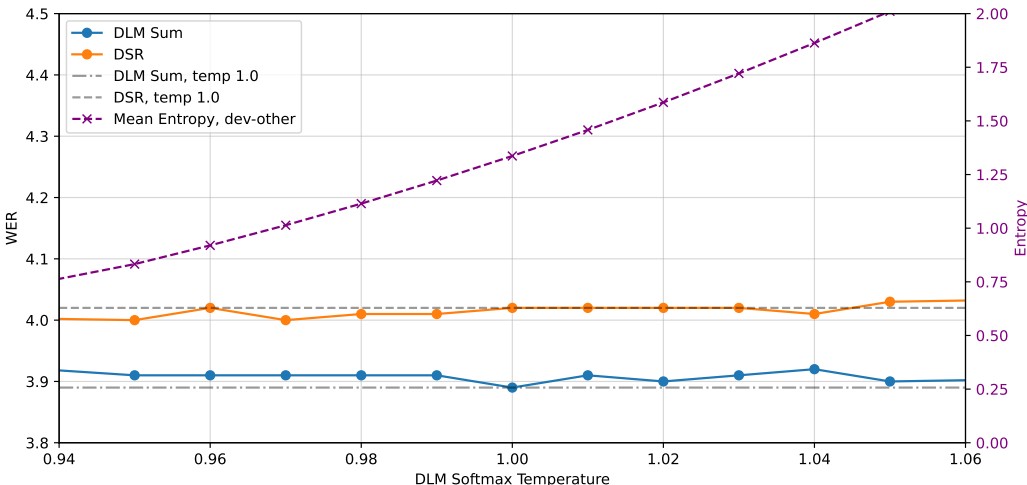

Figure E.27: DLM Softmax Temperature plotted against DLM performance on test-other. DLM is a baseline model trained without data augmentation for 10 epochs. Raw data can be found in Table E.27.

with higher temperature. Results for different temperatures are shown in Figure E.27. Greedy results are not shown as they are invariant under softmax temperature scaling.

There does not seem to be a performance improvement for adjusting temperature in either direction, so we conclude that entropy of the output distribution is not causally related to DLM performance, but rather a consequence of another underlying factor.

### E.3.7 ERROR ANALYSIS BY CATEGORIZATION

Because DLMs are trained to correct errors, it is interesting to see which types of errors are corrected best. For this we bin all words into categories, and then we look at the WER per category. Only substitution errors are considered here, as those account for 80% of all word errors on test-other for the ASR spm10k (TTS) model[16]. When we look at the categories of substitution errors, we consider the word in the reference text, not the word it was substituted with in the hypothesis. For all the following analyses, we use the DLM trained with stdPerturb data augmentation for 10 epochs.

### E.3.8 WORD FREQUENCY

We concatenate the LibriSpeech LM corpus and the LibriSpeech ASR text data to get a large corpus of text. Then the occurence of every word in the combined corpus is counted and the words are sorted by their frequency. Three bins are created: "common" words are the words that with their counts combined make up 50% of all word occurences. "medium" words make up the next 49% of word occurences, and "rare" words are the remaining 1%. Table E.28 shows a selection of words contained in the bins.

We begin by looking at the WER of the ASR hypotheses in each bin, shown in Table E.29.

Immediately we see that the ASR model struggles with rare words, 58.8% of which are misrecognized. But it seems that the DLM is very effective at correcting errors in all three bins equally, with similar procentual reductions in absolute errors across all bins.

---

[16]On test-other, the ASR model spm10k (TTS) has 3.5% substitution, 0.4% deletion and 0.5% insertion word errors. After correction using DLM-sum with stdPerturb DLM trained for 10 epochs, this reduces to 2.8% substitution, 0.3% deletion and 0.4% insertion word errors.

Table E.28: Example words in each frequency bin. The bins were created such that "common" words make up 50% of all word occurences, "medium" words make up the next 49% and "rare" words make up the remaining 1%.

| | Common | Medium | Rare |
|---|---|---|---|
| Size of bin | 83 | 59244 | 913464 |
| | the | before | masin |
| | and | other | peschiera |
| | of | know | balt |
| Selection of words | ... | ... | ... |
| | see | gyroscope | marspeaker |
| | its | brett's | schulberg's |
| | down | cordie | kerstall |

Table E.29: Number of substitution errors categorized by word frequency. "abs" shows the absolute number of substitutions in each frequency bin, "rel" shows the percentage of substitutions in that bin relative to the number of words in that bin. "abs perc" shows the percentage of absolute substitutions relative to the baseline (ASR) model.

| Substitutions by Frequency | test-other | | | | | | | | |
|---|---|---|---|---|---|---|---|---|---|
| | common | | | medium | | | rare | | |
| | abs | abs perc | rel | abs | abs perc | rel | abs | abs perc | rel |
| ASR | 358 | - | 1.35% | 1124 | - | 4.48% | 370 | - | 58.82% |
| DLM greedy | 339 | -5.31% | 1.27% | 1123 | -0.09% | 4.47% | 327 | -11.62% | 51.99% |
| DLM DSR | 305 | -14.80% | 1.15% | 924 | -17.79% | 3.68% | 315 | -14.86% | 50.08% |
| DLM-sum | 287 | -19.83% | 1.08% | 871 | -22.51% | 3.47% | 314 | -15.14% | 49.92% |

Table E.30: Example words for each Part-of-Speech (POS) tag. The POS tags were assigned using the SpaCy library. The assignment of word to POS Tag is not perfect, but good enough for our purposes.

| POS Tag | Count | Example Words |
|---|---|---|
| PROPN | 578003 | shippenburgh, strasburgh, dutchman, ... |
| NOUN | 291990 | town, street, mile, ... |
| VERB | 62502 | contains, built, saw, ... |
| ADJ | 12818 | dirty, pleasant, handsome, ... |
| ADV | 7356 | only, thickly, very, ... |
| ADP | 1561 | in, of, from, ... |
| PRON | 772 | we, the, a, some, ... |
| other tags | 17789 | that, if, was, and, three, ... |

### E.3.9 PART OF SPEECH

Sentences in the english language consist of different types of words, such as nouns, verbs, adjectives, etc. We are interested in seeing if the DLM is better at correcting certain types of words than others. For example, one could hypothesize that ASR models have trouble with recognizing names of people and places (= proper nouns[17]) which are rare in the training data, and with the additional help of the LibriSpeech LM corpus the DLM may be able to correct these errors. We use the SpaCy[18] library to automatically assign every word in the corpus a part of speech (POS) tag, specifically we use the en_core_web_lg:3.8.0 model. A subset of some common POS tags is shown in Table E.30.

The WER categorized by POS tag is shown in Table E.31.

---

[17]A list of POS tags and their meanings can be found at https://universaldependencies.org/u/pos/.

[18]https://spacy.io

Table E.31: Number of substitution errors categorized by Part-of-Speech tag. "abs" shows the absolute number of substitutions in each frequency bin, "rel" shows the percentage of substitutions in that bin relative to the number of words in that bin. "abs perc" shows the percentage of absolute substitutions relative to the baseline (ASR) model.

| Substitutions by POS | test-other | | | | | | | | | | | | | | |
| | PROPN | | | NOUN | | | VERB | | | ADJ | | | other | | |
| | abs | abs perc | rel | abs | abs perc | rel | abs | abs perc | rel | abs | abs perc | rel | abs | abs perc | rel |
|---|---|---|---|---|---|---|---|---|---|---|---|---|---|---|---|
| ASR | 661 | - | 11.94% | 426 | - | 5.92% | 240 | - | 2.93% | 80 | - | 3.12% | 445 | - | 1.54% |
| DLM greedy | 618 | -6.51% | 11.16% | 410 | -3.76% | 5.70% | 228 | -5.00% | 2.78% | 80 | -0.00% | 3.12% | 453 | –1.80% | 1.57% |
| DLM DSR | 567 | -14.22% | 10.24% | 325 | -23.71% | 4.52% | 189 | -21.25% | 2.30% | 69 | -13.75% | 2.69% | 394 | -11.46% | 1.37% |
| DLM-sum | 536 | -18.91% | 9.68% | 327 | -23.24% | 4.54% | 185 | -22.92% | 2.25% | 57 | -28.75% | 2.23% | 367 | -17.53% | 1.27% |

Table E.32: Number of substitution errors categorized by Part-of-Speech tag. "abs" shows the absolute number of substitutions in each frequency bin, "rel" shows the percentage of substitutions in that bin relative to the number of words in that bin. "abs perc" shows the percentage of absolute substitutions relative to the baseline (ASR) model.

*(Table E.32 — full wide table across all POS tags (PRON, CCONJ, ADP, AUX, PROPN, NOUN, VERB, INTJ, ADJ, PART, ADV, SCONJ, NUM, PUNCT, DET, X) with abs / abs perc / rel columns for each; rows ASR, DLM greedy, DLM DSR, DLM-sum.)*

Table E.33: Correction statistics of a baseline DLM trained without data augmentation and trained for 10 epochs. "Before" statistics are from the ASR model, "After" statistics are from the DLM-corrected hypotheses. Each number represents a word in LibriSpeech test-other.

(a) Greedy

| | | After | | |
| | | Correct | Incorrect | Total |
|---|---|---|---|---|
| Before | Correct | 49939 | 339 | 50278 |
| | Incorrect | 273 | 1792 | 2065 |
| | Total | 50212 | 2131 | 52343 |

(b) DSR

| | | After | | |
| | | Correct | Incorrect | Total |
|---|---|---|---|---|
| Before | Correct | 50210 | 68 | 50278 |
| | Incorrect | 275 | 1790 | 2065 |
| | Total | 50485 | 1858 | 52343 |

(c) DLM-sum

| | | After | | |
| | | Correct | Incorrect | Total |
|---|---|---|---|---|
| Before | Correct | 50164 | 114 | 50278 |
| | Incorrect | 392 | 1673 | 2065 |
| | Total | 50556 | 1787 | 52343 |

Table E.34: Correction statistics of DLM with `stdPerturb` data augmentation, and trained for 10 epochs. "Before" statistics are from the ASR model, "After" statistics are from the DLM-corrected hypotheses. Each number represents a word in LibriSpeech test-other.

(a) Greedy

| | | After | | |
| | | Correct | Incorrect | Total |
|---|---|---|---|---|
| Before | Correct | 49827 | 451 | 50278 |
| | Incorrect | 416 | 1649 | 2065 |
| | Total | 50243 | 2100 | 52343 |

(b) DSR

| | | After | | |
| | | Correct | Incorrect | Total |
|---|---|---|---|---|
| Before | Correct | 50181 | 97 | 50278 |
| | Incorrect | 442 | 1623 | 2065 |
| | Total | 50623 | 1720 | 52343 |

(c) DLM-sum

| | | After | | |
| | | Correct | Incorrect | Total |
|---|---|---|---|---|
| Before | Correct | 50147 | 131 | 50278 |
| | Incorrect | 554 | 1511 | 2065 |
| | Total | 50701 | 1642 | 52343 |

The ASR struggles most with proper nouns (PROPN) and nouns (NOUN). Again, the DLM corrects errors of all POS tags about equally well, with some small variations which may not be statistically significant. A complete table is shown in Table E.32.

### E.3.10    CORRECTION VS. DEGRADATION ANALYSIS

Binary classifiers are often evaluated with so-called confusion matrices, which show how often a classifier predicts a positive or a negative class, and how often that prediction is correct. We adapt this idea to our DLM to show how often the ASR model correctly predicts a word, and what happens after a DLM is applied for error correction. On the vertical axis we show the previous state of the word in the ASR hypotheses, whether they are correct or incorrect. On the horizontal axis we show the new state of the word after DLM correction. Here, we only consider the words in the reference text, so these numbers only capture substitution and deletion errors. See Table E.33 and Table E.34 for results on a baseline DLM without data augmentation and a DLM trained with `stdPerturb` data augmentation, respectively.

### E.3.11    ERROR EXAMPLES

Finally, we look at some specific examples of errors that the ASR model makes, and how the DLM corrects them (or fails). We will show examples from the LibriSpeech test-other set, from a DLM trained with `stdPerturb` data augmentation for 10 epochs.

Table E.35: Example 1. The ASR model makes several mistakes, some of which are corrected by the DLM. See text for details.

| Reference | [. . . ] his note book | which [. . . ] father mestienne's | turn father mestienne | died |
|---|---|---|---|---|
| ASR | [. . . ] his **** NOTEBO | which [. . . ] father MEIEEN'S | turn father MAIEIENNEEN | died |
| DLM greedy | [. . . ] his **** NOTEBOOK | which [. . . ] father MAHOMMED'S | turn father MAHOMMED | died |
| DLM DSR | [. . . ] his note book | which [. . . ] father MAHIEN'S | turn father MAHIEN | died |
| DLM-sum | [. . . ] his note book | which [. . . ] father MAIEN'S | turn father MAIEN | died |

Table E.36: Example 2. The ASR makes mistakes which even the DLM can not correct.

| Reference | say awk | ward | in *** | future not awk'ard |
|---|---|---|---|---|
| ASR | say *** | AWKWARD | in THE | future not AWKWARD |
| DLM greedy | say *** | AWKWARD | in THE | future not AWKWARD |
| DLM DSR | say *** | AWKWARD | in THE | future not AWKWARD |
| DLM-sum | say *** | AWKWARD | in THE | future not AWKWARD |

Table E.37: Example 3. The DLM exhibits broken behaviour and hallucinations with greedy decoding.

| Reference | [. . . ] the hashish | the [. . . ] toward porto vecchio |
|---|---|---|
| ASR | [. . . ] the hashish | the [. . . ] toward porto vecchio |
| DLM-greedy | [. . . ] the HASCHICH | the [. . . ] toward porto THE YACHT SEEN <17 words omitted>ALL THE YACHT SEEN OF THE |
| DLM DSR | [. . . ] the hashish | the [. . . ] toward porto vecchio |
| DLM-sum | [. . . ] the hashish | the [. . . ] toward porto vecchio |

The first example is shown in Table E.35. The ASR model makes several mistakes, it misrecognizes "notebook" as "NOTEBO", "mestienne's" as "MEIEEN'S", and "maieienneen" as "MAIEI-ENNEEN". The DLM with greedy decoding completes "NOTEBO" to "NOTEBOOK", and figures out that the two names should be identical, changing both to "MAHOMMED". With DSR and DLM-sum decoding, further audio information is available, and "note book" is recognized correctly, but the names are still wrong, but now more consistent to each other than in the ASR hypothesis.

Our second example is shown in Table E.36. The ASR model misrecognizes several words in the sentence, and the DLM is not able to correct any of them. This is an example where the ASR output is already quite different from the reference, and there is not enough context for the DLM to figure out what the correct words should be. Anecdotally, from looking at many corrections and miscorrections in the test sets, we find that the DLM generally is better at correcting errors on longer sentences than on shorter ones (⤳ more context helps).

We observe very strange behaviour with greedy decoding in some cases, where the DLM output is completely broken and contains hallucinated words. An example is shown in Table E.37. The ASR model makes no mistakes in this sentence, but the DLM with greedy decoding introduces 27 word errors, hallucinating 26 extra words at the end of the sentence.

We find three more examples in the test-other set where the DLM with greedy decoding produces broken output, with word errors of 35, 40, and 82. All of these sentences are quite long, so we assume that the DLM does not perform well on very long sentences. Together these four sentences are responsible for an increase of 0.35% in WER on test-other. We believe these types of broken behaviour and hallucinations are the reason why our greedy results seem so random, and could explain why they are worse than the results reported by Gu et al. (2024). With DSR and DLM-sum decoding, this behaviour vanishes. Figuring out the root cause and a solution to this problem is left for future work, and could potentially lead to significant improvements in the other decoding methods as well.

Table E.38: Random seed experiment. Trained for 5 epochs, data augmentation configuration `stdPerturb`.

| Random seed | Decoding | DLM Performance: WER [%] | | | |
|---|---|---|---|---|---|
| | | dev-clean | dev-other | test-clean | test-other |
| 1 | greedy | 1.95 | 4.05 | 2.25 | 4.60 |
| | DSR | 1.54 | 3.54 | 1.77 | 3.84 |
| | DLM-sum | 1.45 | 3.45 | 1.76 | 3.69 |
| 2 | greedy | 2.09 | 3.99 | 2.25 | 4.53 |
| | DSR | 1.55 | 3.49 | 1.80 | 3.86 |
| | DLM-sum | 1.53 | 3.42 | 1.77 | 3.69 |
| 3 | greedy | 1.94 | 3.98 | 2.23 | 4.48 |
| | DSR | 1.54 | 3.51 | 1.77 | 3.85 |
| | DLM-sum | 1.53 | 3.41 | 1.76 | 3.71 |

Table E.39: DLM with `stdPerturb` data augmentation trained for 10 epochs is applied for multiple iterations, where the output of the DLM is fed back as input to the DLM.

| Iteration | DLM Training Data WER [%] | | | |
|---|---|---|---|---|
| | dev-clean | dev-other | test-clean | test-other |
| ASR | 1.75 | 4.13 | 2.03 | 4.44 |
| ASR $\rightarrow$ DLM$^1$ | 2.11 | 3.80 | 2.21 | 4.43 |
| ASR $\rightarrow$ DLM$^2$ | 2.22 | 3.91 | 2.39 | 4.56 |
| ASR $\rightarrow$ DLM$^3$ | 2.24 | 3.92 | 2.40 | 4.57 |
| ASR $\rightarrow$ DLM$^4$ | 2.23 | 3.92 | 2.40 | 4.56 |

### E.4 ABLATIONS ON MODEL AND TRAINING VARIATIONS

#### E.4.1 RANDOMNESS

We evaluate the run-to-run variance of our DLM training. For this test, we generate a single full training dataset from the LibriSpeech LM corpus and 10x the LibriSpeech ASR training data. Then we randomly initialize three DLMs with unique random seeds and train them on the same training data. We use the `stdPerturb` data augmentation configuration. The results are shown in Table E.38. The greedy results seem a bit less stable than DSR and DLM-sum decoding, but overall the variance stays within $\pm 0.06\%$ WER.

#### E.4.2 LOOPING THE ERROR CORRECTION MODEL

We briefly experiment with feeding the output of the DLM back into itself and see if it improves the WER further. Here we use the top greedy hyp from the ASR model, and then do beam search with the DLM using beam size 64. The resulting WER of repeatedly applying the DLM is shown in Table E.39.

It appears that applying the DLM twice decreases the WER, and further iterations have limited effect. We create a new training dataset with hypotheses not from the ASR model, but from applying the DLM once with beam size 1, and add these new hypotheses to the original DLM training data. We then train a new DLM from scratch on this new dataset. Because the new dataset has twice the size of the original DLM training data, we reduce the number of epochs to 5 instead of 10. The results are shown in Table E.40.

We observe that the DLM may be more robust to multiple iterations of error correction on test-other, but additional iterations still do not improve over the first one. For dev-other, the WER even increases sharper than before due to one hypothesis that has moved so far from the training distribution that the DLM exhibits broken out-of-domain behavior and generates an incomprehensible

Table E.40: DLM trained for 5 epochs on hypotheses with `stdPerturb` data augmentation **and** on outputs from another DLM. The DLM is looped for multiple iterations, where the output of the DLM is fed back as input to the DLM.

| Iteration | DLM Training Data WER [%] | | | |
|---|---|---|---|---|
| | dev-clean | dev-other | test-clean | test-other |
| ASR | 1.75 | 4.13 | 2.03 | 4.44 |
| ASR $\rightarrow$ DLM$^1$ | 1.88 | 3.85 | 2.11 | 4.29 |
| ASR $\rightarrow$ DLM$^2$ | 1.89 | 3.91 | 2.13 | 4.30 |
| ASR $\rightarrow$ DLM$^3$ | 1.89 | 3.97 | 2.13 | 4.31 |
| ASR $\rightarrow$ DLM$^4$ | 1.89 | 4.09 | 2.13 | 4.31 |

Table E.41: DLMs trained for different vocabularies without additional data augmentation, for 5 epochs. ASR baselines are all trained with TTS audio data.

| Vocabulary | Model | Decoding | DLM Performance: WER [%] | | | |
|---|---|---|---|---|---|---|
| | | | dev-clean | dev-other | test-clean | test-other |
| spm10k | ASR | - | 1.75 | 4.13 | 2.03 | 4.44 |
| | DLM | greedy | 1.82 | 3.98 | 2.05 | 4.49 |
| | | DSR | 1.56 | 3.78 | 1.89 | 4.08 |
| | | DLM-sum | 1.55 | 3.67 | 1.84 | 4.00 |
| spm128 | ASR | - | 1.79 | 4.41 | 1.94 | 4.55 |
| | DLM | greedy | 1.72 | 4.17 | 1.90 | 4.29 |
| | | DSR | 1.58 | 3.88 | 1.77 | 4.09 |
| | | DLM-sum | 1.49 | 3.85 | 1.74 | 3.98 |
| char | ASR | - | 1.83 | 4.56 | 1.98 | 4.78 |
| | DLM | greedy | 1.80 | 4.24 | 1.92 | 4.55 |
| | | DSR | 1.61 | 3.94 | 1.95 | 4.20 |
| | | DLM-sum | 1.53 | 3.95 | 1.74 | 4.08 |

sequence of words that doubles in size with every iteration[19]. An arguably critical flaw of this experiment is that the DLM does not know which iteration it is in, and thus can not adapt its behavior to make effective use of additional iterations. We leave a more thorough investigation of this idea to future work.

### E.4.3 DIFFERENT VOCABULARIES

All DLM experiments so far have been conducted with a Sentence Piece 10k subword vocabulary. The prior work by Gu et al. (2024), which this work builds upon, exclusively uses a character vocabulary with remarkable results. Unlike with Sentence Piece subwords, a character vocabulary provides a single unique token representation for every sentence, a 1:1 bijective mapping. Unfortunately these character token representations become quite long compared to the Sentence Piece 10k, and we observe that a character DLM takes about 205h to train for 5 full epochs, whereas an spm10k DLM with the same hyperparamters takes only about 68h to complete the same training, a $\approx$3x increase. Similarly, training the ASR model with a character vocabulary further increases the computation time for training data generation. Sentence Piece 10k training data generation of LibriSpeech LM takes about 75 x 62 minutes, while character data generation takes about 75 x 98 minutes[20].

We train DLMs for each vocabulary with training data from the TTS-trained ASR model with that vocabulary, without additional data augmentation. Results are shown in Table E.41.

The performance of the models roughly follows the order of the vocabulary size, with the character model being the worst and the spm10k being the best. The differences, especially on the test sets,

---

[19]This doubling occurs because our search limits the output length to twice the input length. Otherwise the WER may have degraded even further.

[20]75 inference jobs, each taking 62 or 92 minutes to run.

Table E.42: We attempt to replicate the results from Gu et al. (2024). There are significant differences between the two experiments, for details see text.

| Replication attempts | Decoding | DLM Performance: WER [%] | | | |
|---|---|---|---|---|---|
| | | dev-clean | dev-other | test-clean | test-other |
| ASR Baseline | - | 1.83 | 4.56 | 1.98 | 4.78 |
| Gu et al. (2024) DLM architecture, with YourTTS | greedy | 5.40 | 5.99 | 5.86 | 7.36 |
| | DSR | 3.08 | 5.01 | 3.99 | 5.54 |
| | DLM-sum | 1.99 | 3.95 | 2.02 | 4.33 |
| Our DLM architecture, w/o YourTTS | greedy | 4.98 | 4.91 | 4.37 | 6.14 |
| | DSR | 2.49 | 3.76 | 2.31 | 4.79 |
| | DLM-sum | 1.50 | 3.61 | 1.70 | 3.82 |

are not that far off, and we may close the gap with additional tuning. It is also likely that the DLMs for spm128 and character vocabularies are held back by their worse ASR model. But given the significant increase in training and data generation time, we do not pursue the character and spm128 vocabularies any further in this work.

We run a few more experiments with the character vocabulary for comparison with Gu et al. (2024). To follow their configuration, we increase model dimension to 1280, and set encoder layers to 4 and decoder layers to 16, dropout and layer drop to 0.1 and gradient clipping to 0.1. We make training data with both Glow-TTS and YourTTS, using SpecAugment with only frequency masking, and token substitution with $p = 0.1$. We add 10x LibriSpeech ASR data to the training data, and train for 10 epochs[21]. We use learning rate schedule of 1e-7 to 1e-3 linear warmup over 1 epoch, then constant for 7 epochs, then linear decay to 1e-5 over the last 2 epochs. We use batch size 40k through gradient accumulation.

We start another training, using training data with the data augmentation as specified above, but our usual DLM architecture, learning rate, and without YourTTS data, which results in our best character vocabulary performance.

Results for these two experiments are shown in Table E.42.

Also see Appendix G.2 for some further discussion on character vs. subword vocabularies and the comparison to Gu et al. (2024).

### E.4.4 JOINT AED AND CTC MODEL

It has been shown that learning auxiliary tasks can improve the performance of the main task in end-to-end ASR models (Toshniwal et al., 2017). One such approach for encoder-decoder attention models involves using a CTC loss on the encoder output in conjunction with the existing cross-entropy loss on the decoder output (Watanabe et al., 2017). This arrangement can be understood as two models, a CTC model (the encoder) and an autoregressive transformer model (encoder + decoder) where the encoder is shared between both models. This configuration achieves superior results compared to only using the cross entropy loss of the transformer.

We propose a similar approach for DLMs in addition to the decoder cross-entropy loss. Besides the existing encoder architecture, we add another Linear layer with input dimension as the model dimension, and output dimension twice that of the vocabulary size + blank token. The output of the linear layer is split into two equally sized parts (i.e. each input token to the encoder is mapped to two auxiliary output tokens), and apply softmax. These additional output tokens are necessary to ensure that repeating tokens can be emitted by these auxiliary outputs without them being collapsed by the CTC alignment. Finally, we concatenate the tokens of all encoder frames together and compute the CTC loss to the target sequence. We postfix the target sequence with an end-of-sentence token, just like the input to the encoder. We hypothesize that this makes the task easier to learn as most tokens can flow through the residual stream of the encoder unchanged. Given our ASR models' splendid performance, we expect that most errors in its hypotheses can be fixed with substitutions

---

[21]Here we consider 1 epoch to consist of 1x LM corpus YourTTS audio, 1x LM corpus Glow-TTS audio and 10x LibriSpeech ASR.

Table E.43: Experiments with Auxiliary Loss in the DLM encoder. All models trained for 10 epochs on `stdPerturb` data augmentation configuration.

| Auxiliary Encoder Loss | Decoding | DLM Performance: WER [%] | | | |
|---|---|---|---|---|---|
| | | dev-clean | dev-other | test-clean | test-other |
| no aux loss, baseline | greedy | 2.26 | 3.96 | 2.47 | 4.70 |
| | DSR | 1.51 | 3.46 | 1.78 | 3.73 |
| | DLM-sum | 1.50 | 3.32 | 1.80 | 3.57 |
| with aux loss, weight 1.0 | greedy | 2.79 | 4.10 | 2.70 | 5.44 |
| | DSR | 1.57 | 3.44 | 1.82 | 3.86 |
| | DLM-sum | 1.53 | 3.36 | 1.76 | 3.66 |
| | aux CTC only | 1.96 | 4.06 | 2.25 | 4.37 |
| | joint CTC + AED greedy | 1.93 | 4.00 | 2.23 | 4.41 |
| | joint DSR | 1.56 | 3.49 | 1.79 | 3.84 |
| with aux loss, weight 0.1 | greedy | 2.83 | 4.17 | 2.65 | 5.59 |
| | DSR | 1.48 | 3.48 | 1.74 | 3.72 |
| | DLM-sum | 1.59 | 3.37 | 1.85 | 3.76 |
| | aux CTC only | 1.98 | 4.12 | 2.26 | 4.42 |
| | joint CTC + AED greedy | 2.33 | 4.02 | 2.64 | 4.76 |
| | joint DSR | 1.48 | 3.50 | 1.75 | 3.80 |

and deletions, which are particularly easy to learn in this setup. Multiple insertions however are more challenging, as input tokens would need to be shifted around to accommodate the new output tokens given the limited space in the output sequence. We add the auxiliary loss to layers 6, 12 and 24 of the encoder. The parameters of the auxiliary linear layer are shared between all three selected layers. We ran one experiment with a loss scale of 1.0, and one with a loss scale of 0.1 for all auxiliary losses. Our results are shown in Table E.43.

We see slight degradation in performance on the regular AED-only decoding methods. With joint decoding of the auxiliary CTC scores and the decoder scores, we almost reach baseline Greedy performance again. Using the joint DLM scores in DSR decoding is slightly worse than DSR decoding in the baseline DLM. We did not implement joint decoding for DLM-sum. Overall, the results with auxiliary loss are slightly worse than without. The auxiliary loss increases training time by about 20%, from 133h to 160h for 10 epochs.

### E.4.5 DENSE k-PROBABILITY INPUT TO DLM

We generate training data with the `stdPerturb` data augmentation configuration. Because preliminary experiments showed slight overfitting, we enable dropout in the DLM with $5\%$ for the Dense k-Probability training.[22] Results are shown in Table 5. The performance with DSR decoding of a DLM with dense k-probability input is now matching the performance with DLM-sum decoding of the standard DLM.

One concern with our current approach of using dense k-probabilities to compute the weighted sum of input embeddings is that critical information may be smoothed out, leaving an effectively less informative input representation for the DLM than before. Therefore we also try an alternative approach where we attend over the token embeddings similar to Cross-Attention, attending over the $k$-axis instead of the encoder spatial dimension. To supply probability information, we scale an additional learned embedding by the token probability before adding it to the token embedding. We also bias the attention weights with the token probabilities. This atttention mechanism is applied twice, with feed-forward layers in between. While this approach has faster initial convergence, final performance is equal to the much simpler weighted sum approach.

See Appendix G.2 for further discussion on the relation to Gu et al. (2024) and the character vocabulary.

Table E.44: Trained ASR model, (D)LM on LibriSpeech, WERs [%] on *Out-of-domain* evaluation sets.

| Method | CommonVoice | | VoxPopuli | | Yodas | |
|---|---|---|---|---|---|---|
| | dev | test | dev | test | dev | test |
| ASR only | 22.68 | 27.39 | 17.33 | 16.78 | 22.41 | 23.01 |
| ASR + LM | 20.05 | 24.17 | 15.75 | 15.20 | 20.69 | 21.15 |
| ASR + DLM | 20.56 | 25.33 | 15.90 | 15.62 | 21.18 | 21.99 |

Table E.45: Trained DLM on Librispeech ASR outputs from different ASR models.

| Method | WER [%] | | | |
|---|---|---|---|---|
| | dev-clean | dev-other | test-clean | test-other |
| Conformer ASR + LM | 1.83 | 3.94 | 1.99 | 4.26 |
| EBranchformer ASR + LM | 1.90 | 3.96 | 2.02 | 4.12 |
| Conformer ASR + DLM | 1.68 | 3.70 | 1.83 | 4.15 |
| EBranchformer ASR + DLM | 1.73 | 3.84 | 1.89 | 4.10 |

Table E.46: Results of our best LibriSpeech DLM model. ASR model is Conformer CTC, trained with LibriSpeech and TTS audio. For comparison to prior work, see Table 2.

| Method | WER [%] | | | |
|---|---|---|---|---|
| | dev-clean | dev-other | test-clean | test-other |
| ASR only | 1.75 | 4.13 | 2.03 | 4.44 |
| ASR + DLM (greedy) | 2.09 | 4.08 | 2.33 | 4.75 |
| ASR + DLM (DSR) | 1.50 | 3.40 | 1.74 | 3.66 |
| ASR + DLM (DLM-sum) | **1.48** | **3.26** | **1.70** | **3.44** |

### E.4.6 OUT-OF-DOMAIN GENERALIZATION.

We test how well the DLM generalizes to out-of-domain (OOD) evaluation sets. Specifically, we use the ASR model and LM or DLM trained on LibriSpeech data, and test it on CommonVoice, VoxPopuli, and Yodas evaluation datasets (here taken from the Loquacious corpus). Results are shown in Table E.44.

### E.4.7 GENERALIZATION TO OTHER ASR MODELS

We further test the generalization of our DLM approach to ASR outputs from another ASR model. We use an EBranchformer (Kim et al., 2023) CTC ASR model here, where the Conformer ASR model is the one used for DLM training data generation and the baseline. Results are shown in Table E.45. We see that the DLM improves over the ASR baseline and standard LM for both ASR models.

### E.5 OUR BEST MODELS

Our best LibriSpeech DLM (as shown in Table 2) follows our usual architecture (See Section 3.2, Appendix D.1.5) with the following changes: 1280 model dimension, 5120 feed-forward dimension, 10 training epochs; resulting in a 729M parameter model. We did not use this model in Table 1 as we do not have a directly comparable LM of similar size. DLM training data is generated from the TTS ASR spm10k model with `low` data augmentation. Results over all decoding methods are shown in Table E.46.

---

[22]Enabling 5% dropout in normal DLM training did not lead to performance gains.

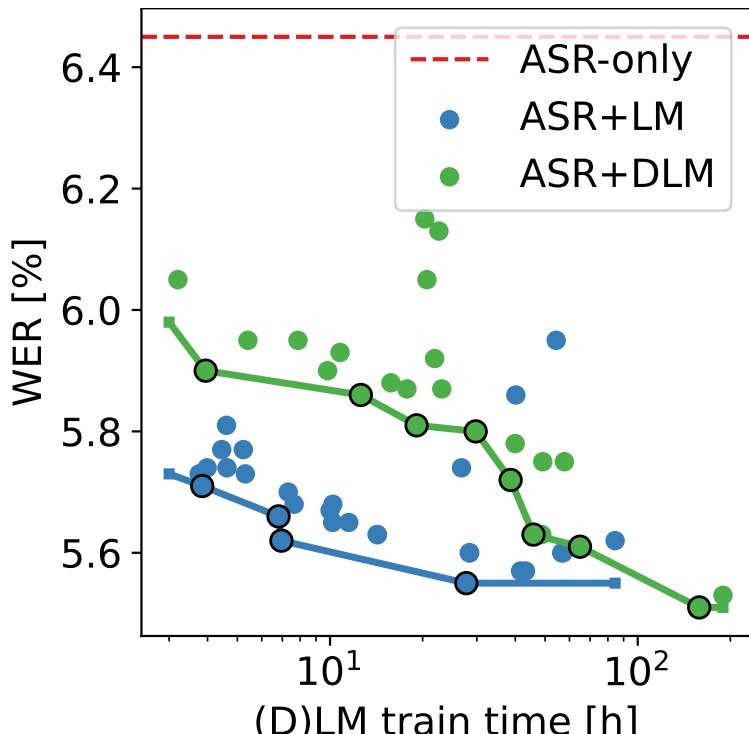

Figure E.28: Scaling plot for Loquacious ASR with LM and DLM.

## E.6 EXPERIMENTS ON ANOTHER CORPUS: LOQUACIOUS

Loquacious (Parcollet et al., 2025) is a large-scale English speech corpus consisting of 25,000 hours of transcribed speech data from diverse sources such as audiobooks, podcasts, YouTube videos, and more.

In contrast to LibriSpeech, there is no separate LM corpus provided with Loquacious. This simplifies the LM vs. DLM comparison: The LM is simply trained on the transcriptions of the ASR training data, while the DLM is trained on hypotheses generated from the ASR model on the speech data of Loquacious. No TTS model is used in this case.

We train a baseline CTC ASR model on Loquacious with the same architecture and hyperparameters as our LibriSpeech ASR models, using a SentencePiece 10k vocabulary.

The main results are shown in Section 4.1, Table 3. Scaling plots are shown in Figure E.28. We see similar trends as with LibriSpeech: The DLM outperforms the LM. For lower training compute budgets, the standard LM performs better, but with increasing training compute the DLM overtakes the LM. Also, the LM starts to overfit with more training compute.

Note that these results are currently the best published WERs on Loquacious, but apart from the original Loquacious paper (Parcollet et al., 2025), there are no other published results on Loquacious for comparison yet.

A recognition speed comparison of ASR + LM vs. ASR + DLM on Loquacious with one-pass search is shown in Figure E.29. This is consistent to the recognition speed comparison on LibriSpeech in Figure 3a.

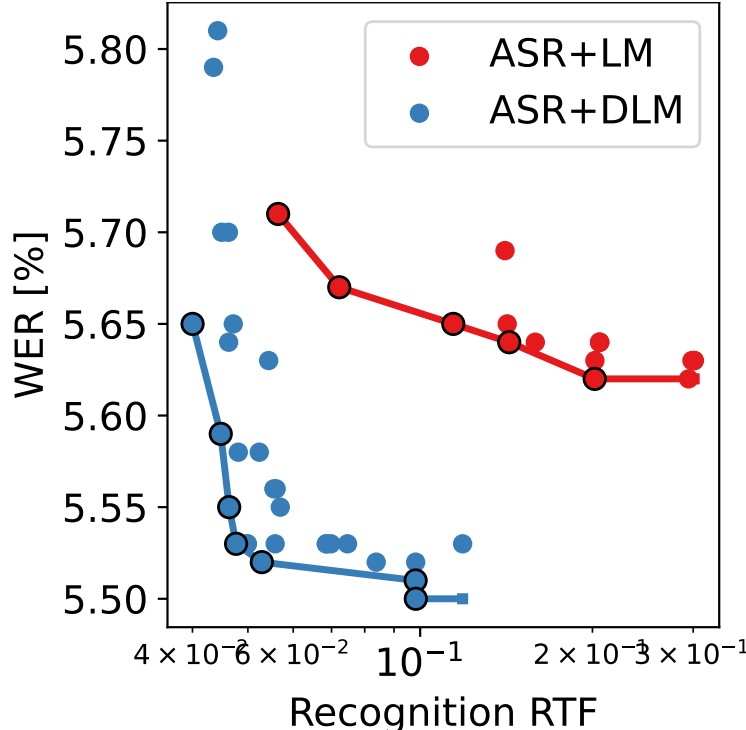

Figure E.29: Recognition speed comparison of ASR + LM vs. ASR + DLM on Loquacious with one-pass search in terms of real-time-factor (RTF).

## F    Implementation Details

### F.1    Bugs

During our research, we encountered some implementation bugs which we noticed and fixed, and we briefly mention them here for completeness.

To use SpecAugment and dropout during training data generation, we enable the training mode flag during ASR model inference. This inadvertently also triggers BatchNorm layers to use the batch-local statistics rather than global running mean and variance, which degrades the ASR model performance noticeably. We fix this by implementing a more granular interface for training mode flags on individual functions.

Our word-to-phoneme lexicon has multiple entries for the same word with different phoneme representations. The TTS training code however only uses the first entry it finds, which leads to out-of-domain behaviour when we pick a random entry during training data generation inference of the TTS model. See Appendix E.2.12 for more details.

At the beginning of our research, the inference of TTS and ASR models was separate, which caused huge load on our filesystems when 75 parallel jobs read TTS audios simultaneously. We fix this by combining the two inference steps into one job, so that only text data has to be loaded and saved.

## G    Extended Conclusions & Discussion

We implemented a data generation pipeline that transforms text data into ASR hypotheses with configurable data augmentation techniques and TTS systems. We present several data augmentation techniques adapted from the field of ASR that can be used to generate more diverse hypotheses for DLM training.

During evaluation of the data augmentation techniques, we found improvements in DLM performance for: TTS noise, SpecAugment, dropout, token substitution, mixup, and by using early ASR checkpoints to generate hypotheses. We found negligible improvements for: TTS audio length scaling, combining TTS systems, top-k sampling and generating multiple epochs of training data. Finally, techniques with a negative impact on DLM performance were normalizing subword splits by retokenizing them before feeding to the DLM in training, using additional phoneme presentations for the TTS which it was not trained on, and only using hypothesis of real audio from the LibriSpeech ASR dataset.

We find that DLM performance generally improves with a higher training data WER, but only up to a certain point. Data augmentation techniques that produce training data with similar WER train DLMs of similar performance, but combining multiple data augmentation techniques leads to better DLM performance than ablations on individual techniques would suggest.

There exists a correlation with the LM score of training data, where DLM performance increases as the LM-score of training data decreases, until a certain point after which DLM performance degrades again. We find that better DLMs have a higher entropy in their output probability distributions ($\leadsto$ less peaky), but artificially increasing the entropy of lower performing DLMs with softmax temperature to match that of better ones does not lead to improvements. There appears to be a trend between entropy of the output probability distribution and Expected Calibration Error, a metric describing the mismatch between predicted probability and actual accuracy.

DLM performance tends to go up with the number of epochs, but we hit a performance ceiling after 10 epochs of training. LM performance seems to saturate already at 5 epochs. I.e. when constrained to a 5 epochs training budget, a traditional LM matches a DLM. But when given more epochs, the DLM surpasses the LM.

Search errors for DLMs are $\leq 1\%$ for all test sets, but model error goes up to $39.6\%$ on test-other.

Real audio does not appear to be important for DLM training, as we can train DLMs with only TTS data and achieve good performance (see Appendix E.2.15). Only LibriSpeech ASR 960h training data, however, is not enough to train a DLM that improves over the ASR baseline (see Appendix E.2.16).

We classify DLM corrections by word frequency and part of speech tag, but do not find strong evidence that DLMs are better at correcting certain categories of words than others. We find that with greedy decoding, the DLM corrects about as many words as it miscorrects previously correct words in the ASR hypothesis. With DSR and DLM-sum decoding, the number of miscorrections decreases drastically, causing a big jump in performance compared to greedy decoding. Looking closer, we find that the DLM exhibits broken behaviour with greedy decoding for a small amount of very long sentences in the test-other set, four of which are responsible for an absolute increase of 0.35% WER on this test set.

We plot the distribution of the WER of individual sentences in the training datasets and their length, and find that their distribution shifts significantly towards higher WER with data augmentation, but their length stays roughly the same.

We trained DLMs with the vocabularies spm10k, spm128 and character, and found that spm10k performs best, but this may be caused by our ASR models for the other vocabularies being worse.

Initial experiments with looping the DLM, i.e. feeding its output back in as input and iterating it multiple times, did not lead to improvements (see Appendix E.4.2), but an architecture closer to diffusion models could potentially yield better results.

## G.1 COMPARISON TO DIFFUSION LANGUAGE MODELS

In Figure 2: There is a tipping point, both in training time and model size, after which denoising LMs start to outperform standard LMs, while for smaller compute budgets standard LMs are better. This is similar to recent observations for diffusion language models under a fixed data-constrained setting (Prabhudesai et al., 2025; Ni et al., 2025).

Diffusion language models share other similarities with DLMs as well. Both operate with an encoder on a corrupted input sequence, and are trained to reconstruct the original sequence. During

training, it is crucial that the model sees a wide variety of corruptions, which is achieved with a noise schedule in diffusion LMs, and with data augmentation techniques in DLMs. We assume this necessity of diverse corruptions is the reason why both diffusion LMs and DLMs need more compute to outperform standard LMs. On the other hand, we believe that operating on the whole sequence with an encoder allows both diffusion LMs and DLMs to better model global context, which leads to better scaling behavior with more compute.

There are also differences between diffusion LMs and DLMs. Diffusion LMs denoise the input with noise in multiple steps, while DLMs use a single step. Also, our DLMs uses an autoregressive decoder, while diffusion LMs typically are non-autoregressive encoder-only.

### G.2 CHARACTER VS. SUBWORD VOCABULARY

Gu et al. (2024) still has better absolute WER results. We hypothesize that this is due to their use of a character vocabulary, while we use subword vocabularies. We realized that a character-based hypothesis gives richer information to the DLM, as only some characters in a word may be wrong, still allowing the DLM to recognize the word, while a subword-based hypothesis may have the whole word wrong.

This was one motivation behind the dense k-probabilities (Section 4.4) as input to the DLM, to give the DLM more information about which subwords are likely to be correct. And indeed, our preliminary experiments with dense k-probabilities showed promising results.

The DLM-sum decoding method is another way to combine the information from multiple ASR hypotheses, which we found to consistently outperform greedy and DSR decoding.

Our character-based ASR model is worse than our subword-based ASR models, which may also contribute to the worse performance of the character-based DLM. More investigation is needed on our character-based ASR model and DLM. We note that the character-based DLM is quite a bit slower than the subword-based DLMs, as the sequence length is much longer.

## H USAGE OF LARGE LANGUAGE MODELS

We used large language models such as ChatGPT or Gemini to help with writing and proofreading parts of this paper.

