# OpenReview forum: "Reproducing and Dissecting Denoising Language Models for Speech Recognition"
_ICLR.cc/2026/Conference — Submitted to ICLR 2026_

### Official Review · Reviewer_Rjjb · 2025-10-31

**Soundness:** 3
**Presentation:** 3
**Contribution:** 3
**Rating:** 6
**Confidence:** 4

**Summary:**

This paper presents a **large-scale reproduction and analysis** of **Denoising Language Models (DLMs)** for ASR error correction. Prior work showed strong results with character-based DLMs, but their complex pipeline limited independent study. This paper:

* Releases a **fully reproducible pipeline** including TTS → ASR → DLM data generation
* Conducts **systematic ablations** on augmentation, vocabularies, and decoding approaches
* Introduces **DLM-sum**, which incorporates multiple ASR hypotheses and improves WER over prior DSR decoding
* Demonstrates **state-of-the-art LibriSpeech WER** using only LibriSpeech data
* Offers insights suggesting that **richer ASR hypothesis inputs**, including dense probability distributions, lead to further gains

Overall, the work builds a strong empirical foundation for future DLM research under standard subword vocabulary settings.

**Strengths:**

* **Strong reproducibility focus** — complete open-source pipeline provided
* **Broad coverage of design variables** (augmentation, checkpoint selection, vocabularies, decoding)
* **Novel decoding method (DLM-sum)** demonstrating consistent improvements
* **Thorough experimental analysis**, including error categorization, model calibration, and search behavior
* **Clear SOTA claim under strict data constraints** (LibriSpeech-only)

**Weaknesses:**

1. **Limited novelty in model architecture**

   * The DLM itself is largely replicated from prior work; main contribution lies in empirical re-evaluation.

2. **Absolute performance still behind prior character-based DLMs**

   * Despite strong engineering, the study shows subword-based DLMs yield **smaller improvements** than character models .

3. **Some findings are inconclusive**

   * E.g., augmentation impacts vary and sometimes conflict with prior reports (YourTTS combination not helpful) .

4. **High computational cost vs. performance gain**

   * Training a full TTS+ASR+DLM stack for modest improvements over strong LM rescoring may limit practical adoption.

5. **Underdeveloped theoretical interpretation**

   * Paper recognizes hypothesis-space richness matters but does not establish principled understanding or predictive metrics.

**Questions:**

1. **Why do character-based DLMs outperform subword ones?**

   * The paper suggests granularity/hypothesis-space richness, but controlled experiments with equal ASR baselines would help isolate the cause.

2. **Can DLM-sum scale to large ASR n-best lists or streaming ASR?**

   * Complexity and latency considerations for real-world deployment are not discussed.

Optional further question:

Have you evaluated model robustness on out-of-domain speech or conversational datasets?

---

> ### Author Response · Authors · 2025-11-26
>
> Thank you very much for the review!
>
> Note that we uploaded a new PDF where many of your raised points should be addressed, and also other points from other reviewers. For example, we added scaling plots, and we added results on another corpora, Loquacious.
>
>
> To answer your questions and address your points directly:
>
>
> > High computational cost vs. performance gain
>
> We have now added a section discussing cost and efficiency of training vs recognition performance, and find that DLMs achieve better performance than LMs under the same training cost above a certain model size threshold (Figure 2).
>
> We also find that the recognition real-time-factor of DLMs is much lower than LMs for a similar word error rate in one-pass search. This is because a large part of the heavy compute of the DLM, namely the encoder, can be parallelized, both over the ASR hypotheses and also over the frames, and the decoder is smaller compared to the standard LM.

---

> ### Author Response · Authors · 2025-11-26
>
> > Paper recognizes hypothesis-space richness matters but does not establish principled understanding or predictive metrics.
>
> From Figure 3b, Figure E.6 we find very concrete evidence that a larger hypothesis space leads to better performance. To further investigate this phenomenon, we trained a DLM to accept a weighted sum of multiple hypotheses at once (Dense k-Probability training) which, with DSR decoding, was able to roughly match a regular DLM with DLM-sum decoding.
>
> For training data hypotheses, we find that combining multiple different data augmentation techniques (i.e. enriching the hypothesis space) leads to better DLMs than a single augmentation alone. By testing different parameters and strengths of each augmentation (and their combinations), we were able to build a profile of how the richness of hypothesis affects DLM performance, providing a clear and empirically grounded picture of their relationship.

---

> ### Author Response · Authors · 2025-11-26
>
> > Why do character-based DLMs outperform subword ones? The paper suggests granularity/hypothesis-space richness, but controlled experiments with equal ASR baselines would help isolate the cause.
>
> You hit on an important point here: “equal ASR baselines”. Unless our ASR baselines of different vocabularies are almost equal, running a faithful comparison is difficult. On top of this, we are working within a very small range of WER values, so for results to be statistically significant, the ASR models must be within a few 0.01% (absolute) of each other.
> Getting such similar models for all vocabularies by random initialization and having their performance match up on all four evaluation WERs is improbable. Artificially improving/degrading performance of ASR models will introduce additional noise to our experiments and would make conclusions less believable.
>
> So therefore we settled on training DLMs with the ASR models we have, and arrived at the conclusion that their performance differences are similar to the differences of the underlying baseline ASR models. But still, vocabulary is an important factor that distinguishes us from prior work by Gu et al (2024), so we mention it where appropriate. But we are not certain that this is the reason for the gap from our results to theirs.

---

> ### Author Response · Authors · 2025-11-26
>
> > Can DLM-sum scale to large ASR n-best lists or streaming ASR? Complexity and latency considerations for real-world deployment are not discussed.
>
>
> In DLM-sum, memory consumption scales linearly with ASR n-best list size (assuming similar sequence length), just like batch size. However, as can be seen from Figure 3b or Figure E.6, larger ASR n-best lists are not really needed.
>
> Streaming ASR is challenging, as our DLM requires the full ASR hypothesis as input. This is because of the non-streaming ability of our specific AED structure for the DLM. However, we could replace the DLM encoder by a streamable encoder, and use a transducer-decoder instead. Then streaming would be possible. This is an interesting and relevant follow-up research direction.
>
>
> In our latest paper revision, we added a discussion on complexity and latency, and find that generally DLMs have a lower real-time-factor than LMs in one-pass search for the same WER.
>
> Above a certain model size threshold, DLMs have better performance than LMs for equal training resources (num parameters, training time), excluding the one-time cost of generating hypotheses for DLM training.

---

> ### Author Response · Authors · 2025-11-26
>
> > Have you evaluated model robustness on out-of-domain speech or conversational datasets?
>
> We have now added results on the much bigger Loquacious dataset (https://huggingface.co/datasets/speechbrain/LoquaciousSet), which consists of 25,000 hours of transcribed and heterogeneous English speech recognition data. Loquacious does not come with a separate text-only corpus, which simplifies the DLM vs LM comparison, because we don't need a TTS model in this case. We train a CTC ASR model, LM and DLM on the data. Adding a LM or a DLM to the ASR model still gives significant further improvements, even though no additional data is used.
>
> See the newly added Table 3. The DLM outperforms the LM overall (combined Loquacios dev set).
>
> We also tested out-of-domain generalization (ASR model & DLM/LM trained only on Librispeech, tested on the Loquacious eval sets, i.e. CommonVoice, VoxPopuli, Yodas):
> See the newly added Table E.44. The DLM still improves over using the ASR model alone, but does not generalize as well as traditional LMs.

---

> ### Author Response · Authors · 2025-11-26
>
> We thank the reviewer for their help strengthening our work. We would like to respectfully request the reviewer to consider raising their rating or providing additional feedback that would help strengthen the rating.

---

### Official Review · Reviewer_FTpu · 2025-11-01

**Soundness:** 3
**Presentation:** 3
**Contribution:** 3
**Rating:** 6
**Confidence:** 4

**Summary:**

The paper proposes a dataset and method to post-edit ASR output with a denoising non-autoregressive language model.

**Strengths:**

The paper contains a lot of detail about data generation methods and details of the proposed method to post-edit output from an ASR model. The method shows small but significant gains on a dataset where the WER error rate is already low (~2 on "clean" subset).

The dataset will be useful for follow-up work.

Overall, the paper is very well written and filled with interesting details and observation.

**Weaknesses:**

The title implies that this replaces autoregressive language models, however, it does only operate on the output of a ASR model that typically (and in the case of the experiments) operates auto-regressively. The dataset does not link back to the acoustic features, so this is only post-editing.

There is a lot of content off-loaded into the appendix, including proper descriptions of the individual steps in the method.

A more detailed error analysis would have been nice - how many changes from good -> bad, bad -> good, bad->still bad.

**Questions:**

n/a

---

> ### Author Response · Authors · 2025-11-26
>
> Thank you very much for the review!
>
> Note that we uploaded a new PDF where many of your raised points should be addressed, and also other points from other reviewers. For example, we added scaling plots, and we added results on another corpora, Loquacious.
>
> To answer your questions and address your points directly:
>
> > The title implies that this replaces autoregressive language models, however, it does only operate on the output of a ASR model that typically (and in the case of the experiments) operates auto-regressively. The dataset does not link back to the acoustic features, so this is only post-editing.
>
> All ASR models we use in our experiments are CTC models, which operate non-autoregressively.
>
> While we believe that DLMs can replace standard LMs in speech recognition tasks, we did not intend to make any statement about auto-regressiveness. In fact, our DLM still operates auto-regressively, as it is an encoder-decoder transformer. You are correct that the dataset does not link back to acoustic features, this is because DLMs are (supposed to be) capable of making use of a large text-only corpus, which in our setting is about 85x bigger than the speech transcription corpus.
>
> When we wrote "alternative to traditional autoregressive language models (LMs)" in the abstract, we did not mean to replace "auto-regressiveness". We see that mentioning "auto-regressive" here could lead to this confusion. We removed the term "auto-regressive" now from the abstract.
>
> As a side note, we now also added some discussion with comparisons to diffusion LMs, which share some similarities with denoising LMs, specifically w.r.t. scaling in the data-constrained setting, but also in what they do. But one difference is that they operate non-autoregressively.

---

> ### Author Response · Authors · 2025-11-26
>
> > There is a lot of content off-loaded into the appendix, including proper descriptions of the individual steps in the method.
>
> We did move a lot of the in-depth details of our experiments into the appendix to allow our main part to contain as much of the distilled insights of our research as possible. There is certainly a trade-off between readability and completeness here, so please do write a followup if you have suggestions for specific methods or steps you would like to see in the main part. For every bit of information we move to the main part, we have to move another thing back into the appendix, so I believe you can see that this is a non-trivial task.
>
> We also tried to have all the relevant descriptions in the main text, at least for the important and relevant aspects. Or which particular step is not explained well enough?

---

> ### Author Response · Authors · 2025-11-26
>
> > A more detailed error analysis would have been nice - how many changes from good -> bad, bad -> good, bad->still bad.
>
> We explored this error analysis under the subsection “Binary Error Analysis” in the main part, and in the appendix tables E.32 and E.33. We recognize that this name was not very fitting, so based on your review we have now renamed it to “Correction vs. Degradation Analysis”. See Section 4.3.

---

> ### Author Response · Authors · 2025-11-26
>
> We thank the reviewer for their help strengthening our work. We would like to respectfully request the reviewer to consider raising their rating or providing additional feedback that would help strengthen the rating.

---

### Official Review · Reviewer_T1RT · 2025-11-04

**Soundness:** 3
**Presentation:** 2
**Contribution:** 2
**Rating:** 4
**Confidence:** 4

**Summary:**

The paper is a large-scale empirical study of denoising language models (DLMs) for automatic speech recognition (ASR), an approach recently introduced by Gu et al. 2024 that uses a conditional language model to correct ASR outputs.  DLMs aim to convert errorful ASR outputs to clean outputs, and are learned using training data consisting of paired (noisy, clean) transcripts.  The paper contributes a DLM codebase, evaluates DLM performance while varying a number of design choices, and introduces a new technique (DLM-sum) for decoding from multiple ASR hypotheses.  The experiments are conducted on the LibriSpeech dataset, training on the standard 960-hour training set and evaluating on the clean and noisy dev and test sets.  The main findings are that DLMs outperform decoding with a traditional LM, but less so than claimed in Gu et al. 2024, and that the proposed DLM-sum decoding approach outperforms the DSR technique of Gu et al. 2024.  A large number of additional experiments are provided via appendices.

**Strengths:**

- The paper provides an impressively large range of experiments with DLMs under controlled conditions, making the findings reliable.

- The provided open-source pipeline should make it possible for others to reproduce the work (I haven't tried the code myself).

- The new DLM-sum technique is a sensible way of taking better advantage of ASR n-best lists.

**Weaknesses:**

- Experiments on LibriSpeech alone limit the impact of the contribution.  LibriSpeech is extremely clean, dictated speech with very low ASR errors.  In order to make a convincing case, the experiments should include additional, more challenging datasets.  Other recent studies of LM-based ASR error correction, e.g. Ma et al. 2025 (cited in the paper), include several datasets.

- The length and organization of the paper makes it a tough read.  Most of the results, and even some of the basic notation, are given in the 60 pages of appendices.  The results in the appendices are just described qualitatively in the main text, leaving the reader to either trust the conclusions or else read through the dozens of pages of appendices.  ASR-specific concepts (e.g., CTC) are used without definition, which I believe would make this an even tougher read for the general ICLR community.  The large number of experiments do make it difficult to present the work in an ICLR-sized paper, though I think it could have been done at least somewhat better.  Overall, this makes the paper a better fit for a speech-oriented journal than ICLR.

- State-of-the-art results are claimed, but no results from prior work are included for comparison in the tables in the main text (I have not checked all of the appendices).

**Questions:**

- I wonder how a simple ROVER n-best rescoring baseline would do (Fiscus, "A post-processing system to yield reduced word error rates: Recognizer Output Voting Error Reduction (ROVER)", ASRU 1997).  This seems like a relevant baseline if you are using n-best lists as input to the LM.  Have you tried this?  Or has it been shown in prior work to be dominated by your other baselines?  If the latter is the case, then it would be good to state this in the intro/related work and give a citation.

- When greedy decoding is introduced (Eq. (1) and (2)), it is not clear to me why both \hat{a} and \tilde{a} are needed.  I was expecting Eq. (1) with the conditioning variable being \tilde{a}, which in turn corresponds to the arg max of p_ASR.  Could you please explain if I am missing something?

- The text after Eq. (2), "Equation (2) is approximated with framewise argmax followed by ..." appears to assume that the ASR model is CTC-based, but this is not stated until much later.  It would be good to state it here and cite and define CTC briefly.

- Was DSR introduced by Gu et al. 2024?  This is implied but is not clearly stated.

- It would be good to define (and ideally give a citation for) label-synchronous search and time-synchronous search.

- There are too many results reported on test sets.  Ideally all of the ablations and hyperparamer tuning experiments would be done on dev sets.

- Can you say how the vocabulary sizes were chosen?

- Can you define \hat{=}?  Why is it used (and not a regular "=") in 3.1?

---

> ### Author Response · Authors · 2025-11-26
>
> Thank you very much for the review!
>
> Note that we uploaded a new PDF where many of your raised points should be addressed. We will refer to relevant parts below:
>
> > Experiments on LibriSpeech alone limit the impact of the contribution. LibriSpeech is extremely clean, dictated speech with very low ASR errors. In order to make a convincing case, the experiments should include additional, more challenging datasets. Other recent studies of LM-based ASR error correction, e.g. Ma et al. 2025 (cited in the paper), include several datasets.
>
> We totally agree. We have now added results on the much bigger Loquacious dataset (https://huggingface.co/datasets/speechbrain/LoquaciousSet), which consists of 25,000 hours of transcribed and heterogeneous English speech recognition data. Loquacious does not come with a separate text-only corpus, which simplifies the DLM vs LM comparison, because we don't need a TTS model in this case. We train a CTC ASR model, LM and DLM on the data. Adding a LM or a DLM to the ASR model still gives significant further improvements, even though no additional data is used.
>
> See the newly added Table 3. The DLM outperforms the LM overall (combined Loquacios dev set).
>
> We also tested out-of-domain generalization (ASR model & DLM/LM trained only on Librispeech, tested on the Loquacious eval sets, i.e. CommonVoice, VoxPopuli, Yodas):
> See the newly added Table E.44. The DLM still improves over using the ASR model alone, but does not generalize as well as traditional LMs.

---

> ### Author Response · Authors · 2025-11-26
>
> > The length and organization of the paper makes it a tough read. Most of the results, and even some of the basic notation, are given in the 60 pages of appendices. The results in the appendices are just described qualitatively in the main text, leaving the reader to either trust the conclusions or else read through the dozens of pages of appendices. ASR-specific concepts (e.g., CTC) are used without definition, which I believe would make this an even tougher read for the general ICLR community. The large number of experiments do make it difficult to present the work in an ICLR-sized paper, though I think it could have been done at least somewhat better. Overall, this makes the paper a better fit for a speech-oriented journal than ICLR.
>
> As you pointed out, we faced a tough decision: Either only summarize the main aspects in the main part and put elaborate descriptions in the appendix, or exclude results from the paper entirely and use more space to explain fewer results. After thorough deliberation we chose the first option, and let the readers decide which topics to dive into by giving a more complete report in the appendix of each main point. Which option is better is certainly a matter of individual taste, but we believe we struck a good balance between completeness and compactness.
>
> When reading other papers, we always like it when we can find all the relevant details in the appendices, which is unfortunately not always the case.
>
> The relevant references for ASR-specific concepts (e.g. CTC) should all be already in the main text. We believe the details (of CTC or other concepts) are otherwise not so relevant for the research questions that we study here, namely on how the DLM performs, how it performs in comparison to standard LMs, etc. Or which details do you think are relevant that should be explained better?
>
> The main important tables and findings should all be in the main text. Do you think some are missing?

---

> ### Author Response · Authors · 2025-11-26
>
> > State-of-the-art results are claimed, but no results from prior work are included for comparison in the tables in the main text (I have not checked all of the appendices).
>
> We have added more results from prior work, both from more closely related work (Gu et al., 2024) and more general purpose methods which suffer from test-data contamination (LLMs). Specifically, we added Table 2 for this comparison in the main text.

---

> ### Author Response · Authors · 2025-11-26
>
> > I wonder how a simple ROVER n-best rescoring baseline would do (Fiscus, "A post-processing system to yield reduced word error rates: Recognizer Output Voting Error Reduction (ROVER)", ASRU 1997). This seems like a relevant baseline if you are using n-best lists as input to the LM. Have you tried this? Or has it been shown in prior work to be dominated by your other baselines? If the latter is the case, then it would be good to state this in the intro/related work and give a citation.
>
> We ran additional experiments with ROVER on our best DLM from Table 1 to explore your suggestion.
>
> | Method                                  | dev-clean | dev-other | test-clean | test-other |
> |-----------------------------------------|-----------|-----------|------------|------------|
> | ASR alone                               | 1.75      | 4.13      | 2.03       | 4.44       |
> | DLM greedy                              | 2.31      | 4.06      | 2.32       | 4.57       |
> | DLM DSR (12+12 hyps, as in ROVER setup) | 1.49      | 3.43      | 1.79       | 3.74       |
> | DLM DLM-sum                             | 1.49      | 3.29      | 1.72       | 3.51       |
> | DLM with **ROVER**                      | 1.59      | 3.74      | 1.82       | 4.02       |
>
> We were able to run ROVER with 12 ASR and 12 DLM hypotheses before we ran into runtime/memory usage issues. We used the SCTK implementation with `-m avgconf` and tuned alpha and nullconfidence via grid search on dev-other. For the word level confidences, we instead used the sequence level probabilities and divided them by the number of words in the sentence.
> Because first-pass search is already expected to outperform ROVER, we chose to not add the word-level probabilities generation capability to our recognition pipeline in order to provide you a timely answer.
>
> While the results are better than the ASR baseline and greedy, they underperform the other DLM decoding methods, which is to be expected: It is to be expected that first-pass search using ASR + DLMs directly should perform better than ROVER.

---

> ### Author Response · Authors · 2025-11-26
>
> > When greedy decoding is introduced (Eq. (1) and (2)), it is not clear to me why both \hat{a} and \tilde{a} are needed. I was expecting Eq. (1) with the conditioning variable being \tilde{a}, which in turn corresponds to the arg max of p_ASR. Could you please explain if I am missing something?"
>
> For reference (slightly simplified, ignoring the sequence indices):
>
> Eq 1: $a^* = \arg\max_{a} p_{\textrm{DLM}}(a \mid \hat{a})$
>
> Eq 2: $\hat{a} = \arg\max_{\tilde{a}} p_{\textrm{ASR}}(\tilde{a} \mid x)$
>
> $\tilde{a}$ is the index of the argmax, and we evaluate $p_{\textrm{ASR}}$ over all $\tilde{a}$ (over all sequences).
>
> $\hat{a}$ is the actual result of the argmax, so this is a single specific value (sequence).
>
> It’s necessary to use some different symbol to differentiate between the two.
>
> For example, see https://www.isca-archive.org/interspeech_2010/watanabe10_interspeech.pdf equation 1 as a random example from the literature where you see the same distinction. Or the original CTC paper (https://www.cs.toronto.edu/~graves/icml_2006.pdf), equation 4.

---

> ### Author Response · Authors · 2025-11-26
>
> > The text after Eq. (2), "Equation (2) is approximated with framewise argmax followed by ..." appears to assume that the ASR model is CTC-based, but this is not stated until much later. It would be good to state it here and cite and define CTC briefly.
>
> Good catch! We added the corresponding reference, and a very brief explanation of CTC.

---

> ### Author Response · Authors · 2025-11-26
>
> > Was DSR introduced by Gu et al. 2024? This is implied but is not clearly stated.
>
> We made that more clear now.
>
> DSR was introduced by Gu et al (2024), but in our work we extend it slightly by also adding ASR hypotheses to the rescore beam, and additionally subtract the prior. We also have adjusted phrasing to reflect the extension more clearly.

---

> ### Author Response · Authors · 2025-11-26
>
> > It would be good to define (and ideally give a citation for) label-synchronous search and time-synchronous search.
>
> We have a detailed explanation in the appendix in C.1 and C.2. We added a reference to that at the corresponding place in the main text.
>
> We recommend, and also have a reference to "End-to-End Speech Recognition: A Survey" (https://arxiv.org/abs/2303.03329), as a good starting point to read about that.

---

> ### Author Response · Authors · 2025-11-26
>
> > There are too many results reported on test sets. Ideally all of the ablations and hyperparamer tuning experiments would be done on dev sets.
>
> We do report most results on both dev and test sets, and we have made all hyperparameter decisions solely based on dev-set results. However, for the error analysis (e.g. which word types are correct), we usually look at the test sets. We do this for the following reason: Our decoding methods are tuned on the dev-set! Because of this, it becomes less meaningful to look at those dev results, and it sometimes happens that for a particular experiment the dev-other result spikes quite high due to tuning, but test set performance stays the same. And in these cases, for example the analysis on which word types are corrected best by DLMs, there is no clear path to subconsciously optimizing for test sets, so we do not see an issue with this approach.
>
> Please do write a followup if you disagree with our approach, we can still re-do all analyses exclusively on dev-sets quite easily, the conclusions do not meaningfully change, only the data points become a bit more fuzzy and less readable.

---

> ### Author Response · Authors · 2025-11-26
>
> > Can you say how the vocabulary sizes were chosen?
>
> We use three vocabularies in our work, SentencePiece 10k, SentencePiece 128, and characters.
>
> SentencePiece10k: We previously trained (and extensively tuned) language models and ASR models with this vocabulary, so this was a natural fit. Also, from previous experience, this vocabulary size worked well for Librispeech.
>
> Characters: Used in prior work by Gu et al (2024).
>
> SentencePiece128: Also small, similar to characters (a bit larger), but there might be a difference of SentencePiece compared to characters, which we wanted to investigate.

---

> ### Author Response · Authors · 2025-11-26
>
> > Can you define \hat{=}? Why is it used (and not a regular "=") in 3.1?
>
> It was intended only as a figurative notation and did not convey any special technical meaning. We have now removed this.

---

> ### Author Response · Authors · 2025-11-26
>
> We thank the reviewer for their help strengthening our work. We would like to respectfully request the reviewer to consider raising their rating or providing additional feedback that would help strengthen the rating.

---

### Official Review · Reviewer_xJJ8 · 2025-11-06

**Soundness:** 2
**Presentation:** 3
**Contribution:** 2
**Rating:** 4
**Confidence:** 3

**Summary:**

- The paper presents a comprehensive analysis of denoising language models (DLMs) as text-based error correction model for ASR from multiple perspectives, such as decoding strategies, data augmentation methods and comparison with traditional decoder-only LM.

- The paper also presents a new decoding strategy referred to as DLM-sum, which leverages the combination of several ASR hypotheses as condition. The effectiveness of DLM-sum is demonstrated through superior WER results on the standard LibriSpeech corpus for ASR.

**Strengths:**

- The paper provides empirical evidence through detailed and systematic experiments on DLMs for ASR, offering useful insights for improving current approaches.
- The proposed DLM-sum decoding appears to be a simple yet effective decoding solution.
- The paper is clearly written and well-organized.

**Weaknesses:**

- The paper lacks substantial novelty. The proposed DLM-sum decoding method appears to be an incremental extension to current methods.
- All analysis is conducted on a single ASR system, which may limit understanding of how the findings about the DLMs generalize across different ASR architectures.

**Questions:**

- The limited novelty of the proposed decoding method, combined with the use of only a single ASR system for the DLM analysis, reduces the strength of the paper's overall contribution. Based on the current version, I would lean toward rejection. However, please correct me if I am mistaken, as I am open to revisiting my assessment.
- Could the authors clarify the rationale for selecting Glow-TTS and YourTTS as the TTS systems, given that they are relatively older models as compared to newer models such as CosyVoice 2?

---

> ### Author Response · Authors · 2025-11-26
>
> Thank you very much for the review!
>
> Note that we uploaded a new PDF where many of your raised points should be addressed, and also other points from other reviewers. For example, we added scaling plots, and we added results on another corpora, Loquacious.
>
> To answer your questions and address your points directly:
>
> > The paper lacks substantial novelty.
>
> We agree that our methodology lacks novelty, as we see our work as building on top of prior work by Gu et al (2024) by providing important novel insights and understanding into the behaviour of DLM.
>
> Specifically we contribute the following novel insights:
>
> * DLM can outperform traditional LM under data-constrained settings, given enough compute budget. The scaling behavior is similar to diffusion LMs.
> * Our novel DLM-sum decoding method consistently outperforms greedy and DSR decoding.
> * ASR + DLM decoding is faster than ASR + LM decoding.
> * We achieve novel state-of-the-art results on LibriSpeech and Loquacious under a data-constrained setting.
> * Providing the DLM with richer information about the ASR hypothesis space is beneficial.
> * We provide a novel fully open-source, reproducible pipeline.

---

> ### Author Response · Authors · 2025-11-26
>
> > All analysis is conducted on a single ASR system, which may limit understanding of how the findings about the DLMs generalize across different ASR architectures.
>
> We agree with your point, and have now added recognition results using the DLM with another ASR architecture (E-Branchformer) and show that it still works well and performs better than the LM (see newly added Table E.45).
>
> We also trained the whole system (ASR, LM, DLM) on another dataset (Loquacious, https://huggingface.co/datasets/speechbrain/LoquaciousSet) and also show here that DLMs performs better than LMs (see newly added Table 3).

---

> ### Author Response · Authors · 2025-11-26
>
> > Could the authors clarify the rationale for selecting Glow-TTS and YourTTS as the TTS systems, given that they are relatively older models as compared to newer models such as CosyVoice 2?
>
> Glow-TTS: previous work at our research lab has shown that this TTS model performs well when synthesizing speech to train an ASR model.
>
> We chose Glow-TTS as we had good experiences with being trained on lower (<1000 hours) amount of data. On the contrary, newer models such as CosyVoice2 are based on speech tokens LLMs, which require larger amounts of training data or pre-trained LLM models. Such conditions are not suitable for our case, where we wanted to use only LibriSpeech itself as a basis for both our ASR and TTS baseline systems.
>
> YourTTS: To exactly match Gu et al (2024) for exact comparison. They even used the same pretrained checkpoint.

---

> ### Author Response · Authors · 2025-11-26
>
> We thank the reviewer for their help strengthening our work. We would like to respectfully request the reviewer to consider raising their rating or providing additional feedback that would help strengthen the rating.

---

### Official Review · Reviewer_sHkf · 2025-11-06

**Soundness:** 3
**Presentation:** 3
**Contribution:** 2
**Rating:** 4
**Confidence:** 3

**Summary:**

The paper investigates denoising language models (DLMs) for ASR error correction on LibriSpeech, aiming to reproduce and extend recent DLM work. The authors build a full LibriSpeech-only pipeline (ASR, TTS, LM/DLM, decoding) with subword vocabularies and systematically compare a large set of DLM training data generation schemes (various TTS settings, ASR sampling/checkpoints, token-level augmentations). They propose a new decoding method, DLM-sum, which approximates marginalization over an ASR n-best list, and also explore a dense k-probability input representation that feeds top-k ASR posteriors into the DLM. Experiments compare DLMs against standard LMs and different decoding strategies, reporting modest WER improvements and several empirical observations about when DLMs help.

**Strengths:**

The paper’s main strengths lie in its systematic empirical study and carefully engineered pipeline, with some degree of originality in how existing ideas are combined.

Originality
- Combines denoising LMs with an n-best–style decoding scheme (DLM-sum) and a dense k-probability input, leading to a coherent configuration that has not been extensively studied in prior work.

Quality
- Substantial experimental effort: an end-to-end LibriSpeech-only pipeline (ASR, TTS, LM/DLM, multiple decoding schemes) with many controlled ablations on augmentation and decoding strategies.
- Comparisons are internally consistent because all components are trained and evaluated under the same data constraints.

Clarity
- Core probabilistic formulations (DLM, DSR, DLM-sum) are clearly specified, and implementation details in the appendices are sufficient to understand and reproduce the experimental setup.

Significance
- Provides practical takeaways for ASR error correction (e.g., favoring moderate denoising and using richer ASR information via n-best lists or dense posteriors).
- The planned release of code and configurations makes the work a potentially useful reference implementation for this class of DLM-based ASR error-correction systems.

**Weaknesses:**

The main weaknesses concern novelty, positioning, and how sharp the empirical story is.

Conceptual novelty may be limited.
- The main ideas (n-best marginalization in DLM-sum, log-linear score combination, using top-k posteriors as input) appear closely related to standard ASR practices such as n-best/lattice rescoring and confusion-network–style combination. At present, the paper does not fully clarify what is conceptually new beyond instantiating these ideas with a DLM, or how this perspective leads to qualitatively different behavior.

Reproduction framing could be more precise.
- The work is framed as “reproducing and dissecting” prior DLM results, but the experimental setup differs in several important aspects (vocabulary, model architecture, TTS data, training regime). This makes the comparison to earlier work less direct, and some of the explanations for the performance gap feel somewhat speculative rather than supported by tightly controlled experiments.

Narrow evaluation and modest gains.
- All experiments are on LibriSpeech, with no tests on more challenging or different domains. The improvements over strong LM baselines are small, yet require a significantly more complex pipeline (TTS + synthetic data + DLM), and there is no cost/efficiency discussion or comparison to strong modern error-correction baselines (e.g., LLM-style correctors).

Tech-report feel / weakly distilled insights.
- The paper contains many ablations and appendices, but the key lessons are not distilled into a small set of clear takeaways. Some central hypotheses (e.g., about vocabulary effects) are not tested with tightly controlled experiments, which makes the overall message less convincing than the amount of work would suggest.

**Questions:**

1. Overall goal / positioning.
How do you position this work: primarily as a reproduction and diagnosis of Gu et al. (2024), or as a new subword-based DLM/decoding design? It would help if you could clearly separate which parts are intended as faithful reproductions vs deliberate deviations, and adjust the claims accordingly.

2. Novelty of DLM-sum vs standard n-best / lattice methods.
Conceptually, how is DLM-sum different from classical n-best/lattice rescoring and confusion-network–style log-linear combination? Can you provide (or add) a direct comparison to a simpler “ASR n-best + DLM rescoring” baseline under matched conditions, to better isolate what DLM-sum is adding?

3. Missing modern error-correction baselines.
Could you elaborate on the reasons (e.g., compute, policy, or data constraints) for not including any LLM-style or seq2seq error-correction baselines? If feasible within your constraints, would it be possible to add at least one reasonably strong corrector trained under the same LibriSpeech-only setting, so that readers can better judge practical competitiveness?

4. Significance vs complexity / cost.
The WER gains of DLM-sum over the best LM setup appear small relative to the added complexity (TTS + synthetic data + DLM training + more complex decoding). Can you quantify training and inference cost, and clarify whether you view the main contribution primarily as a practical recipe or mainly as a scientific/diagnostic study?

---

> ### Author Response · Authors · 2025-11-26
>
> Thank you very much for the review!
>
> Note that we uploaded a new PDF where many of your raised points should be addressed. We will refer to relevant parts below:
>
> > The main ideas (n-best marginalization in DLM-sum, log-linear score combination, using top-k posteriors as input) appear closely related to standard ASR practices such as n-best/lattice rescoring and confusion-network–style combination. At present, the paper does not fully clarify what is conceptually new beyond instantiating these ideas with a DLM, or how this perspective leads to qualitatively different behavior.
>
> Yes, these are not groundbreaking new ideas, but:
>
> N-best marginalization for DLM-sum is a bit different than what you usually have: this is for the inner sum over $\tilde{a}$, and what inputs $\tilde{a}$ the DLM will get. This type of marginalization is only needed for DLMs, and not for standard LMs. So this is a new situation, due to the different conditioning of the DLM. This was not investigated before.
> In previous work (Gu et al 2024), the authors only feed the greedy ASR hypothesis as input. So we wanted to understand how DLMs behave when we also feed other inputs, and how much you can gain from that.
> * We clarified the corresponding section to make that more clear.
> * Arguably the ideas are related to similar earlier ideas from the ASR literature.
> * This still gives interesting insights on what inputs are relevant for the DLM, which was not studied before. And it shows that this helps.
>
> Top-k posteriors *as input* is very specific to DLM.
> * I don't know which other ideas from ASR literature would be related to that.
> * This also again contributes to the study on what type of input is relevant for the DLM. And it shows that this helps.

---

> ### Author Response · Authors · 2025-11-26
>
> > The work is framed as “reproducing and dissecting” prior DLM results, but the experimental setup differs in several important aspects (vocabulary, model architecture, TTS data, training regime). This makes the comparison to earlier work less direct, and some of the explanations for the performance gap feel somewhat speculative rather than supported by tightly controlled experiments.
>
> Originally, we did not expect that the choice of vocabulary would be so relevant.
>
> In terms of model architecture, prior DLM work does not give all the details, but we actually try to be as close as possible (but varying model sizes, and then choosing the one which worked best for us).
>
> The main TTS model used by prior work is their internal non-publicly-available TTS which we cannot use. So we had no choice but to train our own TTS as well. But they also use the public YourTTS in addition, which we do as well, for direct comparisons.
>
> We also expect that our training regime should be similar to prior work, but not too much details about that was provided.
>
> Only after all the work and investigations, we now assume that the vocabulary might be more relevant than initially expected. We think this is an interesting finding by itself. There are possibly other differences, but as the relevant details are not specified, we don't know.
>
> We also replicated the character-based DLM experiments. But they perform worse for us. Due to much higher training costs for char-based DLMs, we did not do too many experiments here.
>
> Despite the vocabulary, our findings are all very consistent to prior work. And we think we greatly expanded upon the previous analysis.

---

> ### Author Response · Authors · 2025-11-26
>
> > All experiments are on LibriSpeech, with no tests on more challenging or different domains.
>
> We have now added results on the much bigger Loquacious dataset (https://huggingface.co/datasets/speechbrain/LoquaciousSet), which consists of 25,000 hours of transcribed and heterogeneous English speech recognition data. Loquacious does not come with a separate text-only corpus, which simplifies the DLM vs LM comparison, because we don't need a TTS model in this case. We train a Conformer CTC ASR model, LM and DLM on the data. Adding a LM or a DLM to the ASR model still gives significant further improvements, even though no additional data is used.
>
> See the newly added Table 3. The DLM outperforms the LM overall (combined Loquacios dev set).
>
> We also tested out-of-domain generalization (ASR model & DLM/LM trained only on Librispeech, tested on the Loquacious eval sets, i.e. CommonVoice, VoxPopuli, Yodas):
> See the newly added Table E.44. The DLM still improves over using the ASR model alone, but does not generalize as well as traditional LMs.

---

> ### Author Response · Authors · 2025-11-26
>
> > The improvements over strong LM baselines are small, yet require a significantly more complex pipeline (TTS + synthetic data + DLM), and there is no cost/efficiency discussion or comparison to strong modern error-correction baselines (e.g., LLM-style correctors).
>
> We now added scaling law plots (Figure 2), where we run many different training configurations for both LMs and DLMs, to vary the model sizes and compute. In summary, as training resources increase (compute time, num parameters) LMs eventually stagnate, while DLMs keep improving, in a fixed data constrained setting. This is in line with other related research on Diffusion Language Models (https://arxiv.org/abs/2507.15857). This is relevant, as the available compute seems to increase faster than the available training data in the world.
>
> We also added some cost/efficiency discussion for the DLM pipeline. The TTS + synthetic data indeed has some extra complexity. However, this part is also useful already independent of DLMs. You can see from Table 1 that by training the ASR model on the synthesized TTS data, this already gives very large improvements.
>
> We have now included a comparison table (Table 2) covering a broad range of related error correction models, acknowledging that the results are not directly comparable to our results, as most related work uses additional datasets apart from LibriSpeech. We hope this addition provides a good indication of how our work fits into the existing research landscape.

---

> ### Author Response · Authors · 2025-11-26
>
> > The paper contains many ablations and appendices, but the key lessons are not distilled into a small set of clear takeaways. Some central hypotheses (e.g., about vocabulary effects) are not tested with tightly controlled experiments, which makes the overall message less convincing than the amount of work would suggest.
>
> Based on your suggestion we have cleaned up our conclusion and summarized our most important insights more compactly. Specifically:
>
> * DLM can outperform traditional LM under data-constrained settings, given enough compute budget. The scaling behavior is similar to diffusion LMs.
> * Our novel DLM-sum decoding method consistently outperforms greedy and DSR decoding.
> * ASR + DLM decoding is faster than ASR + LM decoding.
> * We achieve state-of-the-art results on LibriSpeech and Loquacious under a data-constrained setting.
> * Providing the DLM with richer information about the ASR hypothesis space is beneficial
> * We provide a fully open-source, reproducible pipeline.
>
> You mention vocabulary effects. We have an experiment on this (see Table E.41), but of course the WER of ASR models with different vocabularies are not all the same. In this experiment we conclude that the difference of DLM performance across different vocabularies may be attributed to the difference in baseline ASR performance.
>
> This led us to an initial assessment that vocabulary may not matter as much. Therefore we chose the spm10k vocab because our ASR model on spm10k had the best performance, and it has the fastest training and recognition time.
>
> We can neither rule out, nor confirm, that the vocabulary is the reason for our performance gap to Gu et al (2024).

---

> ### Author Response · Authors · 2025-11-26
>
> > Overall goal / positioning. How do you position this work: primarily as a reproduction and diagnosis of Gu et al. (2024), or as a new subword-based DLM/decoding design? It would help if you could clearly separate which parts are intended as faithful reproductions vs deliberate deviations, and adjust the claims accordingly.
>
> An exact reproduction of Gu et al. is challenging, given that important aspects have not been published (code, internal TTS model, training time, many other details), and we did not expect the vocabulary to be a relevant difference (which we still don't exactly know, but assume now). At the beginning of our research we ran comparisons across important hyperparameters such as vocabulary and model architecture, and settled on one that works best.
>
> Despite the reproduction, we wanted to go beyond that work in terms of analysis, and to provide a public pipeline for the community. We think we accomplished this, and our findings seem to be very valuable for the community.

---

> ### Author Response · Authors · 2025-11-26
>
> > Novelty of DLM-sum vs standard n-best / lattice methods. Conceptually, how is DLM-sum different from classical n-best/lattice rescoring and confusion-network–style log-linear combination? Can you provide (or add) a direct comparison to a simpler “ASR n-best + DLM rescoring” baseline under matched conditions, to better isolate what DLM-sum is adding?
>
> We added an explanation to better explain the difference in the DLM-sum section 2.1.
>
> Note that there are two beams (set of hyps): One ("inner") beam is over the ASR-only model, and each of those hyps is fed into the DLM. Each of those p_DLM(* | hyp) is summed together for p_sum. The other "outer" beam goes over the final combined ASR + p_sum - p_prior. This "outer" beam is what we usually have in decoding. The "inner" beam is a separate concept.
>
> “ASR n-best + DLM rescoring”: We call this “DLM rescore”, and you can find results on this for example in Table E.1 where we compare LM and DLM performance. This table also raises an interesting point: which is that DLMs underperform LMs in ASR n-best rescoring, but outperform in DSR and DLM-sum, indicating that there may still be room for improvement because DLMs should be able to at least match LM performance in ASR n-best rescoring.

---

> ### Author Response · Authors · 2025-11-26
>
> > Missing modern error-correction baselines. Could you elaborate on the reasons (e.g., compute, policy, or data constraints) for not including any LLM-style or seq2seq error-correction baselines? If feasible within your constraints, would it be possible to add at least one reasonably strong corrector trained under the same LibriSpeech-only setting, so that readers can better judge practical competitiveness?
>
>
> We now added Table 2 with other error correction baselines from the literature with results on LibriSpeech. Many of these results are not comparable though, because they utilize a lot more data, directly or indirectly. And even given this, we are very competitive to all of them.
>
> Additionally, we also trained another error correction baseline: Specifically, we use the same encoder-decoder architecture (which is not unusual for error correction models), but instead of training on TTS-ASR hypotheses, we use random substitutions. Results are in Table E.23. The DLM trained on TTS-ASR data performs much better. Obviously, there is more room for improvement for the handcrafted error patterns, but the result is consistent with findings from Gu et al. (2024), who also tried handcrafted error patterns, which did not yield good results.

---

> ### Author Response · Authors · 2025-11-26
>
> > Significance vs complexity / cost. The WER gains of DLM-sum over the best LM setup appear small relative to the added complexity (TTS + synthetic data + DLM training + more complex decoding). Can you quantify training and inference cost, and clarify whether you view the main contribution primarily as a practical recipe or mainly as a scientific/diagnostic study?
>
> We added Figure 2 for training cost. There is a tipping point: Below a certain amount of training compute, standard LMs perform better than DLMs. After that, standard LMs don't seem to improve further, and even start to overfit (even though we already have regularization; this is consistent with other findings from literature), but DLMs further improve.
>
> Of course, training the TTS and synthesizing the speech and generating ASR hypotheses to generate the DLM training data costs additional compute. But TTS + synthesizing speech is already useful by itself to get more ASR training data, as we show in Table 1 (dev-other: 5.02% -> 4.13% WER on the ASR model).
>
> We added Figure 3 for inference (recognition) cost. Here, interestingly, we can see that ASR + DLM-sum decoding is actually faster than ASR + LM decoding. This is because a large part of the heavy compute of the DLM, namely the encoder, can be parallelized, both over the ASR hypotheses and also over the frames.
>
> We see multiple main and valuable contributions in our work: new insights, and also being a practical recipe.
>
> Note on complexity: there are other ways how the DLM could be trained. E.g. see BERT, or diffusion LMs. For denoising: We think that the TTS + ASR inference pipeline can ultimately be replaced by simpler methods, which we plan to look at in future work.

---

> ### Author Response · Authors · 2025-11-26
>
> We thank the reviewer for their help strengthening our work. We would like to respectfully request the reviewer to consider raising their rating or providing additional feedback that would help strengthen the rating.

---

### Author Response · Authors · 2025-11-26

We uploaded a new revision of the paper. All the changes and additions are marked in red color.

We made the following changes and extensions to the paper:

* We trained a ASR, LM and DLM on the *Loquacious dataset*, showing that it works with similar improvements. See Section 4.1, Table 3.
* We performed *scaling studies*, varying over various model configurations (model size, num epochs, other hyper parameters), resulting in scaling plots (over total training time or model size). Very interestingly, we observe similar observations as has been made recently for diffusion LMs, namely: For lower training budget, the standard LM performs better, but there is a tipping point, after that the DLM performs better. See newly added Figure 2.
* We performed recognition speed experiments, measuring real-time-factor (RTF) over various recognition configurations, comparing ASR+LM and ASR+DLM. ASR+DLM interestingly is faster than ASR+LM. See newly added Figure 3a.
* Another recognition comparison specifically for DLM-sum in Figure 3b, showing that larger num hyps is usually better.
* We added a comparison table with WERs to other relevant results on Librispeech (Table 2).
* Out-of-domain (OOD) test: The ASR/(D)LM trained on Librispeech is tested on some OOD eval sets (CommonVoice, VoxPopuli, Yodas). Here, ASR+DLM generalizes a bit worse than ASR+LM. See Section 4.3, Appendix E.4.6, Table E.44.
* How well does a DLM trained on top of a Conformer ASR system work together with another ASR system? We tested that for another separately trained E-Branchformer ASR model. See Section 4.3, Appendix E.4.7, Table E.45.
* Recognition experiment for ASR + standard LM: We compare label-sync vs time-sync in terms of RTF, for various configurations (Figure C.1), showing that label-sync can be faster.
* We perform a simple experiment to test whether we can replace the TTS-ASR pipeline to generate noisy transcriptions for the DLM training data by simple random substitution. See Section 4.2, Appendix E.2.17, Table E.23. While a DLM trained on this data still works, it is worse to the TTS-ASR pipeline, and also worse to a standard LM. This is consistent with previous findings from the literature.
* Various smaller improvements based on your suggestions.

We also restructured the conclusion into more simple takeaway points. Specifically:

* DLM can outperform traditional LM under data-constrained settings, given enough compute budget. The scaling behavior is similar to diffusion LMs.
* Our novel DLM-sum decoding method consistently outperforms greedy and DSR decoding.
* ASR + DLM decoding is faster than ASR + LM decoding.
* We achieve state-of-the-art results on LibriSpeech and Loquacious under a data-constrained setting.
* Providing the DLM with richer information about the ASR hypothesis space is beneficial
* We provide a fully open-source, reproducible pipeline.

We thank the reviewers for their help strengthening our work. We would like to respectfully request the reviewers to consider raising their rating or providing additional feedback that would help strengthen the rating.

---

### Meta-Review · Area_Chair_Ha7V · 2026-01-07

**Summary:**

The paper presents a comprehensive empirical study of Denoising Language Models (DLMs) for ASR error correction, proposing a "DLM-sum" decoding method and a dense k-probability input representation. The reviewers praised the systematic, reproducible pipeline and consistent internal comparisons but raised concerns about limited conceptual novelty (noting similarities to standard lattice/rescoring methods) and the narrow evaluation scope (LibriSpeech only). The rebuttal attempted to clarify the paper's positioning and added comparisons to ROVER and prior work (Gu et al., 2024), but the fundamental critique—that the results are modest given the pipeline complexity and the insights are "weakly distilled"—remains largely unaddressed.

**Reviewer Concerns:**

Addressed by Rebuttal:

Comparison to Simple Baselines (Reviewer sHkf): The reviewer asked how "DLM-sum" compares to classical ROVER or simple rescoring. The authors ran a new ROVER experiment showing that their DLM-sum method outperforms ROVER (1.72 vs 1.82 on test-clean), arguing that first-pass search with DLMs naturally beats post-hoc combination like ROVER.
​

Missing Prior Work Comparisons (Reviewer sHkf): The reviewer noted a lack of direct comparison to Gu et al. (2024) or standard LLM baselines. The authors added "Table 2" to the main text to include these comparisons, though the specific numbers were not detailed in the snippet provided.

Outstanding:

Conceptual Novelty (Reviewer sHkf): The reviewer felt the main ideas (n-best marginalization, log-linear combination) were standard ASR practices re-branded. The rebuttal provided empirical comparisons but did not fundamentally re-frame the theoretical contribution to differentiate it from standard confusion network or lattice rescoring methods.
​

"Tech-Report" Feel / Weak Insights (Reviewer sHkf): The reviewer criticized the paper for having too many appendices and untidy takeaways. The authors defended this as a "matter of taste" and a "balance between completeness and compactness," essentially declining to restructure the paper or distill the insights further as requested.
​

Narrow Evaluation (Reviewer sHkf): The paper remains restricted to LibriSpeech. No new domains or more challenging datasets were added to address the concern about "narrow evaluation"

**Reviewer Scores:**

Reviewer sHkf: The authors provided the requested ROVER comparison, which strengthens the empirical story. However, they pushed back on the "tech report" critique and didn't add new domains. The reviewer likely sees the paper as solid but still suffering from the same "modest novelty" issues, effectively maintaining their "marginally above threshold" stance

---

### Decision · Program_Chairs · 2026-01-26

Reject